# Core and accessory genomic traits of *Vibrio cholerae* O1 drive lineage transmission and disease severity

Alexandre Maciel-Guerra [1,7], Kubra Babaarslan[1,7], Michelle Baker [1,7], Aura Rahman[2], Maqsud Hossain[1,2], Abdus Sadique[2], Jahidul Alam[2], Salim Uzzaman[3], Mohammad Ferdous Rahman Sarker[3], Nasrin Sultana[3], Ashraful Islam Khan[4], Yasmin Ara Begum[4], Mokibul Hassan Afrad[4], Nicola Senin [5], Zakir Hossain Habib[3], Tahmina Shirin[3], Firdausi Qadri [4] & Tania Dottorini [1,6] ✉

In Bangladesh, *Vibrio cholerae* lineages are undergoing genomic evolution, with increased virulence and spreading ability. However, our understanding of the genomic determinants influencing lineage transmission and disease severity remains incomplete. Here, we developed a computational framework using machine-learning, genome scale metabolic modelling (GSSM) and 3D structural analysis, to identify *V. cholerae* genomic traits linked to lineage transmission and disease severity. We analysed in-patients isolates from six Bangladeshi regions (2015-2021), and uncovered accessory genes and core SNPs unique to the most recent dominant lineage, with virulence, motility and bacteriophage resistance functions. We also found a strong correlation between *V. cholerae* genomic traits and disease severity, with some traits overlapping those driving lineage transmission. GSMM and 3D structure analysis unveiled a complex interplay between transcription regulation, protein interaction and stability, and metabolic networks, associated to lifestyle adaptation, intestinal colonization, acid tolerance and symptom severity. Our findings support advancing therapeutics and targeted interventions to mitigate cholera spread.

Cholera is an acute diarrhoeal disease. Worldwide, 1.3 billion people are estimated to be at risk and approximately 1.3 to 4 million cases occur annually, with 21,000 to 143,000 resulting in death[1,2]. In Bangladesh alone, where cholera is endemic, an estimated 66 million people are at risk of cholera with at least 100,000 cases and 4500 deaths per year[1,3]. Globally the O1 serogroup remains the primary cause of cholera[1,2]. The O1 serogroup is divided into the main serotypes Ogawa and Inaba, and subdivided into two biotypes, classical and El Tor (7th pandemic), which are genotypically and phenotypically distinct[4–6]. *V. cholerae* has shown an extraordinary capacity to undergo genetic and phenotypic changes over time, giving rise to successive waves of genetically and phenotypically diverse pandemic clones.

[1]School of Veterinary Medicine and Science, University of Nottingham, College Road, Sutton Bonington, Loughborough, Leicestershire LE12 5RD, UK. [2]NSU Genome Research Institute (NGRI), North South University, Baridhara, Bashundhara, Dhaka 1229, Bangladesh. [3]Institute of Epidemiology, Disease Control and Research (IEDCR), 44, Shaheed Tajuddin Ahmed Sarani Mohakhali, Dhaka 1212, Bangladesh. [4]International Centre for Diarrhoeal Disease Research, Bangladesh (icddr, b), 68, Shaheed Tajuddin Ahmed Sarani Mohakhali, Dhaka 1212, Bangladesh. [5]Department of Engineering, University of Perugia, 06125 Perugia, Italy. [6]Centre for Smart Food Research, Nottingham Ningbo China Beacons of Excellence Research and Innovation Institute, University of Nottingham Ningbo China, Ningbo 315100, P. R. China. [7]These authors contributed equally: Alexandre Maciel-Guerra, Kubra Babaarslan, Michelle Baker. ✉e-mail: tania.dottorini@nottingham.ac.uk

These variants exhibit increased virulence, pathogenicity, resistance and spreading capability[7,8].

Recently, distinctive lineages belonging to the 7th pandemic El Tor (7PET) wave-3 have been observed circulating in Bangladesh[9–11]. The two most prominent circulating lineages identified over the last 20 years are BD-1 and BD-2[9–11], and more recently BD-1.2, responsible for the latest 2022 massive outbreak in the country[10]. Genomic analysis revealed variations between BD-1.2 and BD-2 in the *Vibrio* seventh pandemic island II (VSP-II), *Vibrio* pathogenic island 1 (VPI-1), mobile genetic elements, phage-inducible chromosomal island-like element (PLE), and SXT-related integrating conjugative elements (SXT ICE)[10]. Despite the advances of genomic analysis, the complete genomic repertoire and the mechanisms causing the greater transmission of BD-1.2 remain unknown. Gaps persist in our knowledge regarding whether coding or non-coding single nucleotide polymorphisms (SNPs), or accessory genes, drive the evolutionary shifts. It remains unclear whether gene regulation, metabolic or molecular networks, or folding events play a role. There is even less knowledge about the genomic determinants responsible for the severity of cholera resulting from these lineages. About 1 in 5 people with cholera will experience a severe condition owing to a combination of symptoms (primarily diarrhoea, vomiting and dehydration)[12]. Amongst the major symptoms, watery diarrhoea characteristic of cholera is caused by the cholera toxin (CT)[4–6]. The *V. cholerae* El Tor responsible for the current cholera pandemic has become more virulent by undergoing several changes in CTX genotype and acquiring virulence-related gene islands[13,14].

In this study, we developed a reference-agnostic machine learning method, coupled with genome-scale metabolic modelling (GSMM) and protein structural analysis, to achieve two key objectives as outlined below. The first objective was to identify the genetic variations and signatures of the BD-1.2 lineage evolution beyond what has been found so far[10]. Our analysis considered 129 *V. cholerae* isolates from diarrhoea samples collected between 2015 and 2021, from patients admitted to the icddr,b hospital in Bangladesh. Several genomic studies investigated the evolution of lineages from 1991 to 2017, as well as in 2022[9–11]. However, there remains a gap in research during the intervening period. In our analysis, we discovered a set of 77 SNPs within the coding genome (mapped to 50 known genes), along with 12 annotated accessory genes, including some associated with antibiotic resistance, virulence, motility, colonisation, biofilm formation, acid tolerance and bacteriophage resistance, identified as correlated with BD-1.2 transmission. Our findings go beyond what was recently discovered[9–11] for the lineage.

The second objective was to investigate if correlations exist between the genomic determinants of BD-1.2 strains and clinical manifestations among hospitalised patients from whom the isolates were collected. Machine learning revealed the existence of correlations between genetic determinants in *V. cholerae* and clinical symptoms (duration of diarrhoea, number of stools, abdominal pain, vomiting and dehydration). Overall, the analysis revealed an overlap of 11 mutations, four accessory genes, and one intergenic SNP between the unique genomic determinants associated with BD-1.2 transmission and the clinical symptoms linked to this lineage. Additionally, a distinct set of 17 mutations, 39 accessory genes, and four intergenic SNPs were found exclusively linked to the severity of clinical symptoms. Through detailed GSMMs and 3D structure analysis of these genes, we inferred the mechanistic basis behind the selection of these genomic drivers in BD-1.2 and link to severity of the symptoms.

## Results

### From 2015 to 2021 in Bangladesh, a diverse array of genetic variations characterises the emergence of distinct circulating lineages

To explore the evolutionary dynamics of *V. cholerae* linked to cholera cases in Bangladesh, a genomic analysis was done considering the years 2015 to 2021. We sequenced 129 *V. cholerae* O1 El Tor isolates taken from stool samples of patients between September 2015 and April 2021 admitted to hospitals in six districts (Barisal, Chittagong, Dhaka, Khulna, Rajshahi and Sylhet) of Bangladesh, Supplementary Data 1. Over the duration of this study, isolates belonging to serotypes Inaba and Ogawa were identified, Fig. 1. Consistent with previous studies[10,15], a serotype switch was observed, with Inaba predominantly present in 2016 and 2017, followed by a predominance of Ogawa samples in 2018 and 2019 (Fig. S1). Both serotypes were detected in 2015 and continued to coexist from 2020 onwards. Serotypes were significantly associated with collection years (chi-square test with *p*-value Bonferroni < 0.005) but not significantly associated with collection location (chi-square test with *p*-value Bonferroni > 0.005).

The maximum likelihood phylogeny of the 129 isolates was reconstructed based on the alignment of the core genome (3468 genes) and showed two distinctly evolved lineages, Fig. 1. Comparison with previous studies[9,10], identified these lineages as BD-1.2 ($n = 84$) and BD-2 ($n = 45$), Fig. S2. Apart from the previously reported genetic variations[9,10], we identified additional differences existing between the two lineages in VSP-II, *Vibrio* pathogenic island 2 (VPI-2) and PLE, see Fig. 1. More precisely, in VSP-II, BD-2 isolates had a tryptophan at position 249, while BD-1.2 had a leucine at this position. In addition, in VSP-II, gene VC-514 (*aer*) was present in all BD-2 isolates but absent in BD-1.2. In VPI-2 a SNP led to an amino-acid variation at position 150, with BD-1.2 having an aspartic acid, and BD-2 an asparagine. BD-2 samples exclusively exhibited PLE2, while BD-1.2 samples had both PLE1 and PLE2 along with PLE2. Moreover, further differences were found in nonsynonymous SNPs on core genes and presence/absence of accessory genes, as described in the following section.

The distinct phylogeny patterns of BD-2 and BD-1.2, were also confirmed through a comparative study analysing 1134 isolates from *V. cholerae* El Tor O1 strains across 84 countries, including our isolates, (Supplementary Data 2, 3 and Fig. S3). BD-2 isolates clustered with Indian-1 (IND-1), while BD-1, BD-1.1, and BD-1.2 isolates from Bangladesh clustered with African (T9-T13)[16], Latin America-3 (LAT-3)[13], Asian-2 (AS-2), and Indian-2 (IND-2) lineages (Fig. S3), in agreement with previous results[10].

### Genetic and temporal differentiation of *V. cholerae* BD-1.2 and BD-2 lineages correlate with SNPs in coding and non-coding regions, and accessory genes

To assess the relatedness of *V. cholerae* isolates in our cohort, we measured the number of different core genome SNPs in a pairwise manner across all isolates. We created a network based on clusters of related isolates with less than 15 SNPs, as done previously[17,18]. Across the cohort the median SNP difference was 117 SNPs (ranging from 0 to 1710 SNPs with IQR of 1211). The resulting undirected graph (Fig. 2) revealed that BD-2 and BD-1.2 formed two disconnected graphs each composed of samples from a specific lineage, but with no distinct separations between the Ogawa and Inaba serotypes.

To identify additional potential involvement of genetic elements in shaping the differences between the BD-1.2 and BD-2 isolates in our cohort, beyond current annotations (*ctxB* allele, type of SXT/ICE, VSP-II, VPI-I, *gyrA* gene allele)[10], we looked for patterns of similarities and differences, at a finer scale, searching for the number, type and position of accessory genes as well as mutations in the core genome and intergenic regions across all the isolates. A two-sided Fisher exact test, with Bonferroni correction, was performed to assess the relationship between the BD-2 and BD-1.2 lineages and each of the various genomic features (core and intergenic SNPs and accessory genes). Overall, we found a significantly larger proportion of core genome mutations (51.4%, 1224 core genome SNPs and 73.1%, 160 intergenic SNPs) and a small proportion of accessory genes (11.3%, 115 genes) that exhibited statistically significant differentiation between the two lineages, Supplementary Data 4. Refer to Supplementary Note 1 and Fig. S4 for more

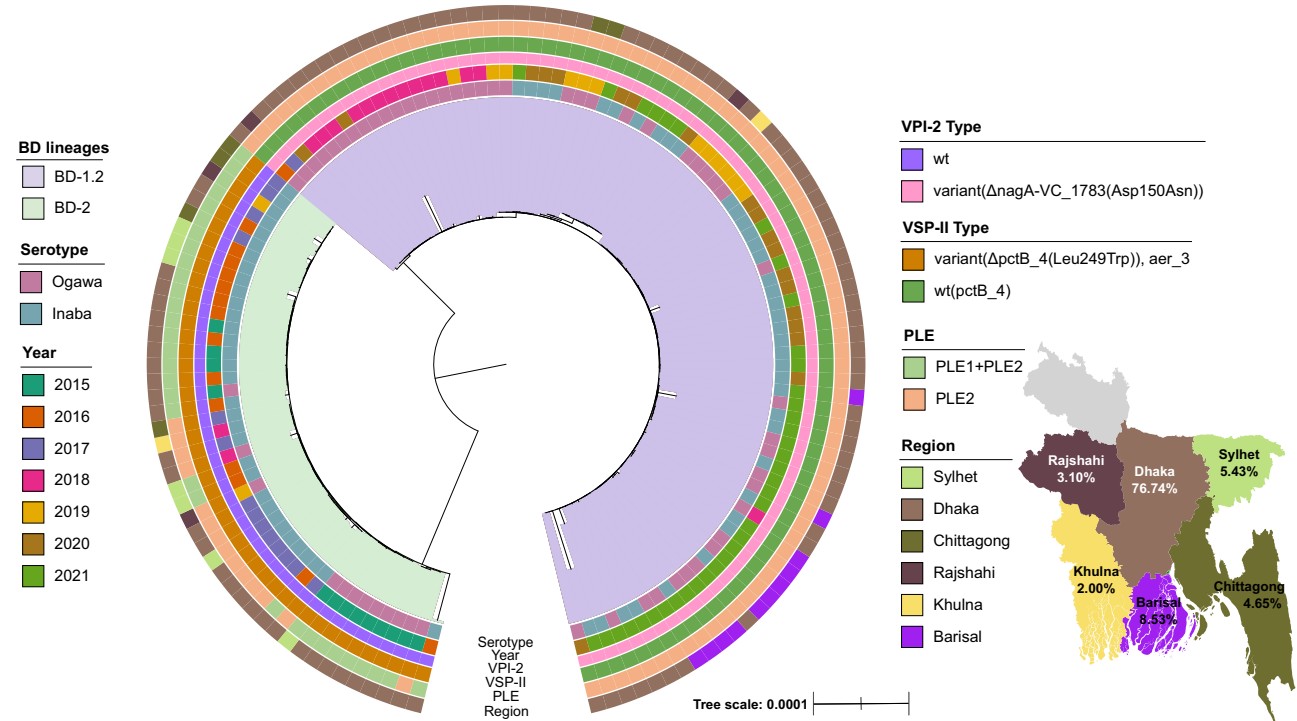

**Fig. 1 | Maximum likelihood phylogenetic tree of the whole cohort based on the core genome of 129 isolates cultured from in-patients admitted to hospitals in six districts (Barisal, Chittagong, Dhaka, Khulna, Rajshahi and Sylhet) of Bangladesh.** The two distinct BD-1.2 and BD-2 lineages are shown in the inner ring. The outer rings display serotypes, year of collection, presence of variants within the *Vibrio* pathogenic island 2 (VPI-2), *Vibrio* seventh pandemic island II (VSP-II) and phage-inducible chromosomal island-like elements 1 and 2 (PLE) and region of collection. A map of Bangladesh[132] showing the proportion of samples collected from each regional division is also shown.

details on the statistical analysis comparing the number of accessory genes, core genome SNPs and intergenic SNPs. The comparative analysis also indicated a temporal shift in the distribution of core genome and intergenic SNPs over the years, showing that BD-1.2 isolates accumulated different SNPs compared to BD-2 isolates as time progressed (Fig. S4E, F).

Out of the 115 accessory genes that differed between the two lineages, 12 were annotated while the remaining 101 were hypothetical. Among these 12 annotated genes, five (*lon_3, endA, adh, hdfR_4* and *bcr_2*) were predominant (over 96% presence) in BD-1.2 and absent in BD-2, and seven (*aer_3, hlyA_2, mcrC, mepM_3, mrr, tetA* and *tetR*) were present (over 97% presence) in BD-2 and absent in BD-1.2. Of the twelve annotated genes, three are known to be antimicrobial resistance genes (*bcr, tetA* and *tetR*)[19]. *TetA* and *tetR* were mainly detected in BD-2 isolates (97.7%), confirmed as primarily tetracycline-resistant through susceptibility testing in both doxycycline and tetracycline antibiotics (Supplementary Data 1). On the contrary, *bcr*, a multidrug efflux pump, was predominantly present in BD-1.2 isolates (96.4% of isolates) and completely absent in BD-2 isolates. Out of the 16 known antimicrobial resistant genes (ARGs) present in the pangenome of this cohort, only *tetA, tetR* and *bcr* were found to statistically separate both lineages. *TetA* and *tetR* were both located in a contig showing high similarity to the SXT-ICE element, SXT(HN1) in BD-2 isolates. Conversely, *bcr* was found in a mobile element in the BD-1.2 isolates with similarity to SXT ICE element, ICE*Vch*Ban5. The presence of these SXT elements in the BD-2 and BD-1.2 lineages was previously shown by Monir et al.[10]. Both contigs contained two identical insertion sequences, mobile genetic elements MGEs, (IS*Shfr9* and IS*Vsa3*), see Fig. S5. Also, among the 12 annotated genes, four (*endA, hlyA, lon* and *mcrC*) were previously found to be related to virulence[18–23]. More information about the function of these genes is given in the Supplementary Note 2.

To assess the extent of our results beyond our cohort, we investigated whether the 12 annotated accessory genes that we had found were also present in other Bangladeshi and Indian lineages. We performed a comparative genomic analysis of 219 *Vibrio cholerae* O1 reference isolates collected in Kolkata, India, and Dhaka, Bangladesh, between the years 2004 and 2022 (ENA public database http://www.ebi.ac.uk/ena, see Supplementary Data 5). The results confirmed the presence/absence patterns of the 12 genes in the BD-1.2 and BD-2 lineages in the reference isolates, aligning with our initial findings, see Supplementary Note 2.

In addition to differences in accessory gene types and patterns, missense mutations associated to allelic variations were found in BD-1.2, when compared to BD-2 strains. We identified 1385 SNPs in the core genome, including 291 non-synonymous and 934 synonymous coding variants, both representing variants in their functional protein-coding form. In addition, 160 intergenic SNPs were found, representing variants in their regulatory form. Many SNPs showcased unique allelic distribution patterns between the two lineages. When mapped back, the non-synonymous SNPs identified 291 amino acid substitutions in 105 genes, including 50 known genes and 55 hypothetical ones (see Supplementary Data 4). Table S1 shows core genes with allelic distribution between BD-1.2 and BD-2 significantly different (i.e., containing polymorphic sites found exclusively in one lineage but absent in the other lineage).

Among the genes exhibiting lineage-specific allelic variation, some contribute to functions including growth, cell wall organisation, colonisation, toxigenicity and resistance, similar to what found previously[10]. Additionally, we found genes with a unique non-synonymous variant in BD-1.2, with roles in toxin transport and acid tolerance, shedding light on functions that may clarify their contribution to the recent prevalence of BD-1.2 over BD-2. See Supplementary Note 3 for more information about these genes. Notably,

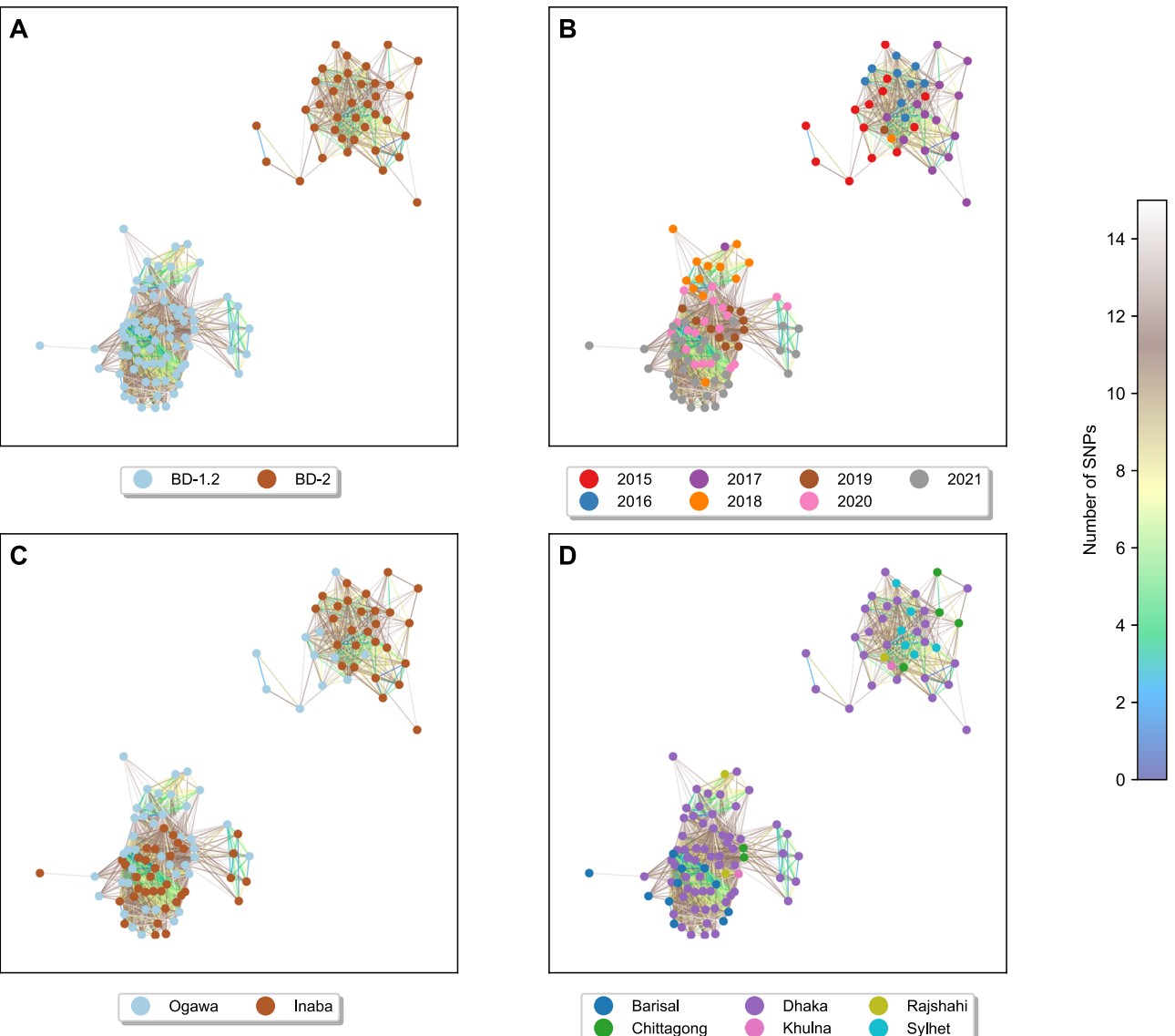

**Fig. 2 | SNP network analysis of highly connected isolates.** Network diagram showing pairwise connections between isolates in our cohort with less than 15 pairwise SNP differences. The panels show the same network with the nodes colour-coded according to (**A**) lineages, (**B**) year of collection, (**C**) serotypes and (**D**) location of collection. The lines between pairs of isolates are colour-coded by the number of SNPs.

*OmpU* is another gene with a statistically significant mutation (G325D) underlying lineages' separation. Amino acid D is predominant in BD-1.2, while the amino-acid G is prevalent in BD-2. To assess for any additional genes separating the BD-1.2 and BD-2 lineages we also conducted an analysis on the pangenomes of the lineages separately but found the results broadly in line with that of the combined pan-genome analysis presented above (Supplementary Note 4 and Supplementary Data 6-10).

To understand the systemic relationships connecting the identified lineage-specific genetic signatures on a mechanistic level, we analysed the 30 core genes in Table S1 with allelic variants that were found exclusively in one lineage but absent in the other lineage using the *V. cholerae* GSM model iAM-Vc960 (Fig. 3). Thirteen of these genes (*murI, ftsI, appC, suhB, glmM, dsbD, licH, cysG_1, cobB, clcA, argG, mak, phhA*) are metabolic and have been identified as playing integral roles in amino acid metabolism, cell wall metabolism, carbon metabolism, amino sugar and nucleotide sugar metabolism and energy metabolism (see Supplementary Data 11). Moreover, for these genes we sought to better understand their role by examining their effects on *V. cholerae*

growth rate, biochemical networks and production of metabolites in the networks. As the effect of mutations/gene knockouts cannot always be observed as change in growth rate (due to the redundancy of the reactions in metabolic networks of bacteria), it can be useful to also consider the changes in metabolite yield. Changes in metabolite yield have been found to correlate with changes in the virulence, persistence, and fitness of some organisms[24]. Furthermore, *V. cholerae* are capable of adapting to ecological niches by altering the metabolites they excrete to create a more favourable environment for *V. cholerae* and/or a less favourable environment for other species competing for the same resources[25,26]. Mutations disrupting larger numbers of metabolite yields may be suggestive of a larger systems-level impact on bacterial metabolic function. Therefore, gene essentiality, flux variability analysis (FVA) and flux balance analysis (FBA) were used to predict, through gene knockouts, the essentiality and the effects of the identified genetic determinants on the growth rates of *V. cholerae*, and also used to further explore their influence on metabolite yield. The latter was done by assessing the influence on metabolite flow within the complete metabolic

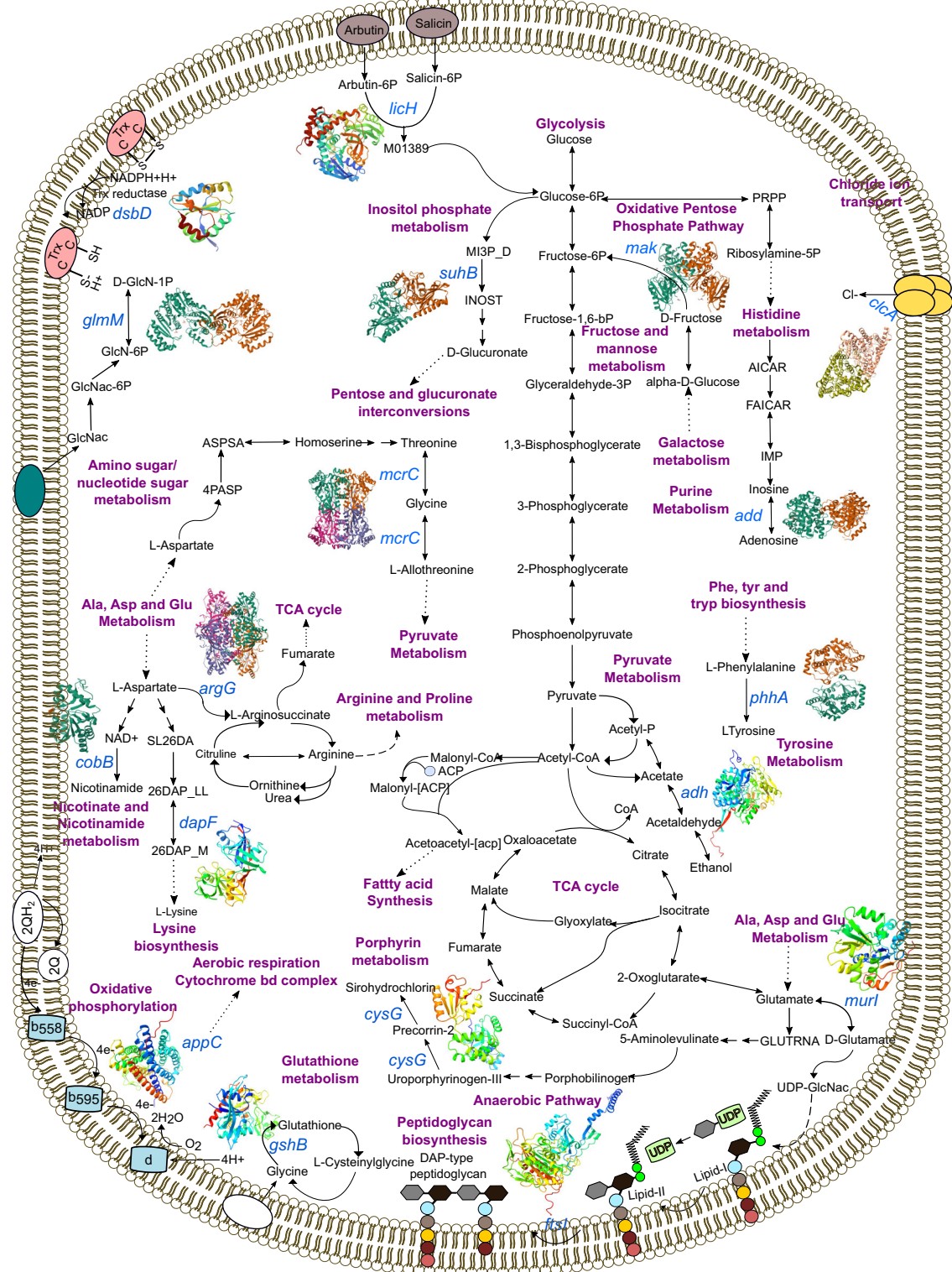

**Fig. 3 | An overview of the metabolic pathways associated to the core genes underlying the BD-1.2 and BD-2 lineages separation.** All genes annotated were found to have reduced flux span through the metabolic system when knocked out. Genes coloured in blue have a significant different allelic distribution between BD-1.2 and BD-2, associated metabolic pathways are labelled in purple. All 3D protein structures were generated in Alphafold[123] under a Creative Commons Attribution 4.0 license (CC-BY 4.0), no changes were made.

network of *V. cholerae*, encompassing all known metabolites and metabolic reactions (see Methods). In this analysis it was important to consider all reactions and metabolites in the model rather than focussing on a subset, as doing so ensures no undue bias or assumptions underlie the results.

The genes *cysG, clcA, adh* and *mcrC*, were found to be essential for growth (i.e., knocking these genes out reduced the biomass growth to less than 0.0001 h⁻¹) in both rich and minimal media. Furthermore, *murI, glmM,* and *dapF* displayed auxotrophic behaviour in minimal media, whereas *cysG, clcA, adh,* and *mcrC* were found to be essential in

rich media with alternative carbon sources. Additionally, three genes, *murI*, *glmM* and *dapF*, were found to be essential for growth in minimal media only. Next, FVA was used to identify biochemical reactions whose flux span was significantly changed (greater than 10% change) by knocking out these genes. In total ten genes (*murI*, *glmM*, *cysG*, *clcA*, *argG*, *mak*, *adh*, *dapF*, *add*, and *mcrC*) when knocked out significantly changed the flux span in at least one reaction through the model by FVA analysis, Supplementary Data 11. Finally, FBA analysis was used to determine the effect of gene knockouts on metabolite yield. Five genes, *murI*, *glmM*, *cycG*, *mak*, and *dapF* were found to reduce at least one metabolite yield to zero in the model when knocked out (given the wildtype yield was greater than 0), Supplementary Data 11[27,28]. Interestingly, the average number of metabolite yields affected by knock-outs of the genes discriminating lineages was significantly higher than a random selection of 100 metabolic genes (*p*-value 0.0429, Mann Whitney U test, two-sided), indicating a stronger influence on metabolite production for this subset of genes.

To further elucidate the metabolic differences between the BD-1.2 and BD-2 lineages, we repeated our previous analyses done on the generalised model using strain-specific models automatically generated by CarveMe[27]. Gene essentiality analysis concurred with the general model (iAM-Vc960), with only a small number of differences (Supplementary Data 12 and Supplementary Note 5). The effect of *murI* gene knockouts differed between lineages, proving non-essential in 94% of BD-1.2 lineage models but only in 76% of BD-2 lineage models. Flux variability analysis of the individual models revealed that *clcA* knockouts led to significant changes in the flux span of the CLt3_2pp reaction, which controls chloride transport, in 96% BD-2 models compared to just 5% of BD-1.2 models. The *clcA* gene has been linked to bacterial acid resistance and it has been suggested that changes to the expression/repression of this gene may help facilitate survival during movement through the intestinal tract[28]. Similarly, flux balance analysis indicated that metabolite yield was changed differently across lineages in response to knocking out *clcA*, with the metabolite yield of chloride reduced to 0 in 95% of BD-1.2 isolates.

In summary, a total of 15 genes found to underlie the genetic and temporal differentiation of *V. cholerae* BD-1.2 and BD-2 lineages, were also found to significantly alter the growth, reaction flux, or metabolite yield of *V. cholerae* when knocked down, either in the generalised iAM-Vc960 GSM model or in the draft strain-specific models. Of interest was the gene *clcA*, which showed differences in both flux span and metabolite changes between lineages in the draft GSM models. The FVA and FBA results indicate that these genes play important metabolic roles. Disruption of these functions could potentially affect bacterial growth or metabolic output, which may contribute to the survival and dominance of one lineage over another. Although our analysis cannot pinpoint a single SNP as responsible for the loss of metabolic function, it suggests that an accumulation of SNPs or gene losses could collectively lead to metabolic changes. We observe the potential for metabolic alterations driven by multiple mutations (SNPs).

Lastly, when mapping the 160 intergenic SNPs back to genomes, we found their location in the upstream/downstream regions of 35 known genes and 34 hypotheticals genes (see Supplementary Data 4). These intergenic SNPs exhibited allelic distribution, with the minor variant prevalent in the BD-2 isolates (68% to 100%), while the major variant dominated in the BD-1.2 isolates (over 98%), only one SNP in BD-1.2 had a major allelic variant at of 47% (Fisher exact test, Bonferroni correction *p*-value < 2.31e-08). Many of these SNPs were located within transcriptional factor binding sites (TFBs) (Supplementary Data 4). Intergenic SNPs, exhibiting significantly different allelic distributions between BD-1.2 and BD-2, mapped across the TFBs of 11 TFs (*ToxT, Fur, AmpR, OmpR, LuxR, LexA, ArgR, PhoP, CRP, ArcA, IHF*) (Fig. S6-S16). More information about the function of these transcriptional factor binding motifs is provided in Supplementary Note 6.

## Machine learning unravels correlations between genomic determinants and clinical symptoms in humans

Beyond identifying the potential involvement of new genetic traits in differentiating the BD-1.2 and BD-2 lineages, we hypothesised that the same or additional genetic features might play a significant role in the manifestation and severity of clinical symptoms in patients when infected with *V. cholerae*. A summary of the distribution of each clinical symptom over the two lineages is given in Fig. S17. We focused on the lineage BD-1.2, which caused the most recent outbreak in Bangladesh. To identify if and which coding and non-coding mutations and/or presence/absence of accessory genes would correlate with the different clinical symptoms, we employed a bespoke, supervised machine learning pipeline.

The pipeline is aimed at mining sequencing data to identify the genetic elements that more strongly correlate with observed clinical symptoms, which in this case are vomiting, dehydration, number of stools, duration of diarrhoea and abdominal pain (see Methods section). The pipeline is a bespoke adaptation of ML-based data-mining methods previously developed within our team to identify correlations between genomic features with phenotypes[17,18,29,30]. In the pipeline, information about different genetic features (SNPs -both from coding and non-coding regions- and presence/absence of accessory genes) can be encoded as input to ML-powered predictive models designed to estimate the likelihood of observing the selected phenotypes under each specific pattern of input values[17]. As long as trained with sufficient observational data, the ML-powered predictive models are able to replicate experimental evidence, in addition to providing information on what inputs correlated most strongly with each phenotypic manifestation. Through such introspective power, the pipeline is able to unravel co-occurrent, multiple mechanisms (mutations, horizontal gene transfer - HGT), variants in their functional protein-coding and regulatory forms, as well as their additive effect on the targeted phenotypes, which in this work, were clinical symptoms.

The following clinical symptoms were selected, namely: vomiting, abdominal pain, diarrhoea duration, 24-hour stool count and dehydration. Each clinical symptom was handled by building a dedicated symptom prediction model, operating using genetic elements as inputs. Two symptoms (vomiting and abdominal pain) were encoded as binary (presence vs absence). The other three symptoms—diarrhoea duration, 24-h stool count, and dehydration—were encoded as multi-class: dehydration as None, Moderate and Severe; diarrhoea duration as <1 day, 1–3 days, 4–6 days, and 7–9 days; and stool count in 24 h as 3–5 times, 6–10 times, 11–15 times, 16–20 times, and 21+ times. We handled the prediction of multi-class symptoms via the implementation of binary predictors.

The symptom prediction models were developed with built-in robustness to potential confounding factors. Specifically, the following list of variables was initially considered as potentially having confounding effects: year of collection, location of patient, sex of patient, age range of patient and serology of *V. cholerae*. Each potential confounder was tested for correlation to the symptom being targeted by the prediction model. If the potential confounder was found correlated to the symptoms (hence moving from potential to proven confounder), then any other input variable also found correlated with the same confounder would be eliminated from the prediction model. All the correlation tests between inputs and symptoms, as well as between inputs themselves, were run using two-sided chi-square tests. Further, possible confounding effects related to random initialisation parameters of SMOTE (see Methods) were contained by running SMOTE multiple times.

The development and optimisation of each symptom prediction model powered by machine learning was based on running a comparative analysis of the predictive performances of different machine learning algorithms, namely: linear support vector machine (linear SVM), non-linear SVM with radial basis function (RBF SVM), random

forest, extra-tree classifier and logistic regression) and two meta-methods (Adaboost and XGBoost). For each algorithm, multiple configurations of the hyperparameters of the learning algorithms were tested. A nested cross validation approach was used to select the best hyperparameters, based on randomly selecting different training and test sets, and using stratified k-fold cross validation metric. Finally, Friedman and Nemenyi tests were used to statistically compare and select the best performing algorithm for each prediction model (see Methods section).

In the end, based on a two-sided chi-square test of independence (p-value < 0.01), the models for abdominal pain, vomiting, number of stools 11–15 times vs. 21+ times, number of stools 11–15 times vs. 16–20 times, dehydration moderate vs severe were found immune to confounding effects due to year of collection, location of patient, sex of patient, age range of patient and serology of *V. cholerae*. The prediction model: diarrhoea duration <1 day vs 1–3 days was found immune to confounding effects due to age range of patient, sex of patient, location of patient, and serology of *V. cholerae*. However, the prediction model was found to be influenced by year of collection; therefore, the inputs that were also correlated to year of collection were removed from the analysis (Supplementary Data 13). Moreover, we were able to successfully develop six binary symptom prediction models featuring adequate prediction performance levels. These were dedicated to predicting the following binary phenotypical outcomes: (i) stools 11–15 times vs. 16–20 times; (ii) stools 11–15 times vs. 21+ times; (iii) moderate vs. severe dehydration; (iv) diarrhoea duration <1 day vs. 1–3 days; (v) presence vs absence of vomit; and vi) presence vs absence of abdominal pain (Supplementary Data 14). The remaining binary predictors were discarded for not performing adequately, either because of unbalanced available sets of observations (needed for training the supervised ML models), or because of more challenging separability of the phenotypes given the selected inputs (no features were statistically significant based on the Fisher exact test). Among the tested pipeline technologies mentioned earlier, logistic regression was identified by the Friedman F-test and the Nemenyi post-hoc analysis as the best performing one (Fig. S18). Of the six binary prediction models, four had an AUC greater than 0.9, Fig. 4. Supplementary Data 15 indicates the performance metrics obtained by all binary predictors for each clinical symptom. Figs. 4 and S19 show the performance results for the logistic regression classifier.

Analysis of the best-performing symptom prediction models allowed us to identify the input features (core genome coding and intergenic SNPs and accessory genes) most strongly correlated to each phenotype (Supplementary Data 16). Seventy-nine different features in total were selected as significantly correlated to at least one of the six symptom prediction models, with 68% being selected in two or more models (Fig. 5). No features were selected for all symptoms. All features associated with number of stools 11–15 times vs. 21+ times were found associated to at least one of the other five symptom prediction models. Forty-five accessory genes (nine known genes, *tufB_2*, *blc*, *pckA*, *luxR_2*, *hcpA_1*, *rpoS*, *dcuA*, *hpt*, *luxR*, and 36 hypothetical genes) and 28 core SNPs over 23 genes (14 known, *clpS*, *gshB*, *dapF*, *fabV_1*, *add*, *tufB*, *lpoA*, *phrB*, *yjcS*, *fabH1*, *cysG_2*, *padC*, *pepN*, *tadA_2*, and nine hypothetical genes) were identified as strongly associated to at least one of the symptoms. From the nine known accessory genes: four (*rpoS*, *hpt*, *luxR* and *pckA*) were found in the vomit model; *dcuA* was found in the abdominal pain model; *hcpA_1* was found only in the number of stools 11–15 times vs. 16–20 times; *luxR_2* was found in two models (vomit and dehydration moderate vs severe); *blc* and *tufB_2* were found in three models (vomit, number of stools 11–15 times vs. 16–20 times and number of stools 11-15 times vs. 21+ times) with *tufB_2* also found in abdominal pain and diarrhoea duration <1 day vs. 1–3 days models. Six SNPs from the genes *tufB*, *dapF*, *clpS*, *gshB* and *fabV* were associated to three symptom prediction models (vomit, number of stools 11–15 times vs. 16–20 times and number of stools

11–15 times vs. 21+ times) with the SNPs from the genes *dapF* and *fabV* also associated with abdominal pain and diarrhoea duration <1 day vs. 1–3 days and the SNP from the gene *tufB* associated with dehydration moderate vs severe.

Among the 45 accessory genes linked to clinical symptoms, six hypothetical genes were also statistically significant in distinguishing the two lineages. Among the other accessory genes selected, four (*blc*, *pckA*, *luxR* and *rpoS*) have important biological functions. In particular, *Blc*, also known as *VlpA*, is a lipocalin, that is correlated to acquisition of drug resistance in *V. cholerae*[31]. *PckA* (phosphoenolpyruvate carboxykinase) is important for gluconeogenesis, a highly conserved pathway in bacteria and humans. Interfering with the gluconeogenesis pathway impacts *V. cholerae* colonisation in mouse models, highlighting its crucial role in sustaining *V. cholerae* growth and viability within the intestines[32]. *LuxR* plays a key role in regulating biofilm production and secretion in *V. cholerae*[33]. *RpoS* is a sigma factor that facilitates physiological adaptation to general starvation and stationary phase growth in different species. *V. cholerae* strains lacking the gene *rpoS* are impaired in their ability to survive in different environmental stresses. *RpoS* was also shown to be important in *V. cholerae* for efficient intestinal colonisation[34].

Out of the 28 core SNPs associated to the clinical symptoms, 11 were also found previously as statistically significant in differentiating the BD-2 and BD-1.2 lineages (see above), Supplementary Data 16. These 11 SNPs mapped to 11 genes (*clpS*, *gshB*, *dapF*, *fabV_1*, *add*, and six hypothetical genes). Among the SNPs mapping to known genes (*clpS*, *gshB*, *dapF*, *fabV_1*, *add*), three are non-synonymous SNPs mapping to *clpS*, *gshB* and *fabV*. In *V. cholerae clpS* regulation involves cAMP receptor protein (CRP)[31]. CRP is important in intestinal colonisation[35]. *GshB*, encodes a glutathione synthetase (GSH), which is associated to resistance to oxidative stress. *V. cholerae fabV* is one of the several triclosan-resistant ENR encoding genes[36].

As in our previous lineage analysis, we sought to better understand the importance of the genes which had been found to better correlate with the severity of the symptoms. We examined for those genes that were metabolic, through FVA and FBA, the effects of such genes on growth rate (gene essentiality), and beyond that, their influence on metabolite yield and reaction flux. Nine symptoms-related genes were identified as metabolic genes in the iAM-Vc960 GSM model (Fig. 6). Eight of these genes were associated to five metabolic systems (Supplementary Data 17). *FabH1* and *gshB* associated with cofactor and prosthetic group metabolism; *pckA* is associated with carbohydrate metabolism; *dcuA* plays a crucial role in C4-dicarboxylate transport; *dapF*, *pepN* and *gshB* are significant in amino acid metabolism; *add* and *pckA* are relevant to nucleotide metabolism; *oppA* and *fabH1* are involved in cell wall metabolism, with *fabH1* relevant for fatty acid biosynthesis (Supplementary Data 17).

Using FBA and FVA analysis, the knockouts of the genes *dapF* and *gshB* were found to halt production of several metabolites. The genes *pckA*, *add*, *dapF*, *oppA*, *gshB* were found to significantly change the reaction flux span, Supplementary Data 17. Both FBA and FVA analysis can infer if potential metabolic adaptation mechanisms for *V. cholerae* can lead to alterations in bacterial virulence, potentially leading to worse symptoms, if genes significantly affect pathways which are associated with important functions such as colonisation, biofilm production and cell wall synthesis. For example, the *gshB* gene, a glutathione reductase, contributes to *V. cholerae* intestinal colonisation[37] and has a role in acid tolerance response[38]. Similarly, *dapF* was found as an essential gene in minimal media and leading to auxotrophic behaviour to the amino-acid lysine. As Pearcy et al.[39] indicated, an auxotrophic behaviour of a gene connected to amino-acid biosynthesis is important because it can provide competitive fitness advantage against commensal bacteria. During the infection stage *V. cholerae* engage and compete with commensal bacteria for nutrient acquisition to support rapid growth and multiplication[40]. Moreover,

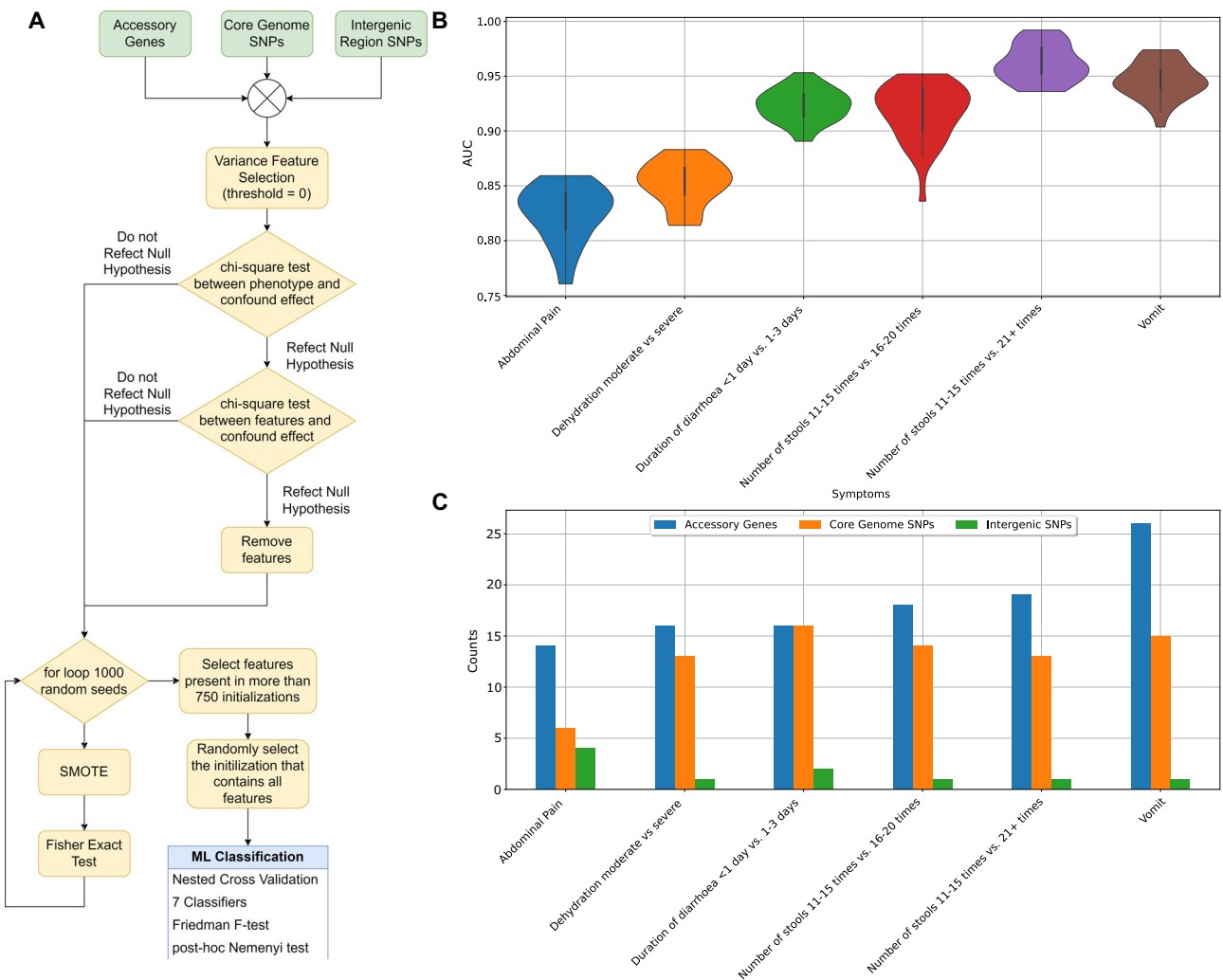

**Fig. 4 | Supervised machine learning pipeline accurately predicts the clinical manifestations of hospitalised patients from the genomic determinants extracted from BD-1.2 isolates, collected among the same hospitalised patients. A** Flow diagram showing machine learning pipeline including data (green), pre-processing steps (yellow) and classification (blue). **B** Machine learning performance results measured by the area under the curve (AUC) from 30 training runs for clinical symptom combination. The results shown are for the best classifier Logistic Regression, as defined by the Nemenyi test (Fig. S18). The violin plots show the distribution of the data, with each data point representing one classification model. Inside each violin plot is a box plot, with the box showing the interquartile range (IQR), the whiskers showing the rest of the distribution as a proportion of 1.5 x IQR and the white circle representing the median value. **C** Number of features (accessory genes, core genome and intergenic SNPs) selected for each symptom. Predictive models were generated for six different clinical symptoms (X-axis): abdominal pain; dehydration Moderate vs. Severe; duration of diarrhoea <1 day vs. 1–3 days; number of stools 11–15 times vs. 16–20 times; number of stools 11–15 times vs. 21+ times; and vomit.

the lysine pathway plays a central role in eubacteria cell wall bio-synthesis, since meso-diaminopimelate is the immediate precursor for the biosynthesis of its main component, peptidoglycan, with *dapF* responsible for the synthesis of meso-diaminopimelate in the lysine pathway[41,42]. The proper synthesis and maintenance of peptidoglycan is essential for bacterial virulence and its viability[43].

To further investigate the link between metabolic gene variations and the clinical symptoms observed in different strains, we utilised draft strain-specific models generated with CarveMe[27]. The gene essentiality analysis results were largely consistent with those of the general model (iAM-Vc960), with only a few differences noted (Supplementary Data 18). The effect of *dapF* gene knockouts varied between models with the gene being essential in 93% (n = 20) and non-essential in 7% (n = 9) of the models. Comparing symptoms between the 'essential' and 'non-essential' groups, dehydration was significantly more severe in the 'non-essential' group (Fisher exact test p-value = 0.05). All strains in this group exhibited severe dehydration, suggesting a link between non-essentiality of the *dapF* gene and the

severity of *V. cholerae* symptoms. In relation to this, the flux balance analysis revealed changes in metabolite yields associated with the genes *dapF* and *cysG_2* across all strain-specific models. For *dapF*, altered metabolite yields were predominantly observed in strains where *dapF* was essential, while knocking out *dapF* in non-essential models had minimal impact on the metabolite yields of murein-related metabolites. This indicates metabolic adaptations linked to bacterial survival in these strains, potentially contributing to more severe disease outcomes. Additionally, knocking out the *padC* gene resulted in significant changes in metabolite yields only in the NGICDV-066 strain. Although conclusions drawn from a single strain are limited, it is notable that this isolate exhibited the most severe clinical symptoms across all measured symptoms, except for the duration of diarrhoea (presence of vomiting, presence of abdominal pain, number of stools (21+ times), presence of severe dehydration, duration of diarrhoea 1-3 days). Flux variability analysis in individual models indicated consistent behaviour across all strain-specific models regarding gene knockouts associated with clinical symptoms. Specifically, five gene

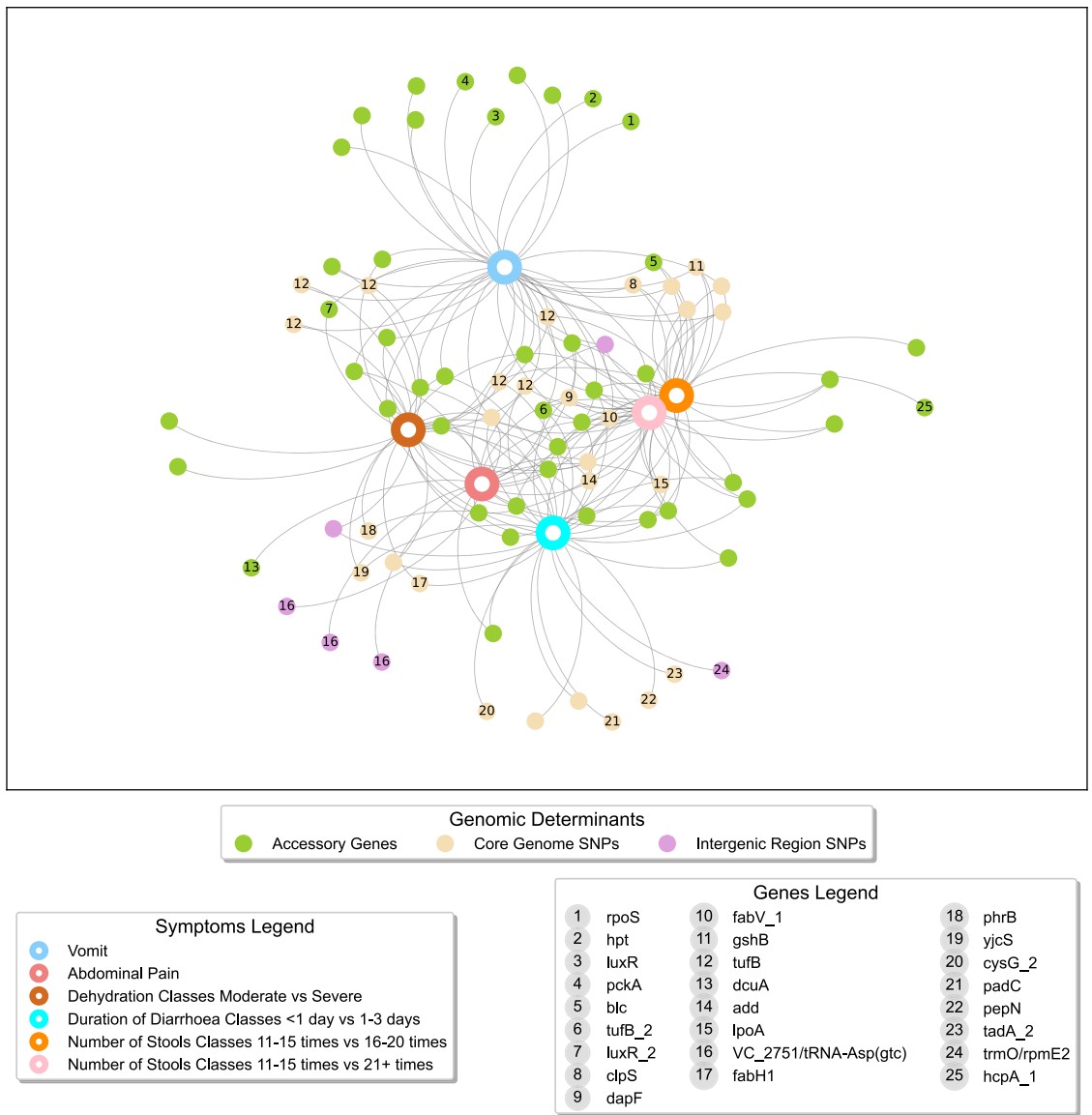

**Genomic Determinants**

- ● Accessory Genes
- ● Core Genome SNPs
- ● Intergenic Region SNPs

**Symptoms Legend**

- ○ Vomit
- ○ Abdominal Pain
- ○ Dehydration Classes Moderate vs Severe
- ○ Duration of Diarrhoea Classes <1 day vs 1-3 days
- ○ Number of Stools Classes 11-15 times vs 16-20 times
- ○ Number of Stools Classes 11-15 times vs 21+ times

**Genes Legend**

| | | | | | |
|---|---|---|---|---|---|
| 1 | rpoS | 10 | fabV_1 | 18 | phrB |
| 2 | hpt | 11 | gshB | 19 | yjcS |
| 3 | luxR | 12 | tufB | 20 | cysG_2 |
| 4 | pckA | 13 | dcuA | 21 | padC |
| 5 | blc | 14 | add | 22 | pepN |
| 6 | tufB_2 | 15 | lpoA | 23 | tadA_2 |
| 7 | luxR_2 | 16 | VC_2751/tRNA-Asp(gtc) | 24 | trmO/rpmE2 |
| 8 | clpS | 17 | fabH1 | 25 | hcpA_1 |
| 9 | dapF | | | | |

**Fig. 5 | Undirected graph network illustrating the genomic features associated with clinical symptom models for *V. cholerae*.** Node colour denotes the genomic determinant category, (i.e. accessory genes and/or core genome coding, and intergenic SNPs) identified by machine learning. Nodes are labelled with numbers corresponding to specific genes associated with each genomic determinant, as detailed in the Genes Legend, while unnumbered nodes are related to unannotated (hypothetical) genes. The clinical symptom models are highlighted in different colours and explained in the legend Symptoms Legend featuring abdominal pain; dehydration Moderate vs. Severe; duration of diarrhoea <1 day vs. 1–3 days; number of stools 11–15 times vs. 16–20 times; number of stools 11–15 times vs. 21+ times; and vomiting.

knockouts (*add, dapF, gshB, padC, pckA*) showed significant flux span changes in all models.

In summary, in relation to gene essentiality, reaction flux and metabolite yield, our results show that *gshB* and *dapF* make interesting candidates for further analysis, as knockout models of these genes predict significant changes to the bacterial metabolic function.

To delve deeper into understanding the functional mechanisms underlying clinical symptoms, we explored the interactome of the proteins associated to the clinical symptoms. The protein-protein interaction network (PPI) analysis revealed the interactome of 36 proteins, selected by the machine learning pipeline, with 109 other proteins, Fig. S20. The KEGG analysis indicated enrichment in ribosome proteins (e.g., RpoS) and fatty acid biosynthesis (e.g., FabH1, FabV) (Fig. S21). The colonisation in the human intestine and virulence of *V. cholerae* is intricately connected to both fatty acid metabolism[44] and the ribosome pathway[45]. The gene onthology (GO) analysis

highlighted enrichment in translation, peptide biosynthetic processes, and gene expression, featuring TufA, TufB, RpoS, GshB (Supplementary Data 19 and 20). The peptide biosynthetic pathway plays a vital role in *V. cholerae* biofilm formation and colonisation[23].

None of the six intergenic SNPs selected by the machine learning pipeline were in TFBs or promoters. These SNPs were located in a region without any functional annotations within 2 kbps upstream or 0.5 kbps downstream of a gene, adhering to the standard database dbSNP cutoffs for SNP-to-gene mapping[46,47]. See Supplementary Data 16 for additional information about the location of these SNPs.

**Structural analysis suggests evolutionary drivers of selection, mechanistic bases for BD-2 and BD-1.2 lineages evolution, and associations to clinical symptoms**

To further understand whether the identified alleles play a causal role in the evolution of lineages and clinical symptoms, we selected two of

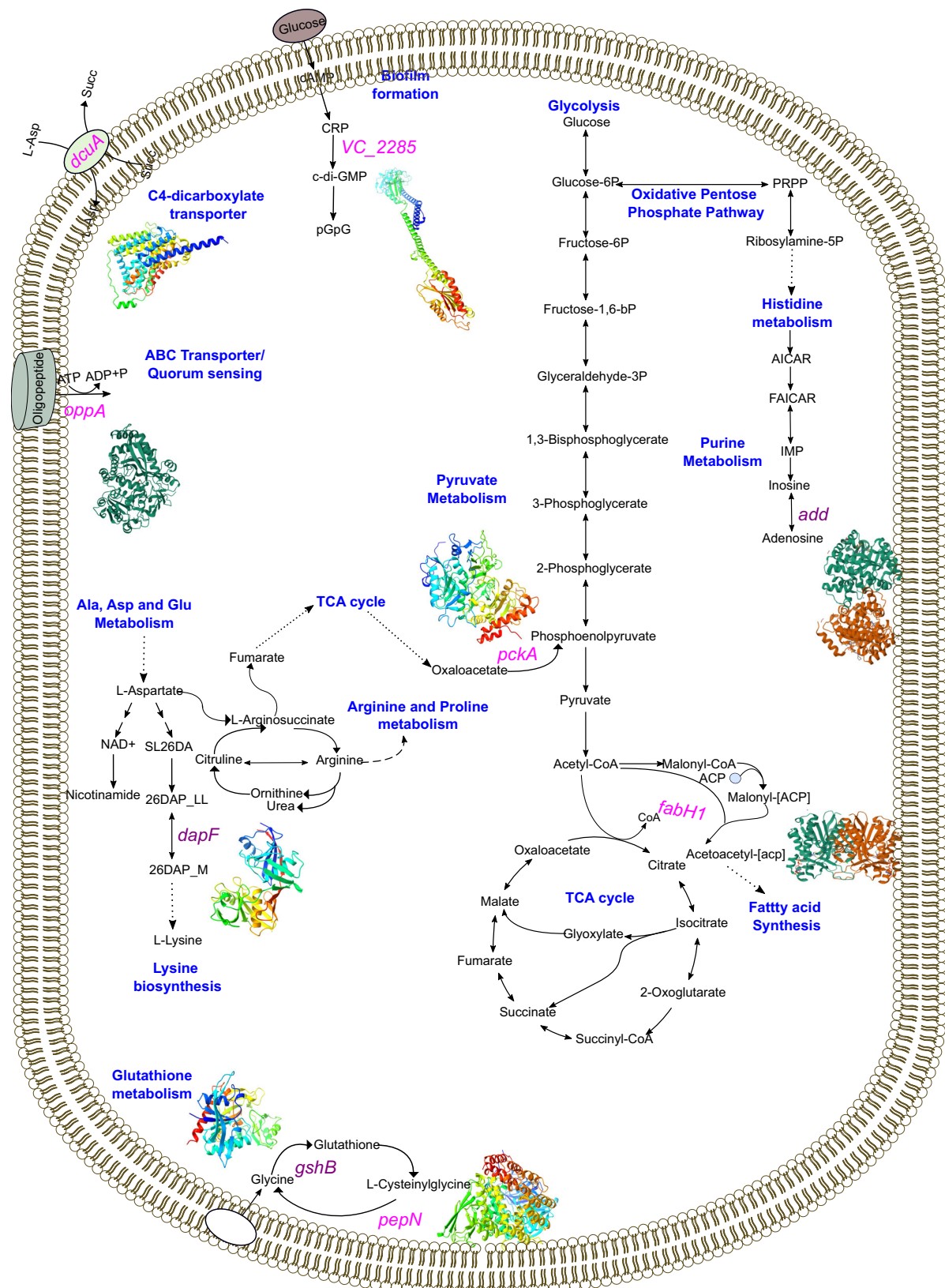

**Fig. 6 | An overview of the metabolic pathways impacted by statistically significant genes underlying clinical symptoms.** All genes annotated were found to have reduced the flux span through the metabolic system when knocked out. Genes coloured in pink and purple carried mutations or are accessory genes associated to the clinical symptom, respectively, and connected metabolic pathways (labelled in blue). The genes coloured in purple were also found as statistically significant in differentiating the BD-2 and BD-1.2 lineages (see previous sections). All 3D protein structures were generated in Alphafold[123] under a Creative Commons Attribution 4.0 license (CC-BY 4.0), no changes were made.

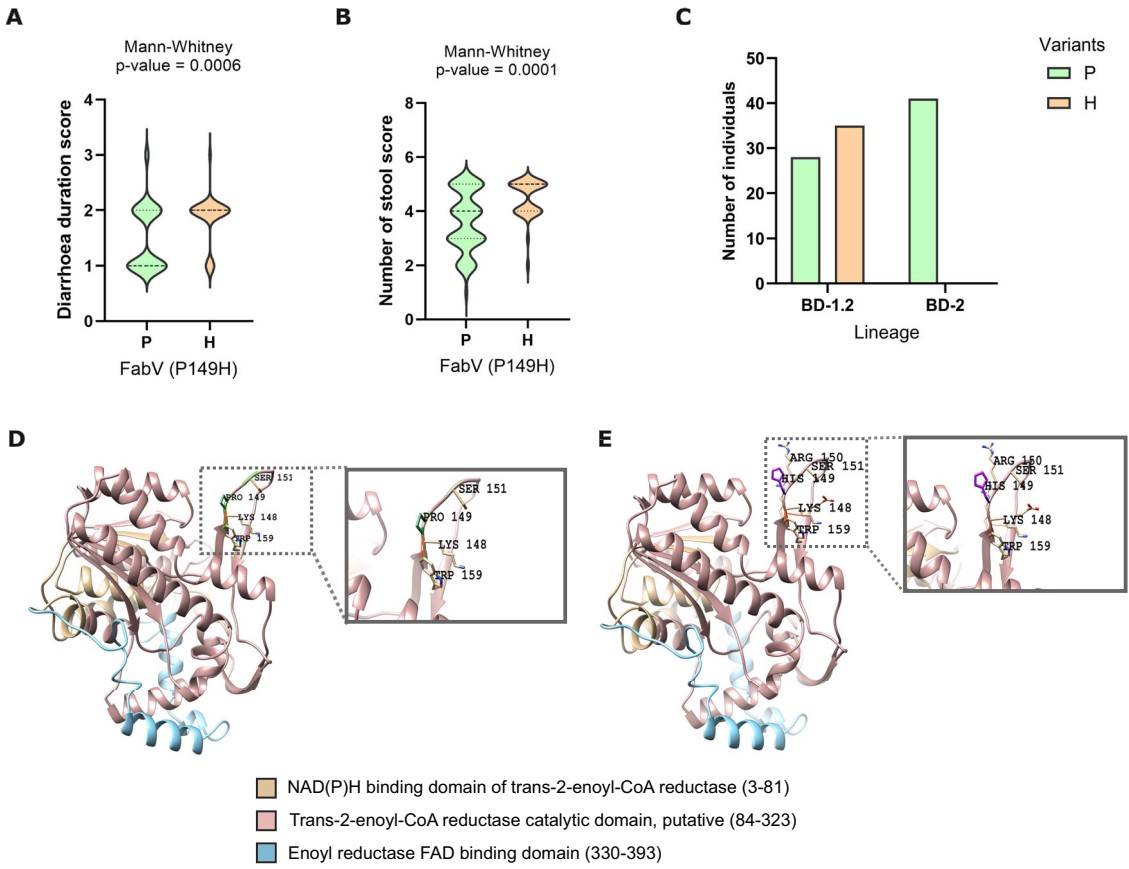

**Fig. 7 | 3D protein structure analysis of FabV allelic variants underlying BD-1.2 and BD-2 lineage evolution and clinical symptoms. A** Violin plot indicating the distribution of the diarrhoea duration score (0: no diarrhoea, 1: <1day, 2: 1–3 days, 3: 4–6 days and 4: 7–9 days) for the isolates containing either Pro149 (P) or His149 (H). Statistical significance was tested with a two-sided Mann Whitney U test, p-value is shown. **B** Violin plot indicating the distribution of the number of stools score (0: <3 times; 1: 3–5 times; 2: 6–10 times; 3: 11–15 times; 4: 16–20 times; 5: 21+ times) for the isolates containing either Pro149 (P) or His149 (H). Statistical significance was tested with a two-sided Mann Whitney U test, p-value is shown. **C** The bar graph displays the number of isolates in the two BD lineages associated with Pro149 (P) and His149 (H). **D** 3D structures of FabV (AlphaFold) with Pro149 and coloured by functional domains. Amino acid residues (Lys148, Ser151, and Trp159) interacting with Pro149 (green) are shown in sticks models. **E** 3D structures of FabV (AlphaFold) with His149 and coloured by functional domains. Amino acid residues (Lys148, Arg 150, Ser151, and Trp159) interacting with His149 (purple) are shown in sticks models.

the top-ranked non-synonymous SNP candidates, prioritising the following aspects in relation to the associated genes: (i) have significant difference of allelic distribution between BD1-1.2 and BD-2; (ii) have a significant correlation, as detected by the ML pipeline, with the selected clinical symptoms; (iii) are characterised as functionally important for *V. cholerae* metabolisms (i.e. significantly impacting reaction flux when knocked out, as highlighted by the GSM model) and/or interactome (i.e. enrichment of the functions and mechanisms related to pathogenesis); (iv) 3D structural mutation analysis could be benchmarked with experimental evidence. This resulted in three genes, all top-ranked by both the Fisher Exact test for BD-1.2 and BD-2 lineage evolution and the ML analysis for the underlying clinical symptoms, namely: *fabV*, *gshB* and *clpS*. We mapped the alleles of *fabV*, *gshB* and *clpS* to their protein structures using both experimental crystal structures and predicted homology models. However, the 3D-structure could be utilised to infer the mechanistic basis only for *fabV* and *gshB*.

In all BD-2 isolates FabV had a proline at position 149 (Pro149) whereas, in BD-1.2 isolates, the Pro149 was found in only 40.5% of cases, with the remaining 59.5% isolates exhibiting histidine at position 149 (His149). The BD-1.2 isolates with His149 showed a higher duration of diarrhoea (1–3 days) and a higher number of stool score (16-20 times and 21+ in 24 h) compared to the BD-1.2 isolates with Pro149, featuring a lower diarrhoea duration (<1 day) and lower number of stools score (11–15 times). The amino acid 149 was located

in the trans-2-enoyl-CoA reductase catalytic domain (Fig. 7A–E), when Pro149 is present, it interacts with Lys148, Ser151, Trp159 through Van der Waals (VDW) interactions, whereas His149 not only forms the aforementioned interactions but also creates an extra VDW interaction with Lys148. Furthermore, His149 interacts with an additional amino acid, Arg150, through a VDW interaction. These additional interactions in the presence of the His149 cause an increase in the stability of the structure ($\Delta\Delta G = 0.101$ kcal/mol $> 0$) and a decrease of the molecule flexibility ($\Delta\Delta S_{Vib}$ ENCoM: $-0.053$ kcal.mol$^{-1}$ K$^{-1}$), which is usually linked to a stronger binding affinity[48,49]. Moreover, the presence of His149 increased the positive charge of the surrounding area (Lys148, His149, Arg150) (Fig. S22), with an overall electrostatic energy increasing from 7.3E + 03 kJ/mol (Pro149) to 7.48E + 03 kJ/mol (His149) within the 5 Å region and with an overall protein total electrostatic energy rising from 2.1E + 05 kJ/mol (Pro149) to 2.52E + 05 kJ/mol (His149). Exposed, positively charged amino acids are suggested to promote interactions with negatively charged cellular systems[50]. The enhanced positive charge of FabV in the presence of His149 might support its role in participating in the breakdown of the negatively charged fatty acids.

*GshB*, a glutathione reductase, has been shown to contribute to *V. cholerae* intestinal colonisation[37] and to have a role in the ability of *V. cholerae* to mount an acid tolerance response[38]. In all BD-2 isolates GshB had a threonine at position 93 (Thr93), whereas in the BD-1.2, the Thr93 was only found in 21.5% of the cases, with most (78.5%) of the

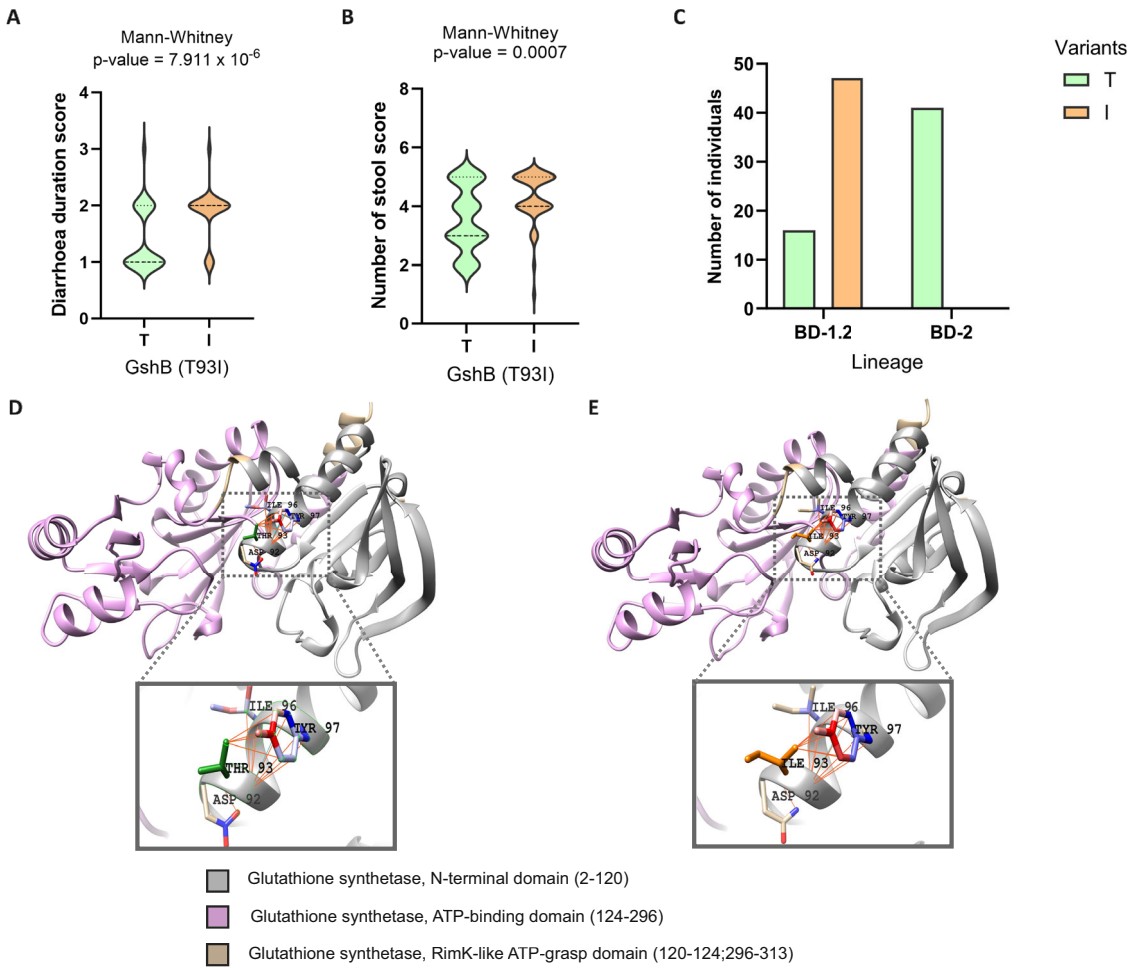

**Fig. 8 | 3D protein structure analysis of GshB allelic variants underlying BD-1.2 and BD-2 lineage evolution and clinical symptoms. A** Violin plot indicating the distribution of the diarrhoea duration score (0: no diarrhoea, 1: <1day, 2: 1–3 days, 3: 4–6 days and 4: 7–9 days) for the isolates containing either Thr93 (T) or Ile93 (I). Statistical significance was tested with a two-sided Mann Whitney U test, p-value is shown. **B** Violin plot indicating the distribution of the number of stools score (0: <3 times, 1: 3–5 times; 2: 6–10 times; 3: 11–15 times; 4: 16–20 times; 5: 21+ times) for the isolates containing either Thr93 (T) or Ile93 (I). Statistical significance was

tested with a two-sided Mann Whitney U test, p-value is shown. **C** The bar graph displays the number of isolates in the two BD lineages associated Thr93 (T) or Ile93 (I). **D** 3D structures of GshB (AlphaFold) with Thr93 and coloured by functional domains. Amino acid residues (Asp92, Ile96, and Tyr97) interacting with Thr93 (green) are shown in sticks models. **E** 3D structures of GshB (AlphaFold) with Ile93 and coloured by functional domains. Amino acid residues interacting with Ile93 (orange) are shown in sticks models.

BD-1.2 isolates exhibiting an isoleucine (Ile93) at this position. The BD-1.2 isolates with Ile93 are associated to a higher duration of diarrhoea (1-3 days) and a higher number of stool score (16-20 times and 21+ in 24 h) compared to the BD-1.2 isolates with Thr93. Thr93 interacts with Asp92, Ile96, Tyr97 through 13 VDW interactions and 1 H-bond; whereas Ile93 not only forms the aforementioned interactions but also creates extra VDW interactions with Tyr97 (Fig. 8A–E). These additional bonds in the presence of Ile93 cause an increase in the stability of the structure ($\Delta\Delta G = 0.384$ kcal/mol >0) and a decrease of the molecule flexibility ($\Delta\Delta S$Vib ENCoM: $-0.055$ kcal.mol-1.K-1), which is usually linked to a stronger binding affinity[48,49]. Moreover, the presence of Ile93 increased the negative charge of the surrounding area (<5 Å) (Fig. S23A, B), with an overall electrostatic energy decreasing from 7.93E + 03 kJ/mol (Thr93) to 7.4E + 03 kJ/mol (Ile93) within the 5 Å region and with an overall protein total electrostatic energy varying from 2.1E + 05 kJ/mol (Thr93) to 1.8E + 05 kJ/mol (Ile93). A decrease in total electrostatic energy is often associated to folding[51], protein folding stability is largely dependent on the hydrophobic interactions of nonpolar residues[52]. The surface, on average, has become more hydrophobic, indicating a possible reorientation of residues or a change in the surface's exposure to the solvent (Fig. S23C, D).

## Discussion

Bangladesh has witnessed the continual genomic evolution of *V. cholerae* lineages, with increased virulence, resistance, global spreading ability and disease severity. The potential of a *V. cholerae* isolate to have a global spreading ability and cause disease is mostly approached by studying its genomics via bioinformatics analysis. Two recent studies[9,10] explored the genomics attributes of the lineage BD-2 predominant between 2004 and 2018 and the emergent lineage BD-1.2 appearing from 2016 onwards and responsible for the 2022 outbreak[9,10]. By comparing these lineages, the authors revealed mutations in *ctxB* allele, SXT/ICE, VSP-II, VPI-1 and *gryA* allele[10] potentially explaining the recent shift in lineage predominance. Despite these knowledge advances, gaps persist in understanding the entire genomic repertoire associated to transmission ability and different disease severity patterns.

Here, we developed an analysis approach that combines, ML-powered data mining, whole-genome sequencing, genome-scale metabolic modelling and 3D structural analysis to uncover, on a finer scale, unknown associations between lineage transmission dynamics, disease severity and the genomic make-up of *V. cholerae* isolates. Machine learning offers a powerful opportunity to analyse entire

genomes efficiently against selected phenotypes (lineages, clinical symptoms), allowing for the identification of genomic features ranked on strength of correlation with the phenotype. This provides a significant advantage to conventional genomics-only methods based on checking for presence/absence or based on similarity searches of known manually chosen determinants. Moreover, our approach allowed various genetic determinants (accessory genes, and core coding and intergenic SNPs) to be analysed simultaneously to capture the co-occurrence, synergism and additive effect of multiple mechanisms and determinants (mutations, accessory genes, horizontal gene transfer, functional, metabolic, and regulatory variants). Determinants identified by ML may contain genes with a known functional relationship with the phenotype as well as genes with no previously known association with that specific phenotype. Altogether, our reference-agnostic approach overcomes limitations of previous genomics studies that only considered one feature type (SNPs, accessory genes) at a time and known genetic elements associated to *V. cholerae* transmission.

Using our method, in addition to confirming the aforementioned mutations identified in recent genomics studies[10], we found further mutations in VSP, VPI, and PLE, exclusive to one lineage and absent in the other, supplementing those previously found by Monir et al.[10]. Moreover, our findings expand known mutations to a wider range of genomic determinants, including 115 accessory genes, 1225 core coding SNPs, and 160 intergenic SNPs crucial for explaining at a more-in depth scale BD-1.2 and BD-2 recent shift. Supplementing the previous knowledge on the type, number and functions of genomics determinants differentiating BD-1.2 and BD-2[10].

For example, five core genes (*skp*, *tamA*, *clcA*, *cysG*, and *valS*) with a unique non-synonymous variant in BD-1.2 and playing key roles on toxin transport and acid tolerance, shed new light on functions and may help clarify their contribution to the recent prevalence of BD-1.2 over BD-2. In addition, non-synonymous SNPs, found uniquely in BD-1.2, were mapped to genes with functions such as colonisation, toxins export, virulence, growth, response to pH and temperature, and phage resistance. For example, the mutation G325D in *ompU* conferring bacteriophage resistance, was found in this work to be statistically important to differentiate the two lineages. OmpU a pore-forming protein of the outer membrane of *V. cholerae* has adhesive properties which may play a role in the pathogenesis of cholera[53], is critical for *Vibrio* fitness[54,55], for dissemination[54], for protection against the bactericidal effect of bile salts[56], cationic peptides[57] and intestinal organic acids[58]. The G325D mutation is located within the L8 loop, which has been reported to be crucial for neutralising infection and conferring resistance against phages[59,60]. Seed et al.[60], showed that in presence of the bacteriophage ICP2 (bacteriophage that preys on *V. cholerae* and was first isolated from cholera patient stool samples[61]) the OmpU virulent mutant (G325D) had a 10,000-fold enrichment over the wild-type, indicating that strong selective pressure is imposed by phage predation during *V. cholerae* infection.

Out of the twelve accessory genes found statistically significant to differentiate the two lineages, five (*lon_3*, *endA*, *adh*, *hdfR_4* and *bcr_2*) were present uniquely in BD-1.2 with functions such as antibiotic resistance and biofilm formation. Increasing evidence indicates that *V. cholerae* has the capability to develop biofilm-like aggregates during infection, potentially serving as a function in pathogenesis and disease transmission. Nonetheless, the composition, control mechanisms governing the formation of these biofilms during infection, and their significance in intestinal colonisation and virulence remain yet to be elucidated[62].

In addition to the coding genome, we found that regulatory networks are associated to lineage differentiation. Among the most relevant intergenic SNPs exhibiting significant allelic distribution between the two lineages is the one mapping in the TFBs of *ToxT*. This TF plays a crucial role in the development of *V. cholerae*-related symptoms[60] and

selectively regulates the expression of virulence genes found in toxin-coregulated pilus (TCP) and cholera toxin (CT)[63,64]. Environmental conditions within the intestinal tract, such as the presence of bile, bicarbonate, reduced oxygen levels, and unsaturated fatty acids, play a significant role in promoting the simultaneous expression of genes responsible for the production of Tcp, CT, and various other genes linked to colonisation[12,63]. The activation of the *ToxT* regulon is also influenced by metabolic cues and quorum sensing[12,63]. Although, transcription factor binding site prediction algorithms tend to over-predict sites. The correlation of experimentally determined SNPs with the predicted sites and their different nucleotide frequency provides a reasonable certainty that the observation reflects the phenomenon. The fact that we found significant intergenic SNPs in TFBs of 11 TFs and not in promoters, suggests a possible important role in such scenario. Higher frequency of SNPs close to transcriptional start sites is related to subtle alteration of gene expression which might result in lineage diversity. In addition to a wider range of genomic determinants found in this study, we also found 23 genes with mapped SNPs (*tyrA*, *gyrA*, *ctxB*, *glmM*, *tamA*, *valS*, *czcA*, *licH*, *mutL*, *kbl*, *cobB*, *mak*, *znuC*, *phhA*, *nagA_1*, *argG*, *cysG_1*, *murI*, *appC*, *putA*, *suhB*, *fadJ* and *recD*) in common between our analysis and Monir's comparison of BD-1 vs BD-2[9] and nine genes with SNPs (*rstA*, *ubiA*, *dsbD*, *clcA*, *thiG*, *rtxA*, *mltD*, *fadJ* and *recD*) in common between our analysis and Monir's comparison of BD-1.1 vs BD-1.2[10].

Roughly 20% of people who contract toxigenic *V. cholerae* show cholera symptoms[12]. Among symptomatic cases, approximately 5% are mild, 35% are moderate, and about 60% are severe. The disease severity depends on pathogenic factors of the bacteria, and host factors including age, nutrition, and immune system[12]. Here, we revealed the existence of correlations between a core set of genetic determinants in *V. cholerae* and clinical symptoms (diarrhoeal duration, number of stools, abdominal pain, vomiting, and dehydration). A recent study[65] investigated these correlations, using machine learning, by analysing gene families in the gut microbiome of household members of cholera patients to predict disease severity. In such study, associations were found in gene families like ribosomal proteins, RNA polymerases, and the sugar phosphotransferase system with symptomatic disease. However, the computational pipeline adopted in such work[65] did not produce high-performance metrics for predictive models. Our pipeline, in contrast to Levade et al.[65], achieved superior performance metrics, and encompassed accessory genes, core genome SNPs, and intergenic SNPs. It considered variants in both functional protein-coding and regulatory forms, revealing their additive effect on diverse clinical symptoms.

Moreover, mechanistic insights were derived through GSMMs and protein-protein interaction networks. Notably, we identified genes crucial for pH homoeostasis, host adaptability, colonisation, virulence, motility, acid tolerance, toxin transport, biofilm formation, and bacteriophage resistance. Important pathways were found underlying these roles, such as the fatty acids biosynthesis which is important for *V. cholerae* since unsaturated fatty acids present in bile inhibit the expression of virulence factors and both cholesterol and unsaturated fatty acids can enhance the motility of *V. cholerae*[66]; and biofilm production which plays a crucial role in the cholera pathogenesis and dissemination of disease[62]. Furthermore, our ML analysis identified genes associated to abdominal pain that were also found important for colonisation in *V. cholerae*. It is known that colonisation of pathogenic bacteria can present clinical symptoms such as abdominal pain[67].

Three non-synonymous SNPs associated to the clinical symptoms were also found as statistically significant in differentiating the BD-1.2 and BD-2 lineages. These SNPs mapped to *clpS*, *gshB* and *fabV*. In *V. cholerae clpS* regulation involves cAMP receptor protein (CRP)[35]. CRP is important in *V. cholerae* gene regulatory network lifestyle switching, adapting gene expression for quorum sensing, intestinal colonisation, and toxin production to its environment[35]. *GshB*, encodes a

glutathione synthetase (GSH), associated to resistance to oxidative stress. It is part of the σ32 regulon, contributing to *V. cholerae* intestinal colonisation[37]. Glutathione controls the potassium efflux system, Kef, and pH homoeostasis involved in Na+ and K+ transport[68]. Impaired glutathione production may affect the stress response[68]. GshB was additionally shown to have a role in the ability of *V. cholerae* to mount an acid tolerance response[38]. *V. cholerae fabV* is one of the several triclosan-resistant ENR encoding genes[36]. Resistance to triclosan also affects resistance to other antibiotics, showing cross-resistance to a wide range of antibiotics (including chloramphenicol and tetracycline)[69]. Moreover, *fabV* exhibits pleiotropic effects controlling pathogenicity in *P. aeruginosa* via modulation of fatty acids synthesis, production of virulence factors and motility[70].

Analysing the 3D structure based on non-synonymous mutations can provide insights into the mechanisms by which these mutations can cause disease[71–74]. Changes in the stability of proteins can lead to manifestation of diseases[73] or symptom variations[71,74]. Among all types of mutations, non-synonymous SNPs have the greatest impact on protein structure and function[75]. In this work we found that different SNPs accumulated in BD-1.2 isolates compared to BD-2 isolates, suggesting different evolutionary dynamics possibly explaining the temporal shift of the two lineages. Our analysis of top-ranked non-synonymous SNPs in protein-coding regions, identified by machine learning as linked to both BD-1.2 lineage evolution and clinical symptoms, specifically FabV and GshB, unveiled that SNPs present in BD-1.2, associated with more severe cholera, led to increased protein stability. That protein stability might be relevant for disease severity is also supported by the fact that no SNPs associated to clinical symptoms were found in any TFBs or promoter signature but only in protein-coding sequences. In this study, we have identified promising targets related to metabolism (*clcA, cysG, adh*), antimicrobial resistance (i.e. *bcr, blc*), and virulence (i.e. *ompU, skp, tamA, valS*). These targets show significant potential for further investigation through experimental studies.

We are aware of the limitations of our current study. Several host factors (retinol deficiency, blood group, genetic factors, innate immune system) confer susceptibility to cholera with higher risk of symptomatic disease[76]. These factors have not been considered in this study due to lack of data. A further limitation of this study was the inability to consider the potential impact of co-infections with either multiple *V. cholerae* lineages/strains or other pathogens. Whilst the presence of more than one *V. cholerae* strain or lineage in a host has recently been shown to be unlikely[77–79], co-infections with other bacteria can occur in diarrhoeal patients. A study of 10,351 confirmed clinical *V. cholerae* cases from 2000-2021 in Bangladesh found that *Campylobacter* spp, enterotoxigenic *E. coli* (ETEC) and rotavirus were the most frequently found co-pathogens, with co-infection rates of 6.7%, 5.7% and 2.4%, respectively[80]. Although the effects on the host of co-infection of *V. cholerae* with *Campylobacter* spp. or rotavirus have not been studied, co-infection with enterotoxigenic *E. coli* (ETEC) has been studied. Chowdhury et al.[81] showed that coinfection with ETEC results in an increased host immune response, and so could potentially affect observed symptoms. The authors have also observed a higher co-infection rate (13%) between *V. cholerae* O1 and ETEC in their cohort. However, for future research we will aim to incorporate these variables to provide a more comprehensive understanding of the interactions between host and pathogen, as well as between different pathogens, in the context of cholera. This study should be considered a proof-of-principle to be further investigated and validated with larger sample sizes and different geographical areas.

With the advent of modern technologies, by strengthening bespoke analytical methods and by performing wider comparisons (asymptomatic vs. symptomatic, patients vs. households, environmental vs intestinal *Vibrio*) we can potentially disentangle the intricate network of correlations between the genetic underpinnings of cholera symptoms and epidemiological transmission risk, uncovering regulatory, metabolic and signalling networks interconnectivity that might help to inform future interventions.

## Methods

### Ethics Statement

Informed written consent was obtained from all adult patients, or guardians on behalf of children. Upon receiving consent, the physician collected the patient's sociodemographic characteristics and medical histories. For the icddr,b isolates, the study protocol was approved by the Institutional Review Board of icddr,b (PR-15127). For the IEDCR isolates, the study was performed in accordance with protocols approved by the Institutional Review Board of IEDCR (IEDCR/IRB/09 and IEDCR/IRB/26). Ethics approval was also obtained from the School of Veterinary Medicine and Science Ethics Committee, University of Nottingham (2811 110724).

### Experimental design

For the study we used 129 *V. cholerae* bacterial isolates obtained from distinct stool samples of patients between 2014 and 2021 from the ongoing Nationwide Cholera Surveillance[82], jointly conducted by IEDCR and icddr,b. The isolates were collected from admitted patients from six divisions of Bangladesh (Barisal *n* = 11, Chittagong *n* = 6, Dhaka *n* = 99, Khulna *n* = 2, Rajshahi *n* = 4 and Sylhet *n* = 7). The isolates included in the study were gathered from patients meeting the case definition of diarrhoea and consenting to be included in the surveillance study. The case definition was used and defined as (i) Diarrhoea (patient age > 2 months): any patient attending hospital with 3 or more loose or liquid stools within 24 h or less than 3 loose/liquid stools causing dehydration; (ii) Diarrhoea (patient age < 2 months): changed stool habit from usual pattern in terms of frequency (more than the usual number of purging) or nature of stool (more water than faecal matter). The case definition of diarrhoea was standardised to ensure consistency across different regions and over the collection timeline. Stool samples were processed by either IEDCR or icddr,b research institutes. For the identification of *V. cholerae*, specimens were streaked onto taurocholate-tellurite gelatin agar (TTGA) and incubated overnight at 37 °C. Specimens were also inoculated in alkaline peptone water for enrichment and incubated for an additional 18–24 h[83] and plated on TTGA. Suspected colonies were serotyped with monoclonal antibody specific to *V. cholerae* O1 (Ogawa and Inaba) and O139 serogroups[84] for the icddr,b isolates, while for the IEDCR isolates serotyping and biotyping was carried out by slide agglutination and PCR using primers in Supplementary Data 21. Further confirmation of the isolates being *V. cholerae* was obtained by whole genome sequencing. Confirmed isolates were tested for antimicrobial susceptibility using disk diffusion methods in accordance with CLSI protocols[85] to antibiotics: ampicillin, azithromycin, ciprofloxacin, ceftriaxone, cefixime, doxycycline, erythromycin and meropenem, using commercially available antibiotic discs (Oxoid, Basing- stoke, United Kingdom). *Escherichia coli* American Type Culture Collection 25922 susceptible to all antimicrobials was used as a control strain for susceptibility studies.

Clinical metadata was collected from patients corresponding to 104 isolates for the 129 isolates in our cohort. Clinical data covered 5 categories (duration of diarrhoea, number of stools, abdominal pain, vomiting, and dehydration), in addition the age and sex of the patient and location of the patient was recorded. Clinical symptoms data (Supplementary Data 14) were binned into categories and ranked in order of increasing severity for data analysis.

- Duration of diarrhoea: number of days the diarrhoea persisted was recorded. Data were binned as a duration score ranging from 1–3, with 1 = < 1 day; 2 = 1–3 days; 3 = 4–6 days.
- Number of stools in 24 h: The number of stools recorded in a 24-h period during the hospital admission was recorded. Data were

binned as a number of stools score ranging from 1–5 with 1 = 3–5 times; 2 = 6–10 times; 3 = 11–15 times; 4 = 16–20 times; 5 = 21+ times.
- Abdominal pain: the presence or absence of abdominal pain was recorded as a 0 for absence and 1 for present.
- Vomiting: The presence or absence of any vomiting in the 24 h prior to admission was recorded with 0 denoting no vomiting and 1 denoting the occurrence of vomiting
- Dehydration: clinical assessment of dehydration was recorded as none, moderate or severe by the clinician.

## DNA purification and extraction

DNA extraction was performed at North South University. All the *V. cholerae* isolates were subjected to genomic DNA extraction in accordance with the manufacturer's protocol of the QIAamp DNA Mini Kit (Qiagen).

## Library construction and whole-genome sequencing

The library preparation and sequencing of the 129 selected strains were carried out at NGRI (NSU Genomics Research Institute, North South University). To prepare the Illumina libraries, approximately 1 μg of high molecular weight *V. cholerae* genomic DNA was utilised. Barcoded libraries were prepared using the Illumina DNA Prep Kit (product code 20060059, NEB, USA) following the manufacturers protocol. Nextera DNA CD index codes were added to attribute sequences to each sample. Following that, paired-end sequencing with 2 × 151 cycles was performed on the Illumina MiSeq platform at NGRI.

## Genome assembly and annotation

All sequences were pre-processed using the Illumina BaseSpace sequencing hub. To clean the data adapters were trimmed and unidentified bases were removed. Genomes were assembled using SPAdes (v3.12)[86] with default parameters and a coverage cut off value of 20. Genomic contamination was assessed using ContEst16S[87] with only genomes identified as *V. cholerae* retained for further analysis. Contigs with length shorter than 500 nucleotides were filtered out of the final assemblies. Genomes were annotated with Prokka (v1.14.6)[88], using default settings with –addgenesz--usegenus.

Screening of annotated genes against ABR databases, virulence and plasmid databases and in silico subtyping.

The whole-genome sequences were screened against the CARD[89] database (accessed 05-06-2022) using Abricate[90] with a minimum coverage of 70% and minimum identity of 90% to identify known AMR-associated genes in the isolate cohort. Genomes were also screened against the VFDB[91] database using Abricate[90] to find virulence associated genes, with 70% coverage and 90% identity) (accessed 05-06-2022). Plasmid screening was conducted using the PlasmidFinder[92] database in Abricate[90], with 70% coverage and 90% identity) (accessed 05-06-2022); no plasmids were identified in the genome sequences. Sequence types were identified through MLST[93] which mapped the sequences to the PubMLST[94] database.

## Pangenome analysis and generation of genetic features input files

All annotated genomes were used as input for pangenome analysis using Roary v3.13[95]. The core genome alignment was taken as input to produce a file of core gene SNPs present in the cohort using SNP sites 2.5.1[96]. SNPs within intergenic regions (IGRs) were extracted using piggy v1.5[97] to generate an alignment of core intergenic clusters. Variants in this alignment were then called using SNP sites 2.5.1. The presence-absence of accessory gene was found from the output of Roary.

In addition, a further pangenome alignment was created consisting of the 129 isolates in our cohort together with 218 isolates collected in Bangladesh from 2004 to 2022 (The European Nucleotide Archive-

ENA (http://www.ebi.ac.uk/ena), accession codes: PRJDB8664, PRJDB12727, PRJDB13928, PRJNA723557).

## Phylogenetic analysis of *V. cholerae* isolates in our cohort in Bangladesh

For both our cohort alone and our cohort together with publicly available Bangladeshi isolates (as detailed above) maximum likelihood phylogenies were reconstructed. Using the core genome alignments generated in Roary v3.13[95], the phylogenies were reconstructed in IQ Tree (v2.2.0.3)[98] with 10000 ultrafast bootstrap replicates and best fitted evolutionary model (HKY + F + I for our cohort only and K3Pu +F + I for the combined Bangladesh alignment) was selected using ModelFinder[99]. The alignment length of the core genome of our cohort was 3459819 nucleotide sites of which 1486 were informative. For the core genome of the combined Bangladeshi isolates, the alignment length was 2086397 nucleotide sites with 844 informative sites. The resulting consensus trees were visualised using iTol v6[100], and branches with less than 95% ultrafast bootstrap support were deleted.

## Phylogenetic relations between *V. cholerae* isolates worldwide

We used WGS data from 1140 *V. cholerae* isolates collected from India, Africa, Haiti and Yemen together with our Bangladesh samples (see Supplementary Data 2 and 3). To generate the input for a phylogenetic tree, SNP variants were called from each isolate against the reference genome VC N16961 (NC_002505.1; NC_002506.1) using Snippy v4.6.0[101] (https://github.com/tseemann/snippy). The cleaned alignment files from Snippy were concatenated via the SeqIO function of biopython v1.83[102] then recombination was masked using Gubbins (v.2.3.4)[103]. The filtered polymorphic sites output from Gubbins was further filtered using SNP-sites[96]. The final SNP input contained 4033464 nucleotide sites with 26995 informative sites. This recombination-free SNP output was then used as input to reconstruct the phylogeny using IQtree (v2.2.0.3)[98] with 1000 ultrafast bootstrap replicates and best fitted model (K3Pu+F + I + G4) was selected by ModelFinder[99]. The sequence ERR025382 (Indonesia-1957) was used as an outgroup, and the tree was rooted here. The resulting consensus tree was visualised using iTol v6[93], and branches with less than 95% ultrafast bootstrap support were deleted.

## Transcriptional binding motifs

Motif searches were conducted using FIMO (Find Individual Motif Occurrences[104] within the MEME (Multiple Em for Motif Elicitation)[105] suite (https://meme-suite.org/meme/tools/fimo). Reference sequences of intergenic regions of DNA from our isolates were generated in Piggy as described above; these were used as input for FIMO. To predict the TFBs the following databases were used: CollecTF (Bacterial TF Motifs); Prokaryotes (Prodoric Release 8.9); Prokaryotes (RegTransBase v4); Combined Prokaryotes. Intergenic regions where motifs were found were variant called using SNP-sites[96] and then aligned to the motif sequences using Clustal Omega v1.2.4[106]. For visualisation of intergenic regions, alignment maps of the intergenic regions were created using Jalview 2.11.3.2 with easyfig python genome figure package[107].

## Promoter analysis for Intergenic SNPs

BPROM/softberry[108] was used to predict promoter region and oligonucleotides from known TF binding sites close to the promoter region.

## Genome-scale metabolic model

All simulations were performed using the Python cobra toolkit v0.26.2. The analysis was conducted on both a manually curated and validated model of *V. cholerae* O1 N16961, iAM-Vc960, taken from Abdel-Haleem et al.[19] and on automatically generated draft strain-specific GSM models. The strain-specific draft models were generated using CarveMe[27]. CarveMe was run using the CPLEX solver and gram negative

template, with gap filling for LB and M9 media using the command: 'carve input.faa --gapfill M9,LB -u gramneg --solver cplex --output model.xml'. Gene essentiality, FVA and FBA analyses as described below were conducted on genes of interest in the generalised iAM-Vc960 and in each of the 129 draft strain-specific models, based on the analysis pipeline in Pearcy et al.[39].

For all gene essentiality, FBA and FVA analyses, a knockout model for each gene of interest was constructed by blocking all corresponding reactions to zero, given that the reaction is not catalysed by an isozyme. We considered the essentiality of a gene under both rich medium conditions and M9 minimal medium conditions. To mimic rich medium conditions, the model was constrained to allow all carbon sources into the system, with a fixed uptake rate of 1 mmol/gDCW/h. If a feasible solution exists, while maximising the biomass equation as the objective function, then the knockout of the gene was not essential. To mimic M9 minimal medium conditions, the model was constrained so one individual carbon source had a maximum uptake of 10 mmol/gDCW/h. This simulation (minimal medium condition) was repeated for each carbon source in the model. The genes whose corresponding knockout model achieved a growth rate of $0.0001\,h^{-1}$ or less were considered essential. Flux variability analysis (FVA) was applied to the wild-type model and each knockout model using the cobra toolbox in python[109]. FVA calculates the minimum and maximum flux through each reaction in the model, given a set of constraints, resulting in the range of possible fluxes for each reaction (flux span). FVA was simulated using glucose as the only carbon source in aerobic minimal M9 medium conditions. Note that reaction loops in the solution were not allowed. A gene knockout was considered to significantly affect the flux if the flux span of at least one reaction was changed by greater than 10% compared to the wildtype solution. For the FBA analysis, a drain reaction (i.e., a reaction that consumes the metabolite of interest) was added to the GSM model for each metabolite. The maximum theoretical yield of each metabolite was calculated by setting its corresponding drain reaction as the objective function, with glucose as the only carbon source in aerobic minimal M9 medium conditions. All metabolites contained within the model were considered in the FBA analysis. In iAM-Vc960 this was 1741 different metabolites, whilst in the draft strain-specific GSM models the number of metabolites spanned the range 1321–1433. The simulations were carried out for the wild-type model and each gene knockout model. A gene knockout was considered to significantly affect metabolite yield if the yield of at least one metabolite was reduced to zero, given that it was non-zero in the wildtype. For each of the selected genes of interest, molecular function, pathways and biological processes were taken from the BioCyc database[110] using the SMART tables for *V. cholerae* O1 biovar El Tor strain N16961. These were added to Supplementary Data 11, 12, 17 and 18 to give context to the analysis results.

## Network analysis based on core genome SNPs

Network of our cohort of 129 *V. cholerae* isolates was created using a pairwise hamming distance comparison based on core genome SNPs in python (NetworkX v2.8.4[111] and Matplotlib v3.6.2[112]). Each node represents an isolate while the edge represents the hamming distance between two isolates multiplied by the total number of SNPs found in our cohorts (2382 SNPs). A threshold of 15 or less SNPs difference was used to filter the edges in the network as suggested by Ludden et al.[113] and used by us previously[17,18].

## Statistical analysis and machine learning of genomic features correlated to a specific lineage or clinical symptoms

To assess if the genomic features were associated with a lineage or to a clinical symptom, we employed a fisher exact test[9,10]. Furthermore, to analyse the relationship between genomic features of the BD-1.2 lineage and clinical symptoms a machine learning pipeline was employed.

Clinical data were collected from 104 out of 129 *V. cholerae* isolates of which 63 belonged to the BD-1.2 lineage. These clinical symptoms were be divided into two groups: binary (vomiting and abdominal pain) and multi-class (dehydration, number of stools and duration of diarrhoea), with the binning within each group described above. In the multiclass group, we applied a one-vs-one approach, i.e., each class is compared individually to another class. For example, dehydration class Moderate is compared against class Severe. For both binary and multiclass groupings, as the classes were unbalanced, we oversampled the minority class as a pre-processing step using a Synthetic Minority Oversampling Technique approach (SMOTE)[114]. The Python package Scikit-learn version 1.2.1[115] was used to make the classification and the package Scipy version 1.9.3[116,117] was used to select the most important features based on a Fisher exact test.

The pipeline first removes features that are either present or absent in all the samples. Second, to measure the influence of confounding effects in the data, it uses a two-sided chi-square test of independence to measure the dependency between the confounding effects (sex of patient, age range of patients, year of collection, location of patient, serology of *V. cholerae*) and the phenotype classes ($p$-value < 0.01 with Bonferroni correction); if the null hypothesis is rejected (i.e. there is a dependency between the confound effect and the phenotype) the pipeline checks if there are features that are dependent on the confounding effect again based on a two-sided chi-square test of independence ($p$-value < 0.01 with Bonferroni correction); if there are features where the null hypothesis is rejected, these features are removed from the analysis. Next, the pipeline oversamples the minority class using a SMOTE approach. Then, based on the oversampled data, it selects the most important features using a two-sided Fisher exact test ($p$-value < 0.1). This process is done in two parts: i) to improve randomisation in the pipeline and avoid confounding effects, a loop over 1000 different random seeds is used for the SMOTE approach in order for it to have different initialisations; for each loop the most important features are selected based on the Fisher exact test; ii) then, the features that are selected in over 75% of the different initialisations are deemed important and a random initialisation is selected that contains all these important features to be used for the prediction models. Next, a panel of machine learning methods (logistic regression (LR), linear support vector machine (L-SVM), radial basis function support vector machine (RBF-SVM), extra-tree classifier, random forest, Adaboost and XGboost) was used to predict the clinical symptoms classes based on the pre-selected features described above. The hyperparameters used were:

- Logistic Regression: inverse of regularisation strength C = [0.0001, 0.001, 0.01, 0.1, 1, 10, 100, 1000, 10000];
- Linear SVM: penalty parameter of the hinge loss error C = [0.0001, 0.001, 0.01, 0.1, 1, 10, 100, 1000, 10000];
- Random Forests, Extra Trees and Adaboost: Number of estimators = [2, 4, 8, 16, 32, 64, 128, 256];
- Non-linear SVM with RBF kernel: γ (RBF kernel coefficient) = [0.0001, 0.0001, 0.001, 0.01, 0.1, 1] and C (L2 penalty parameter) = [0.0001, 0.001, 0.01, 0.1, 1, 10, 100, 1000, 10000];
- XGBoost: Number of estimators = [2, 4, 8, 16, 32, 64, 128, 256] and learning rate = [0.0001, 0.001, 0.01, 0.1, 1].

As per previous works[17,18,29,30]: (i) nested cross-validation[118,119] was employed to assess the performance and select the hyper-parameters of the proposed classifiers and to compare the results obtained by the seven different classifiers used; (ii) a Friedman Statistical F-test ($F_F$) with Iman-Davenport correction was used for statistical comparison of multiple classifiers across multiple analyses[120]; (iii) a post-hoc Nemenyi test was employed to find if there is a single classifier or a group of classifiers that performs statistically better in terms of their average AUC rank after the $F_F$ test has rejected the null hypothesis (stating that the performance of the comparisons on the individual classifiers over

the different datasets is similar)[120]; (iv) an undirected graph was created using NetworkX[111] to visualise how the features (accessory genes, core genome SNPs and intergenic SNPs) correlate with different clinical symptoms.

## Protein-protein interaction network and building protein 3D structures

Protein-protein interaction networks of the proteins encoded by the genes associated with clinical symptoms were obtained using STRING database v12.0 (using reference genome *V. cholerae* O1 biovar El Tor str. N16961) and analysed in Cytoscape 3.10.1[121]. Eighty-one accessory and core genes selected by machine learning were used as input for the PPI, of these only 60 could be mapped to the STRING database. The interactome was constructed using first and second neighbour proteins. Disconnected nodes and nodes with interaction scores lower than medium confidence level (interaction scores <0.400), according to StringDB[122], were filtered out. Functions of the protein in the network were annotated with Gene Ontology terms (biological process, molecular function, cellular component and KEGG pathways) in StringDB[122]. Three-dimensional AlphaFold[123] predicted models were obtained by aligning the protein FASTA sequence to reference sequences from the Uniprot database[124] to find a 3D protein structure. 3D protein structures were then visualised using UCSF Chimera[125] and UCSF ChimeraX[126]. Protein stability analysis and the effect of each mutation were performed with DUET[127], DynaMut[128] and SIFT[129]. The electrostatic potential was analysed and visualised using PDB2PQR and APBS accessed online[130], UCSF ChimeraX[126] and APBS Colouring[130].

## Statistical analysis

Statistical comparisons were made using the SciPy package implementing: 1. A two-sided chi-squared test with Bonferroni correction to evaluate the similarities between the serotypes and the collection year and location of the isolates ($p$-value < 0.005); 2. A two-sided Mann–Whitney $U$ test to evaluate the distribution of the counts of accessory genes, coding and non-coding SNPs in BD-1.2 and BD-2 lineages and along the different collection years ($p$-value < 0.005); 3. A two-sided Fisher exact test, with Bonferroni correction, to assess the relationship between the BD-2 and BD-1.2 lineages and different genomic features - core and intergenic SNPs and accessory genes ($p$-value < 0.005); 4. A two-sided hypergeometric enrichment test (two-sided) with false discovery rate (FDR) for the GSM analysis ($p$-value < 0.01); and 5. A two-sided chi-square test of independence to test if there are symptoms/features that are dependent on the confounding effect ($p$-value 0.01 with Bonferroni correction); 6. A two-sided Fisher exact test to select the most important features in the machine learning pipeline ($p$-value < 0.1); 7. A two-sided Friedman Statistical F-test (FF) with Iman-Davenport correction for statistical comparison of multiple datasets over the seven different classifiers used ($p$-value < 0.05). With 7 classifiers and 6 clinical symptom models, the Friedman test is distributed according to the F distribution with $7 - 1 = 6$ and $(7 - 1) \times (6 - 1) = 30$ degrees of freedom. Therefore, the critical values for $F(6,30)$ using a $p$-value = 0.05 is 2.42052319. The post-hoc Nemenyi test was used to find if there is a single classifier or a group of classifiers that performs statistically better in terms of their average rank after the FF test has rejected the null hypothesis (stating that the performance of the comparisons on the individual classifiers over the different datasets is similar); 8. A two-sided Mann–Whitney $U$ test was used to assess for lineage differences in the numbers of genes, reactions and metabolites in the generated draft strain-specific GSM models. 9. A two-sided Mann–Whitney $U$ test was used to assess the number of affected reactions and metabolites in knockouts of genes discriminating lineages, compared to randomly selected genes.

## Reporting summary

Further information on research design is available in the Nature Portfolio Reporting Summary linked to this article.

## Data availability

Short-read sequence data for all 129 isolates used in this study are deposited in the NCBI SRA and can be found associated with BioProject number PRJNA1021874 [https://www.ncbi.nlm.nih.gov/bioproject/?term=PRJNA1021874]. All previously published public *V. cholerae* sequences used in this study are held in European Nucleotide Archive-ENA or NCBI repositories under accession numbers supplied in Supplementary Data 2. Reference sequences are available from NCBI under accessions: NC_002505.1 [https://www.ncbi.nlm.nih.gov/nuccore/NC_002505.1], NC_002506.1 [https://www.ncbi.nlm.nih.gov/nuccore/NC_002506.1] and European Nucleotide Archive-ENA under accession: ERR025382 [https://www.ebi.ac.uk/ena/browser/view/ERR025382]. Clinical data used in this study is given in Supplementary Data 14.

## Code availability

The code used in this study and draft strain-specific GSMMs are available in the following GitHub repository: https://github.com/tan0101/VibrioCARE under https://doi.org/10.5281/zenodo.13384700[131].

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

## Acknowledgements

This study was supported by Research England grant [H53802] as part of the Internal Global Challenges Research Fund Award from the University of Nottingham (T.D., M.H., T.S., F.Q.), the Turkish Ministry of National Education (K.B.) and UKRI MRC grant (MR/X009246/1) (M.B.).

## Author contributions

Designed and supervised the study: M.H., Z.H.H, T.S., F.Q. and T.D. Planned the methodology: M.H., Z.H.H, N.Se., T.S. and T.D. Writing—original draft: K.B, A.M.G., M.B. and T.D. Writing—review and editing: K.B, A.M.G., M.B., N.Se. and T.D. Carried out the experiments and collected the samples: A.R., M.H., A.S., J.A., S.U., M.F.R.S., N.Su., A.I.K., Y.A.B., M.H.A., Z.H.H., T.S. and F.Q. Data analysis and visualisation: K.B, A.M.G. and M.B. Acquired funding: M.H., T.S., F.Q. and T.D.

## Competing interests
The authors declare no competing interests.
