## [Peer Review File · Nature Communications]

Core and accessory genomic traits of *Vibrio Cholerae* O1 drive lineage transmission and disease severityREVIEWER COMMENTS

Reviewer #1 (Remarks to the Author):

Manuscript is well written and describes an extensive analysis of strains collected from patients between 2015 and 2021 in Bangladesh. After describing genetic differences between the two lineages, the authors apply a computational approach to identify features associated with transmission dynamics and disease.

The results are of interest to the field and as far as I can ascertain, appropriate handling and analysis of the sequence data has been performed. I do have some questions/concerns over the description/robustness of the described disease measures used to support the second objective, linking lineage to clinical manifestations, and these are listed below:-

Line 91-92 – what parameters equate to ‘severe clinical symptoms’ from the list given?

Line 98 states that “.V. cholerae linked to on-going cholera cases..” yet clinical descriptions (line 639) refer to the isolates “gathered from patients meeting the case definition of diarrhea”. More detail about the case definition is needed. Given the timeline of isolate collection (over 6 years and in 6 different regions), were a standardised set of descriptors shared to ensure clinicians used similar criteria for reporting disease parameters? If so, these should be reported or referenced.

Line 322 -34 – this section is hard to follow as it is not always clear how many or which symptoms are being referred to in terms of when an association is found e.g. “Fifty eight accessory genes.....were identified as strongly associated to the symptoms”. Does this mean at least one symptom? A fuller description of the findings are necessary here and the relationship between individual recorded symptoms needs more explanation.

Table S7 lists the clinical details of patients from which isolates were collected. Presenting a lineage-specific summary of this information would be useful.

Line 617 The authors rightly acknowledge that their study does not consider several host factors in terms of host response to infection. Other limitations, such as possible co-infections, either by other pathogenic species or by more than one lineage of V. cholerae, should also be acknowledged and discussed.

Reviewer #1 (Remarks on code availability):

I clicked on the link and it takes me to the correct place but I do not have the skills to necessarily follow through with testing it.

Reviewer #2 (Remarks to the Author):

Maciel-Guerra and colleagues provide further details on the characteristics of Vibrio cholerae epidemic strains. The work includes an elegant suite of in silico approaches to extract information after genome sequencing. It is of interest the ML approach that aims to bridge clinical data and bacterial genomics.

Comments.

1. I urge the authors to establish a pangenome of each of the lineages. Also to provide data on whether there are differences in the core and accessory genomes for each of the lineages. For this work, authors need to integrate all the available and not only focus on the collection of strains sampled in this work.
2. Authors should consider validating whether the in silico data translate into biological differences.

The fact that the isolates are available is a great asset. Of course not every phenotype can be tested but the following ones can be done as follows:

(i) Effect on antibiotic resistant pattern. I suggest authors should test a broad panel of drugs including detergents, for example.

(ii) Metabolic profile. Authors should consider a Biolog-based approach informed by their bioinformatics findings to define the metabolic capabilities of the strains.

(iii) Authors should consider simple genetic complementation experiments to validate whether some of the most relevant alleles conferring a biological phenotype are necessary and sufficient. This can be done with available tools (for example Tn7 system). For example, authors may choose to introduce the allele into a strain lacking it, and/or assess the function of the introduced allele in a mutant background for the allele.

3. The metabolic pathway section does not clearly indicate whether indeed there are differences (or not) between the lineages. As currently written, it remains a catalogue of reactions.

4. The structural analysis section is well done although it lacks validation whether the changes observed do translate into changes of protein function/biology.

Reviewer #3 (Remarks to the Author):

In this article, the genomic traits of different *Vibrio cholerae* strains were compared. A number of isolates from patients was sequenced. They then identified core genes and SNPs present in the different lineages through genomic analyses, genome-scale modeling, and protein-protein interaction networks. They also identified genetic determinants associated with infection symptoms through machine learning. Overall, this is an interesting work. The manuscript is also well-written and easy to follow. I would recommend it for publication.

I have the following comments:

1. Line 235-237: without looking up the supplemental table, it is not clear why the metabolite yield was computed and for which metabolites. The Methods section also does not clarify this. Generally, a concluding sentence on what the FBA and FVA results mean would be helpful.

2. The authors used the same genome-scale reconstruction of one *V. cholerae* strain for all simulations despite analyzing a multitude of isolates from different *V. cholerae* strains. Why did they not generate isolate-specific reconstructions for the 129 genomes using iAM-Vc960 as a template?

3. The code available at <https://github.com/tan0101/VibrioCARE> does not seem to include the genome-scale modeling simulations.

Reviewer #3 (Remarks on code availability):

The code for flux balance analysis and flux variability analysis simulations does not seem to be present in the repository. The code as well as the simulated media need to be uploaded.

Reviewer #4 (Remarks to the Author):

In their work, Maciel-Guerra et al. study virulence determinants in a collection of clinical isolates of *V. cholerae*. They identify key genomic differences between two prevalent lineages and associated them both with clinical characteristics of infection and characterize them using a variety of in silico approaches. Their work is of high relevance for improving our understanding of the determinants underlying genome evolution in pathogens. I particularly like their approach of functionally interpreting the relevance of genomic variants in the different isolates using genome-scale metabolic modelling and structural analysis.

Major points:

1) The authors use machine learning to associated clinical features with variants in the genomes of the individual isolates. I'm not sure if this approach is really adequate since it probably does not allow to account for potential confounders such as location or patient phenotypes. The authors should clarify or potentially repeat/complement the analysis with a statistical approach that can deal with confounders.

2) The authors use genome-scale metabolic modelling to elucidate the relevance of key mutations they have identified for the metabolic activity of *V. cholerae*. For this purpose, they use a *V. cholerae* metabolic model reconstructed for the substrain O1 which they enrich with information for metabolic subsystems from *Escherichia coli* and other bacterial strains. I wonder why the authors did not simply use an automated tool for metabolic network reconstruction like CarveMe or gapseq. The advantage would be that they could reconstruct metabolic models that are specific to their isolates, the models directly come with proper pathway annotation and on top they could directly link SNPs to presence/absence of reactions if build a unique model for each of their isolates, though I'm not sure if there is sufficient genomic difference between the isolates for the latter analysis.

3) While the authors link mutations in metabolic genes to potential functional outcomes (e.g. changes in growth rate, reaction flux or metabolite production) it would be great to see if those mutations particularly strongly influence the production of individual metabolites and/or growth. Thus, the authors should compare their outcomes to changes in growth rates and/or production of individual metabolites from randomly selected non-essential genes.

4) The authors seem to have made only parts of their code available (on <https://github.com/tan0101/VibrioCARE>). While the machine learning part is included, the metabolic modelling part is lacking and should be added.

Minor:

1) Some of the Supplements were converted to PDF and so where not really readable (e.g. several thousand pages at the end of the main manuscript file or Supplementary Table S6).

Reviewer #4 (Remarks on code availability):

Only parts of the code have been made available. The metabolic modelling part is lacking.

Point-by-point response to reviewers:

We thank the Reviewers for their useful comments, which no doubt contributed to make this a better paper. In the following, a point-by-point response to all the questions and comments is provided. The original questions are in blue, our replies in black.

Detailed information of which figures, and supplementary material were changed is provided below and in the individual responses to the reviewers' comments. In addition, the manuscript title and abstract have been shortened to conform to the journal formatting guidelines.

Items included in this submission:

- i. Cover letter.
- ii. Point-by-point Response to reviewers (ResponsetoReviewers.docx),
- iii. Revised manuscript marked-up copy (Manuscript_markedup.pdf).
- iv. Revised manuscript clean copy (Manuscript.docx).
- v. Revised Supplementary Material with additional figure S17 and supplementary notes 4 and 5 as well as edits to figure S18 (former S17), figure S19 (former S18) and dataset legends.
- vi. Additional datasets S6-10, S12.
- vii. Revisions to datasets S3, S4, S11, S13, S15, S16, S17, S19, S20.

REVIEWER COMMENTS

Reviewer #1

Manuscript is well written and describes an extensive analysis of strains collected from patients between 2015 and 2021 in Bangladesh. After describing genetic differences between the two lineages, the authors apply a computational approach to identify features associated with transmission dynamics and disease.

The results are of interest to the field and as far as I can ascertain, appropriate handling and analysis of the sequence data has been performed. I do have some questions/concerns over the description/robustness of the described disease measures used to support the second objective, linking lineage to clinical manifestations, and these are listed below:-

We thank the reviewer for acknowledging the interest of this manuscript and for appreciating our analysis.

Line 91-92 – what parameters equate to 'severe clinical symptoms' from the list given?

We thank the reviewer for the relevant comment. The reviewer raises an important point in relation to how to determine symptoms' severity and how such severity ultimately determines the overall severity of the condition. As explained now in the methods lines 792-807, some symptoms have a severity defined in a semi-continuous scale (number of stools and duration of diarrhoea), other have a categorical classification (dehydration) and others are assessed in terms of their presence/absence only (vomit and abdominal pain). The physician observes the aggregate of all the aforementioned symptoms and eventually decides the overall severity of the disease.

To expand on the above, we provide the protocol (see below) which has been used by the Bangladeshi physicians in this work to assess the condition of the patients (see also Methods lines 792-807):

1. Dehydration (classified as none, moderate or severe, see below);
2. Duration of diarrhoea (number of days the diarrhoea persisted);
3. Number of stools in 24 hours (the number of stools recorded in a 24-hour period during the hospital admission);
4. Abdominal Pain (the presence or absence of abdominal pain);
5. Vomit (the presence or absence of any vomiting in the 24 hours prior to admission).

For the number of stools and duration of diarrhoea, the degree of severity is quantified in an increasing scale using the indicators reported in the Methods section.

On the contrary, the severity of dehydration is quantified following the WHO guidelines (<https://www.who.int/news-room/fact-sheets/detail/diarrhoeal-disease>):

1. no dehydration (not enough signs to classify as some or severe dehydration).
2. some dehydration (two or more of the following signs):
 - restlessness, irritability;
 - sunken eyes;
 - drinks eagerly, thirsty.
3. severe dehydration (at least two of the following signs):
 - lethargy/unconsciousness;
 - sunken eyes;
 - unable to drink or drink poorly;
 - skin pinch goes back very slowly (≥ 2 seconds).

When assessing a patient's condition, the physician considers all the symptoms above and ultimately uses their clinical judgement to assess the overall severity. There are no fixed thresholds as for example a severe dehydration alone may be enough to consider the patient's overall condition as severe. In addition, a combination of less severe symptoms can also signal a severe condition. For instance, moderate dehydration with persistent vomiting can create a severe clinical picture.

Furthermore, in the original manuscript we sometimes used the wording "severe symptoms" implying a threshold when we meant to refer to the severity of the symptoms (i.e. intended as a scale of values). To reflect the considerations above, we have made the following edits to the manuscript:

Abstract lines 11-14: "GSMM and 3D structure analysis unveiled a complex interplay between transcription regulation, protein interaction and stability, and metabolic networks, associated to lifestyle adaptation, intestinal colonization, acid tolerance and symptom severity."

Introduction lines 60-61: "About 1 in 5 people with cholera will experience a severe condition owing to a combination of symptoms (primarily diarrhoea, vomiting, dehydration) (Baker-Austin *et al.* 2018)."

Introduction lines 82-84: "Additionally, a distinct set of 17 mutations, 39 accessory genes, and four intergenic SNPs were found exclusively linked to the severity of clinical symptoms."

Introduction lines 84-86: "Through detailed GSMMs and 3D structure analysis of these genes, we inferred the mechanistic basis behind the selection of these genomic drivers in BD-1.2 and link to the severity of the symptoms."

Methods lines 792-807: "Clinical metadata was collected from patients corresponding to 104 isolates for the 129 isolates in our cohort. Clinical data covered 5 categories (duration of diarrhoea, number of stools, abdominal pain, vomiting, and dehydration), in addition the age

and sex of the patient and location of the patient was recorded. Clinical symptoms data (**Dataset S13**) were binned into categories and ranked in order of increasing severity for data analysis.

- Duration of diarrhoea: number of days the diarrhoea persisted was recorded. Data were binned as a duration score ranging from 1-3, with 1 = <1 day; 2 = 1-3 days; 3 = 4-6 days.
- Number of stools in 24 hours: The number of stools recorded in a 24-hour period during the hospital admission was recorded. Data were binned as a number of stools score ranging from 1-5 with 1= 3-5 times; 2= 6-10 times; 3=11-15 times; 4=16-20 times; 5=21+ times.
- Abdominal pain: the presence or absence of abdominal pain was recorded as a 0 for absence and 1 for present.
- Vomit: The presence or absence of any vomiting in the 24 hours prior to admission was recorded with 0 denoting no vomiting and 1 denoting the occurrence of vomiting
- Dehydration: clinical assessment of dehydration was recorded as none, moderate or severe by the clinician.”

Line 98 states that “. . . *V. cholerae* linked to on-going cholera cases..” yet clinical descriptions (line 639) refer to the isolates “gathered from patients meeting the case definition of diarrhea”. More detail about the case definition is needed. Given the timeline of isolate collection (over 6 years and in 6 different regions), were a standardised set of descriptors shared to ensure clinicians used similar criteria for reporting disease parameters? If so, these should be reported or referenced.

Thank you for your insightful comments, the case definition used was defined as:

- i. Diarrhoea (patient age > 2 months): any patient attending hospital with 3 or more loose or liquid stools within 24 hours or less than 3 loose / liquid stools causing dehydration.
- ii. Diarrhoea (patient age < 2 months): changed stool habit from usual pattern in terms of frequency (more than the usual number of purging) or nature of stool (more water than faecal matter).

The case definition of diarrhoea was standardized to ensure consistency across different regions and over the collection timeline. In addition, subsequent to sample collection, isolates were confirmed as *V. cholerae* using classical microbiological and molecular biology analysis (see methods) and whole genome sequencing. We have added the case definition to the methods (lines 773-778, and below). Furthermore, we have now updated results lines 91-92 to emphasize that the *V. cholerae* isolates were linked to previous cholera cases in Bangladesh.

Results Lines 91-92: “To explore the evolutionary dynamics of *V. cholerae* linked to cholera cases in Bangladesh, a genomic analysis was done considering the years 2015 to 2021.”

Methods Lines 773-778: “The case definition was defined as: i) Diarrhoea (patient age > 2 months): any patient attending hospital with 3 or more loose or liquid stools within 24 hours or less than 3 loose / liquid stools causing dehydration; ii) Diarrhoea (patient age < 2 months): changed stool habit from usual pattern in terms of frequency (more than the usual number of purging) or nature of stool (more water than faecal matter). The case definition of diarrhoea was standardized to ensure consistency across different regions and over the collection timeline.”

Methods lines 785-786: “Further confirmation of the isolates being *V. cholerae* was obtained by whole genome sequencing.”

Line 322 -34 – this section is hard to follow as it is not always clear how many or which symptoms are being referred to in terms of when an association is found e.g. “Fifty eight

accessory genes.....were identified as strongly associated to the symptoms”. Does this mean at least one symptom? A fuller description of the findings are necessary here and the relationship between individual recorded symptoms needs more explanation.

We thank the reviewer for this comment. In the original sentence “Fifty-eight accessory genes...were identified as strongly associated to the symptoms”, we meant “strongly associated to at least one symptom” (as correctly interpreted by the Reviewer). We have now corrected the sentence and given a fuller explanation of which genes were found to be correlated to which symptoms.

The analysis to obtain such result is: we built independent machine-learning symptom prediction models, each designed to predict either presence/absence of a symptom or the degree of severity, starting from a list of genetic elements (SNPs, accessory genes, etc.) provided as inputs. Once the prediction models were fitted to accurately replicate experimental observations (supervised learning), we could inspect the models to retrieve the input variables (i.e. genetic elements) most strongly correlated to the prediction. A final list of genetic elements was obtained by aggregating all the genetic elements which were found to be correlated to symptom prediction within at least one symptom prediction model. In addition, in the updated manuscript submission we provided a table (**Dataset S16**) where we highlight which genetic elements were found relevant for each symptom prediction model. The same information is also represented in Figure 5 where an undirected graph is used to indicate the identified correlations between genetic elements and symptoms.

As an aside, in the updated manuscript we improved our symptom prediction models by better addressing the detrimental effects of possible confounding factors, as suggested by Reviewer 4. Because of this, in the updated manuscript we are reporting updated results. The following edits are reported in the manuscript results lines 393-412.

Results lines 393-412: “Analysis of the best-performing symptom prediction models allowed us to identify the inputs features (core genome and intergenic SNPs, and accessory genes) most strongly correlated to each symptom (**Dataset S16**). Seventy-nine different features in total were selected as significantly correlated to at least one of the six symptom prediction models, with 68% being selected in two or more models (**Fig. 5**). No features were selected for all symptoms. All features associated with number of stools 11-15 times vs. 21+ times were found associated to at least one of the other five symptom prediction models. Forty-five accessory genes (nine known genes, *tufB_2*, *bhc*, *pckA*, *luxR_2*, *hcpA_1*, *rpoS*, *dcuA*, *hpt*, *luxR*, and 36 hypothetical genes) and 28 core SNPs over 23 genes (14 known, *clpS*, *gshB*, *dapF*, *fabV_1*, *add*, *tufB*, *lpoA*, *phrB*, *yjcS*, *fabH1*, *cysG_2*, *padC*, *pepN*, *tadA_2*, and nine hypothetical genes) were identified as strongly associated to at least one of the symptoms. From the nine known accessory genes: four (*rpoS*, *hpt*, *luxR* and *pckA*) were found in the vomit model; *dcuA* was found in the abdominal pain model; *hcpA_1* was found only in the number of stools 11-15 times vs. 16-20 times; *luxR_2* was found in two models (vomit and dehydration moderate vs severe); *bhc* and *tufB_2* were found in three models (vomit, number of stools 11-15 times vs. 16-20 times and number of stools 11-15 times vs. 21+ times) with *tufB_2* also found in abdominal pain and diarrhoea duration <1 day vs. 1-3 days models. Six SNPs from the genes *tufB*, *dapF*, *clpS*, *gshB* and *fabV* were associated to three symptom prediction models (vomit, number of stools 11-15 times vs. 16-20 times and number of stools 11-15 times vs. 21+ times) with the SNPs from the genes *dapF* and *fabV* also associated with abdominal pain and diarrhoea duration <1 day vs. 1-3 days and the SNP from the gene *tufB* associated with dehydration moderate vs severe.”

Table S7 lists the clinical details of patients from which isolates were collected. Presenting a lineage-specific summary of this information would be useful.

We thank the Reviewer for this suggestion. We have now updated Table S7 (**Dataset S13** now) to include the information regarding the lineage for each of isolates. Moreover, we have

included Fig. S17 to summarise the clinical information for the lineages and described it in Results lines 301-305. Fig. S17 consists of bar plots indicating the overall count of the different categories for each symptom for each lineage. The Mann Whitney test compares the distribution of the symptom classes between the two lineages for each symptom.

Results lines 301-305: “Beyond identifying the potential involvement of new genetic traits in differentiating the BD-1.2 and BD-2 lineages, we hypothesized that the same or additional genetic features might play a significant role in the manifestation and severity of clinical symptoms in patients when infected with *V. cholerae*. A summary of the distribution of each clinical symptom over the two lineages is given in **Fig. S17**. We focused on the lineage BD-1.2, which caused the most recent outbreak in Bangladesh.”

Supplementary Figure S17 Legend: “**Figure S17**. Bar plots indicating the overall count of each category in each symptom (**A**. Dehydration, **B**. Vomit, **C**. Abdominal Pain, **D**. Duration of Diarrhoea, **E**. Number of Stools) over the lineages BD-1.2 and BD-2. Two-sided Mann Whitney U tests were used to assess for statistical differences between lineages for each symptom.”

Line 617 The authors rightly acknowledge that their study does not consider several host factors in terms of host response to infection. Other limitations, such as possible co-infections,

either by other pathogenic species or by more than one lineage of *V. cholerae*, should also be acknowledged and discussed.

Thank you for your insightful feedback.

Specifically concerning co-infection by different strains or lineages of *V. cholerae*, whilst it is theoretically possible that different *V. cholerae* strains or lineages could interact within the host, either competitively or synergistically, leading to variations in virulence and pathogenicity (Harris *et al.* 2009; Abdel-Haleem *et al.* 2020; Zhang *et al.* 2016), to our knowledge, no evidence we could not find such an occurrence. For example, Madi *et al.*, 2024 conducted a study performing metagenomic sequencing of 260 stool samples, each collected from a cholera patient. From a reads analysis the authors concluded that *V. cholerae* reads appeared to be from a single strain in all cases. In agreement with this finding, Levade *et al.* 2017 and Lypaczewski *et al.* 2024 addressed this issue by taking multiple isolates from the same individual and also concluded that there was no evidence of infection from multiple *V. cholerae* lineages or strains. We would greatly appreciate if the Reviewer could provide references supporting this occurrence as such an inclusion would contribute to improving the paper.

Concerning co-infections with other pathogens, we agree that these would be important to consider, as they may act as further confounders in the search for correlations between genetic elements and clinical symptoms. Similarly, other factors such as the host's microbiome, genetic background or immune response, can also significantly influence the outcome of infections (Littman and Pamer 2011; Round and Mazmanian 2009).

We appreciate that in our work no information regarding co-infection was collected. Conversely, a study of 10,351 confirmed clinical *V. cholerae* cases from 2000-2021 in Bangladesh found that *Campylobacter* spp., enterotoxigenic *E. coli* (ETEC) and rotavirus were the most frequently found co-pathogens, with co-infection rates of 6.7%, 5.7% and 2.4% respectively (Das *et al.* 2023). We found no studies that considered the effects on the host of *V. cholerae* *Campylobacter* spp. or rotavirus co-infections, however, co-infection with enterotoxigenic *E. coli* (ETEC) has been studied. Chowdhury *et al.* 2010 showed that coinfection with ETEC with *V. cholerae*, results in an increased host immune response, and so could potentially affect observed symptoms. The authors have also observed that about 13% of hospitalized diarrheal patients are concomitantly infected with *V. cholerae* O1 and ETEC.

We thank the reviewer for pointing to this limitation of our study and have now updated the discussion on lines 739-752 to clarify this. In future research, we aim to include co-infection data, where available, to better understand the multifaceted interactions between host and pathogen, as well as between different pathogens.

By acknowledging these limitations, we hope to provide a clearer context for our findings and pave the way for more holistic studies in the future. Based on this we have now updated discussion lines 739-752 to include a summary of these limitations.

Discussion lines 739-752: "A further limitation of this study was the inability to consider the potential impact of co-infections with either multiple *V. cholerae* lineages/strains or other pathogens. Whilst the presence of more than one *V. cholerae* strain or lineage in a host has recently been shown to be unlikely (Madi *et al.*, 2024, Levade *et al.* 2017, Lypaczewski *et al.* 2024), co-infections with other bacteria can occur in diarrheal patients. A study of 10,351 confirmed clinical *V. cholerae* cases from 2000-2021 in Bangladesh found that *Campylobacter* spp., enterotoxigenic *E. coli* (ETEC) and rotavirus were the most frequently found co-pathogens, with co-infection rates of 6.7%, 5.7% and 2.4% respectively (Das *et al.* 2023). Although the effects on the host of co-infection of *V. cholerae* with *Campylobacter* spp. or rotavirus have not been studied, co-infection with enterotoxigenic *E. coli* (ETEC) has been studied. Chowdhury *et al.* 2010 showed that coinfection with ETEC results in an increased host immune response, and so could potentially affect observed symptoms. The authors have also observed a higher co-infection rate (13%) between *V. cholerae* O1 and ETEC in their cohort.

However, for future research will aim to incorporate these variables to provide a more comprehensive understanding of the interactions between host and pathogen, as well as between different pathogens, in the context of cholera.”

Abdel-Haleem, Alyaa M, Vaishnavi Ravikumar, Boyang Ji, Katsuhiko Mineta, Xin Gao, Jens Nielsen, Takashi Gojobori, and Ivan Mijakovic. 2020. 'Integrated metabolic modeling, culturing, and transcriptomics explain enhanced virulence of *Vibrio cholerae* during coinfection with enterotoxigenic *Escherichia coli*', *mSystems*, 5: e00491-20.

Chowdhury, Fahima, Yasmin A. Begum, Mohammad Murshid Alam, Ashraful I. Khan, Tanvir Ahmed, M. Saruar Bhuiyan, Jason B. Harris, Regina C. LaRocque, Abu S. G. Faruque, Hubert Endtz, Edward T. Ryan, Alejandro Cravioto, Ann-Mari Svennerholm, Stephen B. Calderwood, and Firdausi Qadri. 2010. 'Concomitant Enterotoxigenic *Escherichia coli* Infection Induces Increased Immune Responses to *Vibrio cholerae* O1 Antigens in Patients with Cholera in Bangladesh', *Infection and Immunity*, 78: 2117-24.

Das, R., S. Nasrin, P. Palit, R. A. Sobi, A. A. Sultana, S. H. Khan, M. A. Haque, S. Nuzhat, T. Ahmed, A. S. G. Faruque, and M. J. Chisti. 2023. '*Vibrio cholerae* in rural and urban Bangladesh, findings from hospital-based surveillance, 2000-2021', *Scientific Reports*, 13: 6411.

Harris, Jason B., Michael J. Podolsky, Taufiqur R. Bhuiyan, Fahima Chowdhury, Ashraful I. Khan, Regina C. LaRocque, Tanya Logvinenko, Jennifer Kendall, Abu S. G. Faruque, Cathryn R. Nagler, Edward T. Ryan, Firdausi Qadri, and Stephen B. Calderwood. 2009. 'Immunologic Responses to *Vibrio cholerae* in Patients Co-Infected with Intestinal Parasites in Bangladesh', *PLoS Neglected Tropical Diseases*, 3: e403.

Levade, Inès, Yves Terrat, Jean-Baptiste Leducq, Ana A. Weil, Leslie M. Mayo-Smith, Fahima Chowdhury, Ashraful I. Khan, Jacques Boncy, Josiane Buteau, Louise C. Ivers, Edward T. Ryan, Richelle C. Charles, Stephen B. Calderwood, Firdausi Qadri, Jason B. Harris, Regina C. LaRocque, and B. Jesse Shapiro. 2017. '*Vibrio cholerae* genomic diversity within and between patients', *Microbial Genomics*, 3.

Littman, D. R., and E. G. Pamer. 2011. 'Role of the commensal microbiota in normal and pathogenic host immune responses', *Cell Host & Microbe*, 10: 311-23.

Lypaczewski, P., D. Chac, C. N. Dunmire, K. M. Tandoc, F. Chowdhury, A. I. Khan, T. Bhuiyan, J. B. Harris, R. C. LaRocque, S. B. Calderwood, E. T. Ryan, F. Qadri, B. J. Shapiro, and A. A. Weil. 2024. 'Diversity of *Vibrio cholerae* O1 through the human gastrointestinal tract during cholera', *bioRxiv*.

Madi, N., E. T. Cato, M. Abu Sayeed, A. Creasy-Marrazzo, A. Cuénod, K. Islam, M. I. U. Khabir, M. T. R. Bhuiyan, Y. A. Begum, E. Freeman, A. Vustepalli, L. Brinkley, M. Kamat, L. S. Bailey, K. B. Basso, F. Qadri, A. I. Khan, B. J. Shapiro, and E. J. Nelson. 2024. 'Phage predation, disease severity, and pathogen genetic diversity in cholera patients', *Science*, 384: eadj3166.

Round, J. L., and S. K. Mazmanian. 2009. 'The gut microbiota shapes intestinal immune responses during health and disease', *Nature Reviews: Immunology*, 9: 313-23.

Zhang, Shun-Xian, Yong-Ming Zhou, Wen Xu, Li-Guang Tian, Jia-Xu Chen, Shao-Hong Chen, Zhi-Sheng Dang, Wen-Peng Gu, Jian-Wen Yin, Emmanuel Serrano, and Xiao-Nong Zhou. 2016. 'Impact of co-infections with enteric pathogens on children suffering from acute diarrhea in southwest China', *Infectious Diseases of Poverty*, 5: 64

Reviewer #1 (Remarks on code availability):

I clicked on the link and it takes me to the correct place but I do not have the skills to necessarily follow through with testing it.

We appreciate the reviewer's validation of the link.

Reviewer #2

Maciel-Guerra and colleagues provide further details on the characteristics of *Vibrio cholerae* epidemic strains. The work includes an elegant suite of in silico approaches to extract information after genome sequencing. It is of interest the ML approach that aims to bridge clinical data and bacterial genomics.

We thank the reviewer for their assessment of this work.

Comments.

1. I urge the authors to establish a pangenome of each of the lineages. Also to provide data on whether there are differences in the core and accessory genomes for each of the lineages. For this work, authors need to integrate all the available and not only focus on the collection of strains sampled in this work.

We thank the reviewer for these comments and for the suggested analyses. We have conducted a pangenome analysis for each lineage, examined differences in the core and accessory genomes, and integrated sequences beyond our collection.

Overall comparing the core and accessory genome sequences across two closely related lineages using both a combined pangenome and the lineage-specific pangenome approach yielded identical sets of core and accessory genes, after fixing annotation issues arising with lineage-specific pangenomes (described below).

In addition, comparison of the BD-1.2 and BD-2 isolates in both our cohort and publicly available data by either constructing separate pangenomes for each lineage or a combined pangenome confirms that our cohort is broadly representative of all available BD-1.2 and BD-2 genome sequences available to date, and further confirms that the pipeline using a combined genome gives better results than analysing independent pangenomes, which results in many false positives due to annotation ambiguities.

However, the suggested analysis has proven valuable, and we have compared it with the pipeline used in the original manuscript. While we do not believe it replaces the original pipeline for several reasons detailed below, it does add value to the manuscript.

For clarity we would first like to explain the issues that the method suggested by the reviewer (i.e. establishing a separate pangenome for each lineage) presents compared to the pipeline we did (one pangenome for both lineages).

When creating a pangenome, the annotation of each gene is done by comparing the sequence similarity across all isolates' sequences. Those genes present in 99% of the isolates (in our case 128 or all 129 sequences) are identified as belonging to the core of the pangenome, and those present in less than 99% (in our case 127 sequences or less) are identified as being in the accessory of the pangenome. This allows to compare which genes are present and conserved across the sequences of all isolates in the pangenome and which genes are not conserved, being present in only some isolates. To achieve this, initially the DNA sequences are annotated using software such as Prokka to produce CDS and gene annotations in the format of gff files. Then a pangenome software (Roary, in this case), takes the sequences of each gene (taking as an input gff files from the gene annotation pipeline) and performs a

complex clustering of similar sequences using multiple methods. In detail, the clustering process begins by converting CDS sequences to protein sequences, then the actual clustering is performed using CD-HIT and comparing the protein sequences against each other with BLASTP at 95% identity. Further clustering is performed using the Markov clustering algorithm (MCL). Paralog and homolog issues are resolved by leveraging gene synteny information.

During this process, genes that may have been initially annotated (e.g. by Prokka) with different names but are variants of the same gene may be clustered together. The annotation issue in Prokka arises because reference databases, like NCBI and Uniprot used to assign names in the initial sequence annotation, contain variants of the same gene from different bacterial strains annotated with different names but with high sequence similarity. For example, variants of the same gene such as *hlyA* and *hlyA_1*, which share 99% protein sequence identity, are treated as distinct gene entries despite being essentially the same gene and are clustered together and assigned the most frequent gene name (i.e. *hlyA*, in this case). If using two separate pangenomes we would encounter a major issue as explained below in (i) Gene naming consistency.

By constructing a combined pangenome, we were able to identify and compare the conserved genes (core genes) in both lineages (BD-1.2 and BD-2) and the genes present in variable proportions in each lineage (accessory genes). On the contrary if using two separate pangenomes we would encounter another major issue as explained below in relation to: (ii) Core vs. accessory gene classification.

We then examined statistical differences in the presence of each gene in the BD-1.2 and BD-2 lineages to identify genes exclusively or predominantly present in one lineage. For core genes present in all isolates, the software produced an alignment of the core gene sequences, enabling us to detect lineage-specific mutations at a finer level. Again, if on the contrary we use separate lineage pangenomes, we would encounter another major issue in relation to (iii) core gene alignment and variant calling.

Finally, using the software Piggy, we clustered and compared the intergenic regions of each gene sequence, identifying mutations in core intergenic regions. Again, if separate lineage pangenomes were used, we would encounter another major issue in relation to (iv) Intergenic region comparison.

Let us now discuss the four anticipated major issues encountered during the analysis steps, when following the route making use of two separate pangenomes (one per lineage).

- (i) Gene naming consistency: The complex procedure for clustering genes relies on sequence similarity estimated across all gene sequences within the cohort and their synteny. When performing similarity and clustering analyses on separate datasets (i.e. separate lineages), as explained above with the Uniprot example of *hlyA* and *hlyA_1*, which share 99% protein sequence identity and are variants of the same gene, are treated as distinct gene entries despite being essentially the same gene. For example, consider a conserved gene, geneX, with two highly similar variants, geneX_A and geneX_B with 98% protein sequence similarity. Although variants of the same gene, they may be named geneX_A in BD-1.2 core genome (if this variant is predominant in BD-1.2) and geneX_B in BD-2 core genome (if this variant is predominant in BD-2). Comparing these independent lineages could mistakenly suggest that geneX_A is exclusive to BD-1.2 and geneX_B to BD-2, preventing further comparison. In contrast, constructing a combined pangenome, as done in our original pipeline, would correctly cluster and align these variants under a single gene name, geneX, allowing us to identify mutations (i.e. variants) specific to each lineage.

- (ii) Core vs. accessory gene classification: In separate pangenomes, some genes may be core in one lineage (because they are present in all sequences) and accessory in another (because they are present in some sequences). This would require manual correction to ensure fair comparison, increasing the risk of errors due to annotation ambiguities. On the contrary, with a combined pangenome, these genes are automatically considered accessory and correctly processed for presence/absence comparison. For example, if gene X is present in all sequences of lineage A and in some sequences of lineage B, it would be core in A and accessory in B when processed separately. Combining pangenomes ensures gene X is treated consistently as an accessory gene. If instead two pangenomes are created there is the risk that in one lineage a variant calling is done, as the gene would be erroneously considered a core gene. To illustrate this concept, consider two lineages A and B with 100 sequences in each. If the same gene X is present in 100 sequences of lineage A and in 10 sequences of lineage B, it would be considered a core gene in lineage A and an accessory gene of lineage B if the pangenomes are done independently. This would require manual correction to compare across lineages as presence/absence. If instead the pangenomes are combined gene X is in 110 sequences of 200 and so automatically considered an accessory gene and processed using a presence/absence comparison across lineages.

- (iii) Core gene alignment and variant calling: Part of our analysis involves comparing mutations in genes present in the core of both BD-1.2 and BD-2. This is possible because core genes are aligned into a multi-sequence alignment file, enabling single nucleotide variant calling. With separate pangenomes, we lack a single alignment across both lineages, complicating the assessment of variants in conserved genes. Two potential workarounds include: a) Manually mapping variants by matching gene names and positions, however this is error-prone and complicated by gene naming issues (see point (i)); and b) Alternatively, aligning all sequences to a single reference genome and calling variants, then comparing across lineages. However, this method has limitations as the chosen reference genome may not represent either lineage accurately. For *V. cholerae* O1 El Tor the established reference is *V. cholerae* O1 El Tor N16961 originating from Bangladesh in 1975, which is neither from BD-2 or BD-1.2 lineages. Another *V. cholerae* O1 El Tor reference strain sometimes used is C6706, but again this sequence originated from another lineage, being isolated in Peru in 1991. Due to the lineage specific difference in our isolates compared to the reference genomes from other lineages, using a reference to call the variants between BD-2 to BD-1.2 would be disadvantageous compared to the combined pangenome method employed by us. The combined pangenome approach allows for direct comparison of mutations in the two lineages, providing a more accurate analysis.

- (iv) Intergenic region comparison: Our original pipeline allowed us to cluster, align, and compare intergenic regions using Piggy, which utilized the combined pangenome output. With separate pangenomes, mapping intergenic regions across lineages would be time-consuming and error-prone, as these regions may vary in length and upstream and downstream genes.

In summary of the above paragraphs, using lineage-specific pangenomes naturally leads to multiple annotation errors, which we were able to fix only because we had the results from the combined pangenome available. Notably, once the results from the lineage-specific pangenomes were fixed, they were found identical to those of the combined pangenome.

Nonetheless, there is value in analysing separate pangenomes for general applications, as population structure can bias pangenomes (Horesh *et al.* 2021). Consequently, we have added this additional analysis to the Results (lines 210-213), Supplementary Material, and supplementary datasets S6-S10.

Results Lines 210-213: “To assess for any additional genes separating the BD-1.2 and BD-2 lineages we also conducted an analysis on the pangenomes of the lineages separately but found the results in line with those of the combined pangenome analysis presented above (Supplementary Note 4 and Datasets S6-S10)”.

Supplementary Material: “**Supplementary Note 4**”

Using a combined pangenome may introduce biases due to the population structure of the underlying sequences. To address this concern, we conducted additional analyses:

Firstly, we constructed lineage-specific pangenomes for BD-1.2 and BD-2, comparing both core and accessory genes across each lineage.

Secondly, we repeated the analysis using isolates from our cohort along with publicly available BD-1.2 and BD-2 isolates. These analyses broadly confirmed our original results and are summarized below. However, it is important to note that while employing independent pangenomes addresses potential biases related to population structure, it also introduces several challenges within the framework of our comparative analysis. Specifically:

- (i) Gene naming consistency: The complex procedure for clustering genes relies on sequence similarity estimated across all gene sequences and gene synteny. When performing similarity and clustering analyses on separate datasets (i.e. separate lineages), as explained above with the Uniprot example of *hlyA* and *hlyA_1*, which share 99% protein sequence identity and are variants of the same gene, are treated as distinct gene entries despite being essentially the same gene. For example, consider a conserved gene, geneX, with two highly similar variants, geneX_A and geneX_B with 98% protein sequence similarity. Although variants of the same gene, they may be named geneX_A in BD-1.2 core genome (if this variant is predominant in BD-1.2) and geneX_B in BD-2 core genome (if this variant is predominant in BD-2). Comparing these independent lineages could mistakenly suggest that geneX_A is exclusive to BD-1.2 and geneX_B to BD-2, preventing further comparison. In contrast, constructing a combined pangenome, as done in our original pipeline, would correctly cluster and align these variants under a single gene name, geneX, allowing us to identify mutations (i.e. variants) specific to each lineage.

- (ii) Core vs. accessory gene classification: In separate pangenomes, some genes may be core in one lineage (because they are present in all sequences) and accessory in another (because they are present in some sequences). This would require manual correction to ensure fair comparison, increasing the risk of errors due to annotation ambiguities. On the contrary, with a combined pangenome, these genes are automatically considered accessory and correctly processed for presence/absence comparison. For example, if gene X is present in all sequences of lineage A and in some sequences of lineage B, it would be core in A and accessory in B when processed separately. Combining pangenomes ensures gene X is treated consistently as an accessory gene. If instead two pangenomes are created there is the risk that in one lineage a variant calling is done, as the gene would be erroneously considered a core gene. To illustrate this concept, consider two lineages A and B with 100 sequences in each. If the same gene X is present in 100 sequences of lineages A and in 10 sequences of lineage B, it would be considered a core gene in lineage A and an accessory gene of lineage B if the pangenomes are done independently. This would require manual correction to compare across lineages as presence/absence. If instead the pangenomes are combined gene X is in 110 sequences of 200 and so automatically considered an accessory gene and processed using a presence/absence comparison across lineages.

- (iii) Core gene alignment and variant calling: Part of our analysis involves comparing mutations in genes present in the core of both BD-1.2 and BD-2. This is possible because core genes are aligned into a multi-sequence alignment file, enabling single nucleotide variant calling. With separate pangenomes, we lack a single alignment across both lineages,

complicating the assessment of variants in conserved genes. Two potential workarounds include: a) Manually mapping variants by matching gene names and positions, however this is error-prone and complicated by gene naming issues (see point (i)); and b) Alternatively, aligning all sequences to a single reference genome and calling variants, then comparing across lineages. However, this method has limitations as the chosen reference genome may not represent either lineage accurately. For *V. cholerae* O1 El Tor the established reference is *V. cholerae* O1 El Tor N16961 originating from Bangladesh in 1975, which is neither from BD-2 or BD-1.2 lineages. Another *V. cholerae* O1 El Tor reference strain sometimes used is C6706, but again this sequence originated from another lineage, being isolated in Peru in 1991. Due to the lineage specific difference in our isolates compared to the reference genomes from other lineages, using a reference to call the variants between BD-2 to BD-1.2 would be disadvantageous compared to the combined pangenome method employed by us. The combined pangenome approach allows for direct comparison of mutations in the two lineages, providing a more accurate analysis.

- (iv) Intergenic region comparison: Our original pipeline allowed us to cluster, align, and compare intergenic regions using Piggy, which utilized the combined pangenome output. With separate pangenomes, mapping intergenic regions across lineages would be time-consuming and error-prone, as these regions may vary in length and upstream and downstream genes.

Despite these challenges, both our combined pangenome pipeline and the lineage-specific pipeline generally align, as described below.

Construction of separate pangenomes for BD-1.2 and BD-2 isolates in our cohort.

Using Roary (Page *et al.* 2015), as in the combined pangenome analysis, we generated the pangenome for the 45 BD-2 isolates and separately for the 84 BD-1.2 isolates in our cohort (separate pangenomes). **Dataset S6** provides the number of core, accessory, and unannotated genes in each pangenome, showing comparable numbers of core and accessory genes in both.

Difference and similarities of separate lineage-specific pangenomes compared against a single pangenome for both BD-1.2 and BD-2, using our cohort

As shown in **Dataset S7**, the comparison of the core and accessory genes between the lineages demonstrates a significant overlap. Specifically, the same 2152 genes are present in both the BD-1.2 and the BD-2 core genomes, with 10 genes appearing as accessory in both lineages' accessory genomes. These findings match the results from our original combined pangenome covering both lineages, as the same 2152 core genes and 10 accessory genes were consistently annotated.

A small subset of genes (**Dataset S7**) appears as core of one lineage-specific pangenome and the accessory in the other (16 accessory genes in BD-1.2 are core genes in BD-2, and 17 accessory genes in BD-2 are core genes in BD-1.2). This discrepancy is directly linked to the challenges inherent in conducting separate lineage-specific pangenomes, as discussed previously (point ii). Nevertheless, these 33 genes are all categorized as accessory in our single combined pangenome and were used in original pipeline's statistical comparison of their presence/absence.

A total of 155 genes (44 in BD-1.2 and 111 in BD-2) were found exclusively in one lineage in the separate pangenomes (**Dataset S7**), compared to 137 (34 in BD-1.2 and 103 in BD-2) genes exclusive to a single lineage in the single pangenome pipeline. This discrepancy in gene count (i.e. the additional genes found exclusively in one lineage using separate pangenomes) arises from annotation ambiguities (see point (i) above) where variants of the same gene are named differently, thus considered distinct in lineage-specific pangenomes (see **Dataset S8**). Hence, once the annotation ambiguities are resolved from the lineage

specific pangenomes, the accessory gene set found to be exclusively present in only a single lineage in the separate pangenome analysis was identical to that found in the combined single pipeline and used as input into the statistical analysis.

In summary, in this specific study (i.e. comparing the core and accessory genome sequences across two closely related lineages) using both a combined pangenome and the lineage-specific pangenome approach yielded identical results, with the same genes considered as core and accessory, confirming that a combined pangenome approach is not bring unduly biased by population structure.

In addition, we compared the BD-1.2 and BD-2 isolates in our cohort with publicly available BD-1.2 and BD-2 sequencing data available to date, by either constructing separate pangenomes for each lineage or a combined pangenome

Construction of an extended pangenome including both our cohort and publicly available data.

To ensure that our cohort of 129 isolates was representative of the BD-1.2 and BD-2 lineages as a whole, we repeated the analysis integrating our cohort to the publicly available BD-1.2 and BD-2 sequences available to date (listed in **Dataset S2**), used in the phylogenetic analysis shown in **Figure S3**, and also used in Monir *et al* 2023 (Monir *et al* 2023). This gave a total of 256 isolates as input for an expanded pangenome (106 BD-1.2 and 150 BD-2 isolates) for an expanded pangenome.

In this extended (256 isolates) pangenome there are 3362 core genes of which 2148 are annotated, compared to 2183 annotated core genes in the original pangenome (129 isolates), **Dataset S9**. Comparing between these two pangenomes we have an overlap of 2135 core genes, with only 13 additional core genes present in the extended pangenome. Four of these 13 genes were present as accessory genes in the original pangenome, and the other 9 (*rluA*, *prpF*, *dgcM*, *epsE_2*, *bcr*, *dinG*, *ddl_2*, *dgcT*, *mcpQ_2*) but named differently due to annotation ambiguities as discussed above. Comparing accessory genes, there are 156 annotated accessory genes in the original pangenome, compared to 316 annotated accessory genes in the extended pangenome as would be expected with a greater number of isolates, with an overlap of 150 genes. There were 166 additional accessory genes in the extended pangenome but only 3 of these (*hly_2*, *cat_1* and *luxO_1*) were found in significantly different distributions of BD-1.2 and BD-2 sequences, when subjected to the same statistical pipeline as employed for the original pangenome. The gene *hly_2* was also found to be statistically significant in the original pipeline but was named *hlyA_1*. The genes *cat_1* and *luxO_1* were present in all 129 of the isolates in the original pipeline so was considered as a core gene and analysed for significant mutations, in the extended pangenome these genes were not present in 18% of BD-1.2 isolates and 17% of BD-2 isolates respectively. In total 14 annotated accessory genes from the extended pangenome were statistically significantly discriminating lineages of these 11 were found in the original analysis, 2 (*cat_1* and *luxO_1*) were instead analysed as core genes, as described above, and one, *ptsG_1* (a glucose transporter) did not meet the p-value threshold in the original pangenome analysis but was found to be statistically significant in the extended pangenome.

Overall, this analysis suggests that the original cohort of 129 isolates were largely representative of the BD-1.2 and BD-2 lineages and so the pangenome analysis to statistically compare across lineages is a valid representation of the broader behaviour of these lineages.

A single combined pangenome of 129 isolates in our cohort is representative of the wider separate BD-1.2 and BD-2 lineages

Finally, we considered how the original pipeline compared to separated lineage-specific pangenomes using the extended cohort. Considering the number of overlapping core and

accessory genes in the expanded lineage-specific pangenomes (i.e. a pangenome of 106 BD-1.2 isolates and a separate pangenome of 150 BD-2 isolates), the same 2133 genes are present in both the BD-1.2 and the BD-2 core genomes, with 12 genes appearing as accessory in both lineages' accessory genome (**Dataset S10**). These genes were also similarly found as core and accessory genes within our original cohort, except for a small number of annotation ambiguities as described above.

In total, 302 genes unique to one lineage and 43 genes core in one lineage but accessory in the other were analysed similarly as before. Of these genes, 133 were also found to be exclusively present in the analysis of our cohort alone and were tested for significance in **Dataset S4**. A further 136 genes were found to be exclusively present in one lineage (either BD-1.2 or BD-2) in this expanded cohort of our isolates plus publicly available sequences, however all of these were present in less than 6 isolates and were not statistically significant. Of the remaining 76 genes were the results of annotation ambiguities associated with constructing separate lineage-specific genomes. Sixty-five of these were found to be core genes in the expanded pangenomes once annotation was manually corrected and were also considered as core genes within the analysis of our cohort. Similarly, 11 genes were found to be accessory genes in the expanded pangenomes after correction and were also found as accessory genes our cohort. Of note, the gene *cph2* (a phytochrome-like protein) in the expanded pangenome, whilst correctly considered as an accessory gene as identified as separating lineages compared to our initial cohort-only analysis, was not annotated in our smaller cohort. This gene had been previously reported to differentiate BD-1 and BD-2 lineages (Monir *et al.*, 2022), which our cohort-only analysis had missed due to the lack of annotation.

This analysis confirms that our cohort is broadly representative of all available BD-1.2 and BD-2 genome sequences available to date, and further confirms that the pipeline using a combined genome, gives better results than analysing independent pangenomes which results in many false positives due to annotation ambiguities.”

Supplementary Datasets S6-S10:

Dataset S6. Summary of core, accessory, and unannotated gene counts in the pangenomes for the 84 BD-1.2 isolates and the 45 BD-2 isolates in our cohort.

Dataset S7. Summary of overlapping and non-overlapping genes in the core and accessory sets of the pangenomes for the 84 BD-1.2 isolates and the 45 BD-2 isolates in our cohort.

Dataset S8. Genes found in the separate lineage specific pangenomes of either BD-1.2, BD-2 showing the annotation problems and how these genes map to the initial cohort-only analysis.

Dataset S9. Summary of core, accessory, and unannotated gene counts in the pangenomes for 106 BD-1.2 isolates (84 from our cohort and 22 publicly available genomes, see Dataset 2) and 150 BD-2 isolates (45 from our cohort and 105 publicly available genomes, see Dataset 2) and the combined pangenome of all our and public isolates for BD-1.2 and BD-2 (256 isolates).

Dataset S10. Summary of overlapping and non-overlapping genes in the core and accessory sets of the pangenomes for 106 BD-1.2 isolates (84 from our cohort and 22 publicly available genomes, see Dataset 2) and 150 BD-2 isolates (45 from our cohort and 105 publicly available genomes, see Dataset 2).

Horesh, Gal, Alyce Taylor-Brown, Stephanie McGimpsey, Florent Lassalle, Jukka Corander, Eva Heinz, and Nicholas R. Thomson. 2021. 'Different evolutionary trends form the twilight zone of the bacterial pan-genome', *Microbial Genomics*, 7.

Monir, Md Mamun, Talal Hossain, Masatomo Morita, Makoto Ohnishi, Fatema-Tuz Johura, Marzia Sultana, Shirajum Monira, Tahmeed Ahmed, Nicholas Thomson, and Haruo Watanabe. 2022. 'Genomic Characteristics of Recently Recognized *Vibrio cholerae* El Tor Lineages Associated with Cholera in Bangladesh, 1991 to 2017', *Microbiology Spectrum*, 10: e00391-22.

Monir, Md Mamun, Mohammad Tarequl Islam, Razib Mazumder, Dinesh Mondal, Kazi Sumaita Nahar, Marzia Sultana, Masatomo Morita, Makoto Ohnishi, Anwar Huq, Haruo Watanabe, Firdausi Qadri, Mustafizur Rahman, Nicholas Thomson, Kimberley Seed, Rita R. Colwell, Tahmeed Ahmed, and Munirul Alam. 2023. 'Genomic attributes of *Vibrio cholerae* O1 responsible for 2022 massive cholera outbreak in Bangladesh', *Nature Communications*, 14: 1154.

Page, A. J., C. A. Cummins, M. Hunt, V. K. Wong, S. Reuter, M. T. Holden, M. Fookes, D. Falush, J. A. Keane, and J. Parkhill. 2015. 'Roary: rapid large-scale prokaryote pan genome analysis', *Bioinformatics*, 31: 3691-3.

2. Authors should consider validating whether the *in silico* data translate into biological differences. The fact that the isolates are available is a great asset. Of course not every phenotype can be tested but the following ones can be done as follows:

(i) Effect on antibiotic resistant pattern. I suggest authors should test a broad panel of drugs including detergents, for example.

We thank the Reviewer for their feedback on our manuscript regarding the effect on antibiotic-resistant patterns in *V. cholerae*. In response to the Reviewer's query regarding testing a broad panel of drugs, including detergents, we would like to address the rationale behind our experimental design and clarify why we believe this request may not align with the scope of our study.

Firstly, we note the Reviewer's suggestion to test a broad panel of drugs, including detergents. We acknowledge the importance of comprehensive antimicrobial testing, and in our study, we have indeed evaluated a substantial range of antimicrobial agents. The *V. cholerae* isolates used in this study were obtained from Bangladesh's national cholera and antimicrobial resistance surveillance, conducted by the two national authorized centres, icddr,b and IECDR, making use of the governmental approved protocols. Under this national study protocol, these centres regularly monitor the resistance pattern against 8 antibiotics (ampicillin, azithromycin, ciprofloxacin, ceftriaxone, cefixime, doxycycline, erythromycin, meropenem) and this data was provided in Table S1 (now Dataset S1). The selection includes antibiotics that are used for diarrheal cases in Bangladesh, although changes/preferences depend on current antibiotic susceptibility pattern. These antibiotics were chosen based on the treatment protocols recommended by health authorities in Bangladesh for managing acute watery diarrhoea (e.g. cholera and other bacterial diarrheal diseases). Azithromycin and doxycycline, for instance, are currently the first choice of treatment for cholera in both adults and children (some other antibiotics are not suitable for pregnant women or young children). Hence, we do emphasize that these are the most appropriate antimicrobials used by the health authorities.

However, we must address the request to test detergents, as it appears to deviate from established practices in our field or in the health field according to the UK or Bangladeshi health authorities. In Bangladesh, as in much of the scientific literature, testing detergents against *Vibrio cholerae* is not a common practice. A review of existing literature reveals that there are over 500 studies focusing on antimicrobial drug testing in *Vibrio cholerae*, only one has included detergents in their analyses (Najim 2017). Furthermore, the inclusion of detergents is not directly relevant to our study's objectives, which primarily focus on genomic traits *V. cholerae* driving lineage transmission and disease severity.

Our paper is not primarily on antimicrobial resistance. We did investigate for resistance to better understand the lineage transmission dynamics, but our results do not unravel any novelties compared to recent work by (Monir *et al.* 2023), which we have referenced extensively. Any further investigation on this matter would be out of scope given our research objectives. Notably, neither our study nor Monir *et al.* conducted specific analyses on detergents or additional drugs beyond those traditionally tested for antimicrobial susceptibility.

In summary, while we appreciate the reviewer's suggestion for a broader panel of drug testing, including detergents, we believe that such an approach would not be warranted within the context of our study.

Monir, Md Mamun, Mohammad Tarequl Islam, Razib Mazumder, Dinesh Mondal, Kazi Sumaita Nahar, Marzia Sultana, Masatomo Morita, Makoto Ohnishi, Anwar Huq, Haruo Watanabe, Firdausi Qadri, Mustafizur Rahman, Nicholas Thomson, Kimberley Seed, Rita R. Colwell, Tahmeed Ahmed, and Munirul Alam. 2023. 'Genomic attributes of *Vibrio cholerae* O1 responsible for 2022 massive cholera outbreak in Bangladesh', Nature Communications, 14: 1154.

Najim, Shaymaa Suhail. 2017. 'The Efficiency of Dettol as Detergent against Microbial Biofilm formation isolated from UTI infections', Current Research in Microbiology and Biotechnology, 5: 1380-84.

(ii) Metabolic profile. Authors should consider a Biolog-based approach informed by their bioinformatics findings to define the metabolic capabilities of the strains.

We thank the Reviewer for this comment. However, we did not need a Biolog-based approach to define the metabolic capabilities of the strain as we relied on an already validated GSM model established in the literature: iAM-Vc960. Such GSM model was extensively tested, underwent peer review and was published in mSystems (Abdel-Haleem *et al.* 2020). In the construction and curation stage of such model, the authors used a standard protocol described in Thiele and Palsson (2010), and widely accepted as the gold standard, with 1962 citations. In brief, the authors used another previously validated GSM iJO1366 (Orth *et al.* 2011) as a starting point for the reconstruction (this model is itself validated experimentally, it is present in the BiGG database (Norsigian *et al.* 2020) and has over 1200 citations). Reactions were added/removed based on evidence for their presence in literature or evidence from the authors transcriptomics analysis. The biomass function was then built based on iJO1366 (Orth *et al.* 2011) and a closely related *V. vulnificus* GSMM (Kim *et al.* 2011), itself also validated with experimental data. Experimental transcriptomics data was then used to further refine the iAM-Vc960 model. Then, simulated growth rates were compared to experimental growth rates obtained from a literature search. In addition, predicted gene essentiality was compared to both a high confidence set of 233 genes from previous studies and additionally to gene essentiality data for 3602 *V. cholerae* C6706 (a closely related strain) genes, obtained from the OGEE database (Chen *et al.* 2012). After these validation steps the authors concluded that the agreement between the experimental and predicted data validated the reconstruction (Abdel-Haleem *et al.* 2020).

To further assess whether a Biolog-based approach would be necessary to further support the GSM model, we conducted a literature search. Focussing on the BiGG database (King *et al.* 2016; Norsigian *et al.* 2020; Schellenberger *et al.* 2010) which holds high-quality manually curated models (Fang, Lloyd, and Palsson 2020), we evaluated 70 models present in this database. Of these, only four confirmed the metabolic reactions of the strain using a Biolog-based approach, iML1515 (Monk *et al.* 2017), iYL1228 (Liao *et al.* 2011), STM_v1_0 (Thiele *et al.* 2011) and iAF1260 (Feist *et al.* 2007). The validation methods amongst all the models varied with transcriptomics data, gene essentiality data, growth rate comparisons and radio

labelling experiments all used in various combinations (Feist *et al.* 2007; Feist *et al.* 2014; Abdel-Haleem *et al.* 2018; Monk *et al.* 2013; Monk *et al.* 2016; Nagarajan *et al.* 2013; Kavvas, Seif, *et al.* 2018; Shiratsubaki *et al.* 2020; Thiele *et al.* 2005; Broddrick *et al.* 2016; Nogales *et al.* 2012; Nogales, Palsson, and Thiele 2008; Orth *et al.* 2011). Importantly 51% (n=36) of the 70 models did not have any experimental validation: iAF1260b, iJN1463, iAF692, iAM_Pb448, iAM_Pc455, iAM_Pk459, iAM_Pv461, iAPECO1_1312, iB21_1397, iBWG_1329, iE2348C_1286, iEC55989_1330, iECABU_c1320, iECB_1328, iECBD_1354, iECD_1391, iECDH10B_1368, iEcE24377_1341, iECED1_1282, iECH74115_1262, iEcHS_1320, iECIAI1_1343, iECIAI39_1322, iECNA114_1301, iECO103_1326, iECO111_1330, iECO26_1355, iECOK1_1307, iECP_1309, iECS88_1305, iECSE_1348, iECSF_1327, iECSP_1301, iECW_1372, iEKO11_1354, iETEC_1333, and iG2583_1286 (Feist *et al.* 2010; Monk *et al.* 2013; Feist *et al.* 2006; Abdel-Haleem *et al.* 2018). Despite this lack of validation, all the considered models were assessed of acceptable quality for entering the BiGG database (Norsigian *et al.* 2020).

Finally, it is important to point out once more that in our work we were not generating a GSM model, we were instead using an already experimentally validated model as an analysis tool, as we have previously done in other works published in Nature communications (Baker *et al.* 2024) and mSystems (Percy *et al.* 2021), without further confirmation of the metabolic capabilities being deemed necessary by reviewers. Further to this, we would like to emphasise that the other reviewers of this paper (Reviewer 1, 3, and 4), of whom two (3 and 4) appear to be GSM experts, also did not request metabolic validation and appreciated our approach in using GSM modelling. Specifically, Reviewer 4 states “I particularly like their approach of functionally interpreting the relevance of genomic variants in the different isolates using genome-scale metabolic modelling and structural analysis.”. To draw a wider comparison, we considered other published works that used validated GSM models in a similar way and found that these also did not perform additional metabolic testing of the previously published models (Pacheco, Moel, and Segrè 2019; Heinken and Thiele 2015; Cesur *et al.* 2020). Given the above, we believe that metabolic testing of strains is out of the scope of our study, which aimed develop a computational framework to identify *V. cholerae* signatures of genomic traits linked to lineage transmission dynamics and disease severity. Instead, such experimental work would be better suited to papers developing novel genome scale metabolic reconstructions.

Abdel-Haleem, A. M., H. Hefzi, K. Mineta, X. Gao, T. Gojobori, B. O. Palsson, N. E. Lewis, and N. Jamshidi. 2018. 'Functional interrogation of Plasmodium genus metabolism identifies species- and stage-specific differences in nutrient essentiality and drug targeting', PLoS Computational Biology, 14: e1005895.

Abdel-Haleem, Alyaa M, Vaishnavi Ravikumar, Boyang Ji, Katsuhiko Mineta, Xin Gao, Jens Nielsen, Takashi Gojobori, and Ivan Mijakovic. 2020. 'Integrated metabolic modeling, culturing, and transcriptomics explain enhanced virulence of *Vibrio cholerae* during coinfection with enterotoxigenic *Escherichia coli*', mSystems, 5: e00491-20.

Baker, Michelle, Xibin Zhang, Alexandre Maciel-Guerra, Kubra Babaarslan, Yinping Dong, Wei Wang, Yujie Hu, David Renney, Longhai Liu, and Hui Li. 2024. 'Convergence of resistance and evolutionary responses in *Escherichia coli* and *Salmonella enterica* co-inhabiting chicken farms in China', Nature Communications, 15: 206.

Broddrick, J. T., B. E. Rubin, D. G. Welkie, N. Du, N. Mih, S. Diamond, J. J. Lee, S. S. Golden, and B. O. Palsson. 2016. 'Unique attributes of cyanobacterial metabolism revealed by

improved genome-scale metabolic modeling and essential gene analysis', Proceedings of the National Academy of Sciences of the United States of America, 113: E8344-e53.

Cesur, Müberra Fatma, Bushra Siraj, Reaz Uddin, Saliha Durmuş, and Tunahan Çakır. 2020. 'Network-Based Metabolism-Centered Screening of Potential Drug Targets in *Klebsiella pneumoniae* at Genome Scale', Frontiers in Cellular and Infection Microbiology, 9.

Chen, W. H., P. Minguéz, M. J. Lercher, and P. Bork. 2012. 'OGEE: an online gene essentiality database', Nucleic Acids Research, 40: D901-6.

Fang, Xin, Colton J. Lloyd, and Bernhard O. Palsson. 2020. 'Reconstructing organisms in silico: genome-scale models and their emerging applications', Nature Reviews Microbiology, 18: 731-43.

Feist, A. M., C. S. Henry, J. L. Reed, M. Krummenacker, A. R. Joyce, P. D. Karp, L. J. Broadbelt, V. Hatzimanikatis, and BØ Palsson. 2007. 'A genome-scale metabolic reconstruction for *Escherichia coli* K-12 MG1655 that accounts for 1260 ORFs and thermodynamic information', Molecular Systems Biology, 3: 121.

Feist, A. M., H. Nagarajan, A. E. Rotaru, P. L. Tremblay, T. Zhang, K. P. Nevin, D. R. Lovley, and K. Zengler. 2014. 'Constraint-based modeling of carbon fixation and the energetics of electron transfer in *Geobacter metallireducens*', PLoS Computational Biology, 10: e1003575.

Feist, A. M., J. C. Scholten, BØ Palsson, F. J. Brockman, and T. Ideker. 2006. 'Modeling methanogenesis with a genome-scale metabolic reconstruction of *Methanosarcina barkeri*', Molecular Systems Biology, 2: 2006.0004.

Feist, A. M., D. C. Zielinski, J. D. Orth, J. Schellenberger, M. J. Herrgard, and BØ Palsson. 2010. 'Model-driven evaluation of the production potential for growth-coupled products of *Escherichia coli*', Metab Eng, 12: 173-86.

Heinken, Almut, and Ines Thiele. 2015. 'Anoxic Conditions Promote Species-Specific Mutualism between Gut Microbes *In Silico*', Applied and Environmental Microbiology, 81: 4049-61.

Kavvas, E. S., Y. Seif, J. T. Yurkovich, C. Norsigian, S. Poudel, W. W. Greenwald, S. Ghatak, B. O. Palsson, and J. M. Monk. 2018. 'Updated and standardized genome-scale reconstruction of *Mycobacterium tuberculosis* H37Rv, iEK1011, simulates flux states indicative of physiological conditions', BMC Systems Biology, 12: 25.

Kim, Hyun Uk, Soo Young Kim, Haeyoung Jeong, Tae Yong Kim, Jae Jong Kim, Hyon E Choy, Kyu Yang Yi, Joon Haeng Rhee, and Sang Yup Lee. 2011. 'Integrative genome-scale metabolic analysis of *Vibrio vulnificus* for drug targeting and discovery', Molecular Systems Biology, 7: 460.

King, Z. A., J. Lu, A. Dräger, P. Miller, S. Federowicz, J. A. Lerman, A. Ebrahim, B. O. Palsson, and N. E. Lewis. 2016. 'BiGG Models: A platform for integrating, standardizing and sharing genome-scale models', Nucleic Acids Research, 44: D515-22.

Liao, Y. C., T. W. Huang, F. C. Chen, P. Charusanti, J. S. Hong, H. Y. Chang, S. F. Tsai, B. O. Palsson, and C. A. Hsiung. 2011. 'An experimentally validated genome-scale metabolic reconstruction of *Klebsiella pneumoniae* MGH 78578, iYL1228', Journal of Bacteriology, 193: 1710-7.

Monk, J. M., P. Charusanti, R. K. Aziz, J. A. Lerman, N. Premyodhin, J. D. Orth, A. M. Feist, and BØ Palsson. 2013. 'Genome-scale metabolic reconstructions of multiple *Escherichia coli* strains highlight strain-specific adaptations to nutritional environments', Proceedings of the National Academy of Sciences of the United States of America, 110: 20338-43.

Monk, J. M., A. Koza, M. A. Campodonico, D. Machado, J. M. Seoane, B. O. Palsson, M. J. Herrgård, and A. M. Feist. 2016. 'Multi-omics Quantification of Species Variation of *Escherichia coli* Links Molecular Features with Strain Phenotypes', *Cell Syst*, 3: 238-51.e12.

Monk, J. M., C. J. Lloyd, E. Brunk, N. Mih, A. Sastry, Z. King, R. Takeuchi, W. Nomura, Z. Zhang, H. Mori, A. M. Feist, and B. O. Palsson. 2017. 'iML1515, a knowledgebase that computes *Escherichia coli* traits', *Nature Biotechnology*, 35: 904-08.

Nagarajan, H., M. Sahin, J. Nogales, H. Latif, D. R. Lovley, A. Ebrahim, and K. Zengler. 2013. 'Characterizing acetogenic metabolism using a genome-scale metabolic reconstruction of *Clostridium ljungdahlii*', *Microb Cell Fact*, 12: 118.

Nogales, J., S. Gudmundsson, E. M. Knight, B. O. Palsson, and I. Thiele. 2012. 'Detailing the optimality of photosynthesis in cyanobacteria through systems biology analysis', *Proceedings of the National Academy of Sciences of the United States of America*, 109: 2678-83.

Nogales, J., BØ Palsson, and I. Thiele. 2008. 'A genome-scale metabolic reconstruction of *Pseudomonas putida* KT2440: iJN746 as a cell factory', *BMC Systems Biology*, 2: 79.

Norsigian, Charles J, Neha Pusarla, John Luke McConn, James T Yurkovich, Andreas Dräger, Bernhard O Palsson, and Zachary King. 2020. 'BiGG Models 2020: multi-strain genome-scale models and expansion across the phylogenetic tree', *Nucleic Acids Research*, 48: D402-D06.

Orth, J. D., T. M. Conrad, J. Na, J. A. Lerman, H. Nam, A. M. Feist, and BØ Palsson. 2011a. 'A comprehensive genome-scale reconstruction of *Escherichia coli* metabolism--2011', *Molecular Systems Biology*, 7: 535.

Pacheco, Alan R., Mauricio Moel, and Daniel Segrè. 2019. 'Costless metabolic secretions as drivers of interspecies interactions in microbial ecosystems', *Nature Communications*, 10: 103.

Pearcy, Nicole, Yue Hu, Michelle Baker, Alexandre Maciel-Guerra, Ning Xue, Wei Wang, Jasmeet Kaler, Zixin Peng, Fengqin Li, Tania Dottorini, and Xiaoxia Lin. 2021. 'Genome-scale metabolic models and machine Learning reveal genetic determinants of antibiotic resistance in *Escherichia coli* and unravel the underlying metabolic adaptation mechanisms', *mSystems*, 6: e00913-20.

Schellenberger, Jan, Junyoung O Park, Tom M Conrad, and Bernhard Ø Palsson. 2010. 'BiGG: a Biochemical Genetic and Genomic knowledgebase of large-scale metabolic reconstructions', *BMC Bioinformatics*, 11: 1-10.

Shiratsubaki, I. S., X. Fang, R. O. O. Souza, B. O. Palsson, A. M. Silber, and J. L. Siqueira-Neto. 2020. 'Genome-scale metabolic models highlight stage-specific differences in essential metabolic pathways in *Trypanosoma cruzi*', *PLoS Neglected Tropical Diseases*, 14: e0008728.

Thiele, I., T. D. Vo, N. D. Price, and BØ Palsson. 2005. 'Expanded metabolic reconstruction of *Helicobacter pylori* (iIT341 GSM/GPR): an in-silico genome-scale characterization of single- and double-deletion mutants', *Journal of Bacteriology*, 187: 5818-30.

Thiele, Ines, Daniel R Hyduke, Benjamin Steeb, Guy Fankam, Douglas K Allen, Susanna Bazzani, Pep Charusanti, Feng-Chi Chen, Ronan MT Fleming, and Chao A Hsiung. 2011. 'A community effort towards a knowledge-base and mathematical model of the human pathogen *Salmonella Typhimurium* LT2', *BMC Systems Biology*, 5: 1-9.

Thiele, Ines, and Bernhard Ø Palsson. 2010. 'A protocol for generating a high-quality genome-scale metabolic reconstruction', *Nature Protocols*, 5: 93-121.

(iii) Authors should consider simple genetic complementation experiments to validate whether some of the most relevant alleles conferring a biological phenotype are necessary and

sufficient. This can be done with available tools (for example Tn7 system). For example, authors may choose to introduce the allele into a strain lacking it, and/or assess the function of the introduced allele in a mutant background for the allele.

We thank the Reviewer for this suggestion. As a result, we carefully considered whether the simple genetic complementation experiments suggested by the Reviewer could be used to validate the correlations between alleles and the clinical symptoms in our results.

Unfortunately, there are significant technical difficulties in using Tn7 system experiments for validation, as outlined below. The proposed approach cannot capture the multifactorial nature of the genetic mechanisms underlying the C_k^n combinations for each of the six symptoms prediction that we developed. However, before addressing these technical issues, we must discuss an even more serious issue related to the Reviewer's suggested validation experiments, which makes their request inapplicable. Regardless of whether it would be possible or not to introduce the allele into a strain lacking it, and/or assess the function of the introduced allele in a mutant background for the allele, no form of genetic complementation experiment would be testable to link changes in the alleles to the manifestation of symptoms (vomit, abdominal pain, duration of diarrhoea, number of stools and dehydration). Since clinical symptoms are a result of pathogen-human host interaction, the only way to validate our results, which correlate allele variants to measurable clinical symptoms, would be by testing the Tn7-modified strains in an *in vivo* human model. Since human trials are clearly not possible or ethical, we would only be able to test the clinical manifestations of infections with the genetically modified *V. cholerae* strains in animal models, of which there are only two viable ones (for *V. cholerae*): infant mice or infant rabbits (Ritchie *et al.* 2010). Neither is suitable to validate our work. Specifically, *V. cholerae* readily colonizes the intestine of suckling mice, but they do not develop overt diarrhoea or other cholera symptoms (Ritchie *et al.* 2010; Walton *et al.* 2023). Thus this model is only useful to identify gene products that promote colonisation and not for factors underlying cholera pathology (Ritchie *et al.* 2010; Walton *et al.* 2023). In the infant rabbit, which has shown to be promising in the study of *V. cholerae* progression following oral infection (Ritchie *et al.* 2010), the model results in 100% lethality within a short-time period (24 hours) and requires the measurement of caecal fluid at necropsy, as diarrheal fluid loss is not measurable (Ritchie *et al.* 2010). Furthermore, standard protocols for this model require neutralising stomach acid prior to inoculation with the pathogen (Abel and Waldor 2015), another major difference with natural human infection which could severely alter symptom manifestation. Hence, again, this model would not be suitable to study factors underlying the different signs underlying cholera symptoms in humans (vomit, abdominal pain, duration of diarrhoea, number of stools and dehydration) where the mortality is lower and diarrheal fluid loss is a major symptom.

Due to *V. cholerae* not naturally colonizing in the intestines of adult mammals other than humans (Ritchie *et al.* 2010) there are no other animal models of cholera (Ritchie *et al.* 2010, Calvignoni *et al.* 2020). As explained in the methods lines 792-807, some symptoms in our work have an associated degree of severity defined on a semi-continuous scale (number of stools and duration of diarrhoea), others have a categorical classification (dehydration) and others are assessed in terms of their presence/absence only (vomit and abdominal pain). However, both the mice and rabbit models are incapable of replicating human cholera symptoms.

Hence, to summarise, whilst there are technical issues with genetic complementation experiments as explained below, we would like to emphasise that even if these experiments were to be conducted, no meaningful validation results would be achieved due to a lack of suitable *in vivo* models to replicate symptomatic response to infections in humans.

Regarding the technical problems with genetic complementation approaches, our study indicates that the clinical phenotypes are determined by a combination of genetic elements, and not by a single gene. Although specific genetic complementation tools as the one cited by the Reviewer can cope with multi-allele testing, there is still a limitation on the number even in

the hands of the developers of these systems (Barth *et al.* 1976, Lichtenstein *et al.* 1982, McKown *et al.* 1988). For our study specifically, the observed complexity means that simple complementation experiments will not capture the multifactorial nature of the genetic mechanisms underlying the biological phenotype. Identifying and introducing all relevant alleles into a model system, and then accurately assessing their combined effects, would be technically challenging.

To assess the scale of the problem, we considered the features (genetic elements) found as most strongly correlated with the observed clinical symptoms, by the machine learning analysis. Recall that in our research, we created machine learning symptom prediction models dedicated to each symptom for which the available number of samples was adequate. As illustrated in the manuscript, we were able to obtain six symptom prediction models, namely: abdominal pain presence/absence; dehydration “moderate” vs “severe”; duration of diarrhoea <1 day vs. 1-3 days; number of stools 11-15 times vs. 16-20 times; number of stools 11-15 times vs. 21+ times; and vomit presence/absence. For each one a subset of genetic elements (features) was identified as being the most strongly correlated to the prediction outcome.

To address the Reviewer’s request, we considered if there would be a subset of genetic elements which could potentially be used for biological testing with a Tn7 system. Further, in case such subset would be found, which and how many genetic elements belonging to the subset would also be amongst those selected by the machine learning prediction models as the most strongly correlated to clinical symptoms. This is a classical combinatorial problem of selecting a subset of items from a collection where the order of the selection does not matter. For example, suppose we have a set of 3 genes A, B and C, then in how many unique combinations could two genes be selected from these three genes? In our specific case, for the six symptoms prediction models we had the following number of features: abdominal pain presence/absence (24 features); dehydration “moderate” vs “severe” (30 features); duration of diarrhoea <1 day vs. 1-3 days (34 features); number of stools 11-15 times vs. 16-20 times (33 features); number of stools 11-15 times vs. 21+ times (33 features); and vomit presence/absence (42 features). More formally, a k -combination of a set S is a subset of k distinct elements of S . If the set S has n elements, the number of k -combinations can be written as:

$$C_k^n = \frac{n!}{k!(n-k)!}, \text{ when } n > k$$

Figure 1 below shows the total number C_k^n of combinations we need for each one of the six symptom prediction models. In the figure, the x-axis represents the number k of elements being chosen. The figure indicates that when k increases C_k^n quickly increases to very large numbers, which would be unfeasible to test during any reasonable time frame and to test all the combinations of alleles into a strain. Moreover, the number of complementation tests we would need to perform has never been attempted so far in *Vibrio cholerae* (Santoriello *et al.* 2020, Cheng, Ottemann, and Yildiz 2015). Finally, although the suggested analysis is unfeasible for the two major reasons we described above, to perform these analyses would require an entire paper in itself to describe the methodology and results, as other authors have done in the literature (Van der Henst *et al.* 2016; Santoriello *et al.* 2020; Cheng, Ottemann, and Yildiz 2015). This was not our aim. Instead, we employed a computational method to investigate the associations between genomic traits and phenotypes, similar to other studies published in Nature Communications (Kavvas *et al.* 2018, Mageiros *et al.* 2021, Green *et al.* 2022, Nguyen *et al.* 2022, Wu *et al.* 2022, Monir *et al.*, 2023), our focus was on exploring these interactions computationally.

Figure 1. Number of possible combinations of features to choose from based on the number of features selected for each clinical symptom prediction model. Y-axis is presented on a log10 scale.

Finally, there are additional specific technical challenges associated with the Tn7 system (Chao *et al.* 2016). The Tn7 transposon system targets specific sites in the genome (typically Tn7 sites). If these sites are not present, or they are in regions that do not allow for proper expression of the inserted genes, the experiment may fail (Choi and Schweizer 2006). Ensuring that the inserted alleles are expressed at levels that mimic the native context can be difficult and overexpression or under expression could lead to misleading results about the role of the allele in the phenotype. Therefore, ensuring that the introduced genes are regulated and expressed in a manner consistent with their native context can be challenging, which might lead to non-physiological results that again in our case given we are targeting clinical symptoms is relevant. Furthermore, the size of the genetic element that can be introduced may also be constrained by the transposon capacity, which might be a limitation if the genetic element is large or if multiple genes need to be introduced (Chao *et al.* 2016), as is the case in our work where multiple genes contribute to the biological phenotype.

Abel, Sören, and Matthew K. Waldor. 2015. 'Infant Rabbit Model for Diarrheal Diseases', *Current Protocols in Microbiology*, 38: 6A.6.1-6A.6.15.

Barth, P. T., *et al.* "Transposition of a deoxyribonucleic acid sequence encoding trimethoprim and streptomycin resistances from R483 to other replicons." *Journal of Bacteriology* 125.3 (1976): 800-810.

Calvignoni, M., Mazzantini, D., Celandroni, F. and Ghelardi, E., 2023. Animal and in vitro models as powerful tools to decipher the effects of enteric pathogens on the human gut microbiota. *Microorganisms*, 12(1), p.67.

Chao, M. C., S. Abel, B. M. Davis, and M. K. Waldor. 2016. 'The design and analysis of transposon insertion sequencing experiments', *Nat Rev Microbiol*, 14: 119-28.

Cheng, A. T., K. M. Ottemann, and F. H. Yildiz. 2015. '*Vibrio cholerae* Response Regulator VxrB Controls Colonization and Regulates the Type VI Secretion System', *PLoS Pathogens*, 11: e1004933.

Choi, Kyoung-Hee, and Herbert P. Schweizer. 2006. 'mini-Tn7 insertion in bacteria with single attTn7 sites: example *Pseudomonas aeruginosa*', *Nature Protocols*, 1: 153-61.

Green, A.G., Yoon, C.H., Chen, M.L. *et al.* A convolutional neural network highlights mutations relevant to antimicrobial resistance in *Mycobacterium tuberculosis*. *Nat Commun* 13, 3817 (2022). <https://doi.org/10.1038/s41467-022-31236-0>

Kavvas, E.S., Catoi, E., Mih, N. *et al.* Machine learning and structural analysis of *Mycobacterium tuberculosis* pan-genome identifies genetic signatures of antibiotic resistance. *Nat Commun* 9, 4306 (2018). <https://doi.org/10.1038/s41467-018-06634-y>

Lichtenstein, C., Brenner, S. Unique insertion site of Tn7 in the *E. coli* chromosome. *Nature* 297, 601–603 (1982). <https://doi.org/10.1038/297601a0>

Mageiros, L., Méric, G., Bayliss, S.C. *et al.* Genome evolution and the emergence of pathogenicity in avian *Escherichia coli*. *Nat Commun* 12, 765 (2021). <https://doi.org/10.1038/s41467-021-20988-w>

McKenzie, Gregory J., and Nancy L. Craig. 2006. 'Fast, easy and efficient: site-specific insertion of transgenes into Enterobacterial chromosomes using Tn7 without need for selection of the insertion event', *BMC Microbiology*, 6: 39.

McKown, R. L., *et al.* "Sequence requirements of *Escherichia coli* attTn7, a specific site of transposon Tn7 insertion." *Journal of bacteriology* 170.1 (1988): 352-358.

Monir, Md Mamun, Mohammad Tarequl Islam, Razib Mazumder, Dinesh Mondal, Kazi Sumaita Nahar, Marzia Sultana, Masatomo Morita, Makoto Ohnishi, Anwar Huq, Haruo Watanabe, Firdausi Qadri, Mustafizur Rahman, Nicholas Thomson, Kimberley Seed, Rita R. Colwell, Tahmeed Ahmed, and Munirul Alam. 2023. 'Genomic attributes of *Vibrio cholerae* O1 responsible for 2022 massive cholera outbreak in Bangladesh', *Nature Communications*, 14: 1154.

Nguyen, L., Van Hoeck, A. & Cuppen, E. Machine learning-based tissue of origin classification for cancer of unknown primary diagnostics using genome-wide mutation features. *Nat Commun* 13, 4013 (2022). <https://doi.org/10.1038/s41467-022-31666-w>

Ritchie, Jennifer M., Haopeng Rui, Roderick T. Bronson, and Matthew K. Waldor. 2010. 'Back to the Future: Studying Cholera Pathogenesis Using Infant Rabbits', *MBio*, 1: 10.1128/mbio.00047-10.0

Van der Henst, Charles, Tiziana Scignari, Catherine Maclachlan, and Melanie Blokesch. 2016. 'An intracellular replication niche for *Vibrio cholerae* in the amoeba *Acanthamoeba castellanii*', *The ISME Journal*, 10: 897-910.

Walton, Madison G, Isabella Cubillejo, Dhruvajyoti Nag, and Jeffrey H Withey. 2023. 'Advances in cholera research: from molecular biology to public health initiatives', *Frontiers in Microbiology*, 14: 1178538.

Wu, S., Feng, J., Liu, C. *et al.* Machine learning aided construction of the quorum sensing communication network for human gut microbiota. *Nat Commun* 13, 3079 (2022). <https://doi.org/10.1038/s41467-022-30741-6>

3. The metabolic pathway section does not clearly indicate whether indeed there are differences (or not) between the lineages. As currently written, it remains a catalogue of reactions.

We thank the Reviewer for this comment and agree that this part of the manuscript was poorly written. Using the generalised metabolic model iAM-VC960 it was not possible to conclusively assess differences in the metabolic pathways between lineages. We have now additionally added draft strain specific models of each of our genomes which now allow for this comparison, see Methods 884-891 and below. We have written this analysis in light of the

suggested strain specific models, highlighting where there are differences between the lineages in the Results Lines 258-267 and below.

Methods Lines 884-891: “All simulations were performed using the Python cobra toolkit v0.26.2. The analysis was conducted on both a manually curated and validated model of *V. cholerae* O1 N16961, iAM-Vc960, taken from Abdel-Haleem *et al.* 2020 (Abdel-Haleem *et al.* 2020) and on automatically generated draft strain specific GSM models. The strain specific draft models were generated using CarveMe (Machado *et al.* 2018). CarveMe was run using the CPLEX solver and gram negative template, with gap filling for LB and M9 media using the command: ‘carve input.faa --gapfill M9,LB -u gramneg --solver cplex --output model.xml’. Gene essentiality, FVA and FBA analyses as described below were conducted on genes of interest in the generalised iAM-Vc960 and in each of the 129 draft strain specific models, based on the analysis in pipeline in Percy *et al.* (Percy *et al.* 2021).”

Results Lines 258-267: “Gene essentiality analysis concurred with the general model (iAM-Vc960), with only a small number of differences (**Dataset S12**). The effect of *murl* gene knockouts varied between lineages, proving to be non-essential in 94% of BD-1.2 lineage models, but only non-essential in 76% of BD-2 lineage models. Flux variability analysis in the individual models showed that knockouts of the gene *clcA* led to significant flux span changes in most BD-2 models (96%), in the reaction CLt3_2pp, controlling chloride transport out of the cell. However, the same knockouts changed the flux span in only 5% of BD-1.2 models. The *clcA* gene has been linked to bacterial acid resistance and it has been suggested that changes to the expression/repression of this gene may help facilitate survival during movement through the intestinal tract (Cakar *et al.* 2018). Similarly, flux balance analysis indicated that metabolite yield was changed differently across lineages in response to knocking out *clcA*, with the metabolite yield of chloride reduced to 0 in 95% of BD-1.2 isolates.”

Abdel-Haleem, Alyaa M, Vaishnavi Ravikumar, Boyang Ji, Katsuhiko Mineta, Xin Gao, Jens Nielsen, Takashi Gojobori, and Ivan Mijakovic. 2020. 'Integrated metabolic modeling, culturing, and transcriptomics explain enhanced virulence of *Vibrio cholerae* during coinfection with enterotoxigenic *Escherichia coli*', mSystems, 5: e00491-20.

Percy, Nicole, Yue Hu, Michelle Baker, Alexandre Maciel-Guerra, Ning Xue, Wei Wang, Jasmeet Kaler, Zixin Peng, Fengqin Li, Tania Dottorini, and Xiaoxia Lin. 2021. 'Genome-scale metabolic models and machine Learning reveal genetic determinants of antibiotic resistance in *Escherichia coli* and unravel the underlying metabolic adaptation mechanisms', mSystems, 6: e00913-20.

4. The structural analysis section is well done although it lacks validation whether the changes observed do translate into changes of protein function/biology.

We thank the Reviewer for appreciating our structural analysis

As delineated in Lines 532-596, our working hypothesis posits that the observed structural alterations may not necessarily correlate directly with changes in protein function or biology. Rather, we contend that the observed additional bindings potentially contribute to the augmented stability of the protein structure, a phenomenon often associated with heightened binding affinity or enhanced interaction capacity with natural substrates.

We maintain that our hypothesis, emphasizing a bolstered interaction capacity with normal substrates, finds robust support in the results provided in the original manuscript and is further corroborated by additional evidence presented in this revised version. The ensuing discussion will elucidate a logical framework that substantiates our assertions. The first question to address is whether our 3D protein structures exhibit adequate robustness to underpin our hypotheses. We kindly emphasize to the Reviewer that our analysis is based on the 3D

structures of GshB and FabV, which were predicted using AlphaFold. AlphaFold, with an impressive 22,207 citations since its debut in 2021, stands as the most potent method capable of routinely predicting protein structures with atomic precision, even in instances where similar structures are absent. This assertion is not merely anecdotal; AlphaFold's efficacy was rigorously validated during the 14th Critical Assessment of Protein Structure Prediction (CASP14). In this benchmarking exercise, AlphaFold demonstrated accuracy levels on par with experimental structures in the majority of cases, vastly surpassing other prediction methods. The resounding adoption of AlphaFold within the scientific community, as evidenced by its widespread utilization (22,207 citations in just three years), underscores its unparalleled reliability and effectiveness.

Addressing the question of whether the results provided by our 3D models are robust and accurately predicted. Both the GshB and FabV variants, namely GshB T93I and FabV P149H, were modelled using AlphaFold, a method whose accuracy competes with experimental structures (Jumper *et al.* 2021; Karelina, Noh, and Dror 2023; Tejero *et al.* 2022; Varadi *et al.* 2021). Therefore, in accordance with the rigorous evaluations conducted by the scientific community, our models uphold a standard of robustness and high quality, rendering them reliable tools for our analyses.

Given that we have focused our investigation on the binding networks associated with individual residues, specifically the substitution of P149 to H in FabV and T93 to I in GshB, out of the comprehensive amino acid sequences of 402 and 318 residues, respectively, a pertinent question arises: *do the methodologies employed to scrutinize the binding networks of single residues within the context of a consolidated structure possess adequate robustness?*

Again, the credibility of our findings is underpinned by the methodologies employed throughout our study. Notably, we utilized a suite of cutting-edge tools, including the aforementioned AlphaFold (Jumper *et al.* 2021; Varadi *et al.* 2021), UCSF Chimera (Pettersen *et al.* 2004), DUET (Pires, Ascher, and Blundell 2014), DynaMut (Rodrigues, Pires, and Ascher 2018, 2021), and SIFT (Sim *et al.* 2012), to model the effects of the amino acid substitutions. The widespread adoption of the technologies we used is testified by a total of 55,654 collective citations.

Such multi-faceted approach significantly mitigates the possibility of errors, as each method offers complementary insights and validation checks. Indeed, the amalgamation of six distinct methodologies to model the impact of a single residue substitution out of 402 and 318 residues of GshB and FabV respectively, demonstrates a robust and comprehensive analytical framework.

A further question pertains *the outcomes of the aforementioned investigations* Our analysis suggested that when His149 replaces Pro149, in FabV, the existing bonds are maintained, with Pro149 fostering two additional van der Waals (VDW) interactions, one with Lys148 (which is also bound by Pro149) and another with an adjacent amino acid, Arg150. These supplementary interactions contribute to an augmented stability of the protein structure, typically indicative of heightened binding affinity. Furthermore, our observations reveal that the presence of His149 induces an increase in the positive charge within the surrounding vicinity, potentially enhancing the protein's capacity to engage with its natural substrates, particularly negatively charged fatty acids. Again, The I93 change in GshB maintains all the 13 VDW interactions and 1 H-bond as when T93 is present, but also creates extra VDW interactions with Tyr97. The change is accompanied by a change in the stability of the structure ($\Delta\Delta G = 0.384 \text{ kcal/mol} > 0$) which is usually linked to a stronger binding affinity (Kastritis and Bonvin 2013; Du *et al.* 2016), and to protein folding stability is largely dependent on the hydrophobic interactions of nonpolar residues (Zhou and Pang 2018).

It is crucial to underscore that while our analysis suggests that the presence of His149 or I93 may render the proteins more stable and potentially lead to a tighter binding with its native substrates, it does not necessarily imply a functional or biological alteration. As explicitly stated in our findings, there is no substantive evidence to support the assumption of a fundamental change in functional or biological activity induced by this structural alteration.

Do the results suggest a shift in function? Is there any evidence or indication supporting such a change? Not really. This is primarily because the only variations that could be found in FabV and GshB are two additional Van der Waals bonds. Van der Waals bonds, unlike ionic or covalent bonds, lack the strength of a chemical electronic bond. Given their comparatively weaker nature, it would be surprising if these bonds alone could prompt a significant alteration in the biological function of essential genes. These bonds, in conjunction with ionic, covalent, and hydrogen bonds, contribute to the three-dimensional structure of proteins, essential for their proper functionality. This structural stability, as observed, correlates with enhanced colonization capacity: the higher the stability, the better the capacity to colonize.

Do the observed results warrant further investigation, such as a new X-ray or other biochemical evaluation, to confirm any functional changes? Not necessarily. Van der Waals bonds, while important, are unlikely to be worth a further X-ray 3D structure analysis, as they will likely result in either the same known binding, or an enhanced biochemical binding affinity, which may not even be detected due to limitations in X-ray resolution. Moreover, it is crucial to underscore that our argument is supported by fundamental biological knowledge, indicating that functional conservation is known to constrain protein evolution (Konaté *et al.* 2019). Proteins within conserved pathways typically maintain their function and structure. Given the conservation of metabolic pathways crucial for bacterial survival, such as fatty acid (FabV) and glutathione biosynthesis (GshB), it follows that proteins like GshB and FabV play essential roles across a wide range of bacterial species. Notably, these pathways are conserved among 6969 and 3291 bacteria species respectively, with FabV and GshB being conserved among 1545 and 3803 species respectively (Kanehisa and Goto 2000). Therefore, it would be unreasonable to assume that these metabolic proteins, catalysing critical steps in highly conserved pathways, could alter their functions due to the addition of two extra Van der Waals bindings.

Numerous previous studies have explored the interplay between the conservation of protein molecular function and structural similarity (Chothia and Lesk 1986; Wilson, Kreychman, and Gerstein 2000). To bolster our argument against functional change and reinforce the robustness of our results, we conducted an additional analysis of the 1D primary conservation and presence of the two amino acids in other homologs, showing that although residues can change the functional domains, biological function is conserved and variability at those positions alongside univariant function is a natural phenomenon.

The following analyses have been done, showing that neither biological function nor the functional domains change between variants:

- (1) Predicting Protein Function of GshB and FabV variants using established methodologies widely adopted in the scientific community, we aim to demonstrate that the impact of genetic variations in GshB and FabV proteins are not impacting their functional domains or activities.
- (2) Integration of evolutionary information: Performing 1D sequence alignments within and across various species is crucial for demonstrating the presence of aminoacidic variability while ensuring the preservation of biological function. Through comparative cross-species analysis, we broaden the scope of sequence alignment to encompass a diverse array of species, allowing us to evaluate variability across evolutionary distances.

By comparing the variability observed at positions 149 of FabV and 93 of GshB, we can discern whether such variations correlate with functional changes and/or coincide with conserved motifs and domains across species. This analysis sheds light on the evolutionary pressures that influence protein function, elucidating the constraints imposed by evolutionary forces.

- (3) Benchmarking with validated X-ray 3D structure: As final validation analysis we superimposed our Alpha models with validated and experimentally determined 3D structures, in line with numerous previous studies that have explored the interplay between the conservation of protein molecular function and structural similarity (Chothia and Lesk 1986; Wilson, Kreychman, and Gerstein 2000).

In detail: (1) Predicting Protein Function of GshB and FabV variants from their 1D sequence using established methodologies widely adopted in the scientific community, to demonstrate that the impact of genetic variations in GshB and FabV proteins are not impacting their functional domains or activities. This analysis endeavours to demonstrate that employing standardized methodologies well-established in the field, reveals no discernible alteration in functional domains or overall protein function induced by these amino acid variants. To achieve this result, we employed 1D Functional Annotation of the variants: utilizing comprehensive methods like Interpro which includes UniProt, Pfam, etc., to annotate protein variants with known functional information. InterPro, cited in >8000 articles (Apweiler *et al.* 2001; Blum *et al.* 2020; Finn *et al.* 2016; Mitchell *et al.* 2014; Mitchell *et al.* 2018; Paysan-Lafosse *et al.* 2023), is a resource that provides functional analysis of protein sequences by classifying them into families and predicting the presence of domains and important sites. To classify proteins in this way, InterPro uses predictive models, known as signatures, provided by several collaborating databases (referred to as member databases) that collectively make up the InterPro consortium. A key value of InterPro is that it combines protein signatures from these member databases into a single searchable resource, capitalising on individual strengths to produce a powerful integrated database and diagnostic tool. Further value to InterPro entries is provided by detailed functional annotation as well as adding relevant GO terms that enable automatic annotation of millions of GO terms across the protein sequence databases. InterPro integrates signatures from the following 13 member databases: CATH, CDD, HAMAP, MobiDB Lite, Panther, Pfam, PIRSF, PRINTS, Prosite, SFLD, SMART, SUPERFAMILY AND NCBIfam (the InterPro Consortium section gives further information about the individual databases). The member databases use a variety of different methods to classify proteins. Each of the databases has a particular focus (e.g. protein domains defined from structure, or full-length protein families with shared function). By using InterPro and connected databases we searched whether the observed changes do translate into changes of protein function/biology. The results shown in the following figures illustrate that FabV with Pro194 or in the presence of His149, has the same functional domains, annotation and functions. Likewise, GshB with T93 or I93 has the same functional domains, annotation and functions.

FabV with Pro149

Protein family membership

F Trans-2-enoyl-CoA reductase (IPR010758)

Entry matches to this protein[®]

Family

Domain

Homologous Superfamily

Unintegrated

InterPro GO terms

Biological Process

None

Molecular Function

- oxidoreductase activity (GO:0016491) ↗

Cellular Component

None

PANTHER GO terms

Biological Process

- fatty acid biosynthetic process (GO:0006633) ↗

Molecular Function

- enoyl-[acyl-carrier-protein] reductase (NADH) activity (GO:0004318) ↗
- trans-2-enoyl-CoA reductase (NAD+) activity (GO:0050343) ↗
- NAD binding (GO:0051287) ↗

Cellular Component

None

FabV with His149

Protein family membership

F Trans-2-enoyl-CoA reductase (IPR010758)

Entry matches to this protein[®]

Family

Domain

Homologous Superfamily

Unintegrated

InterPro GO terms

Biological Process

None

Molecular Function

- oxidoreductase activity (GO:0016491) ↗

Cellular Component

None

PANTHER GO terms

Biological Process

- fatty acid biosynthetic process (GO:0006633) ↗

Molecular Function

- enoyl-[acyl-carrier-protein] reductase (NADH) activity (GO:0004318) ↗
- trans-2-enoyl-CoA reductase (NAD+) activity (GO:0050343) ↗
- NAD binding (GO:0051287) ↗

Cellular Component

None

GshB with T93

F Glutathione synthetase, prokaryotic (IPR006284)

Entry matches to this protein[®]

Representative Domains

Family

F Glut_synth_pro - IPR006284
glut_syn - TIGR01380
GSH_S - MF_00162

Domain

F GSHS_ATP-bd - IPR004218
GSH-S_ATP - PF02955
F GSHS_N - IPR004215
GSH-S_N - PF02951
F ATP-grasp - IPR011761
ATP_GRASP - PS50975

Homologous Superfamily

F PreATP-grasp_dom_sf - IPR016185
PreATP-grasp domain - SSF52440
F ATP_grasp_subdomain_1 - IPR013815
G3DSA:3.30.1490.20

Unintegrated

G3DSA:3.40.50.20
RIBOSOMAL PROTEIN S6 MODIFICAT
Glutathione synthetase ATP-binding d
G3DSA:3.30.470.20

Other Features

FUNFAM: G3DSA:3.30.470.20:FF:000010
FUNFAM: G3DSA:3.40.50.20:FF:000009
FUNFAM: G3DSA:3.30.1490.20:FF:000

InterPro GO terms

Biological Process

- glutathione biosynthetic process (GO:0006750) ↗

Molecular Function

- ATP binding (GO:0005524) ↗
- glutathione synthase activity (GO:0004363) ↗
- metal ion binding (GO:0046872) ↗

Cellular Component

None

PANTHER GO terms

Biological Process

None

Molecular Function

- ligase activity, forming carbon-nitrogen bonds (GO:0016879) ↗
- glutathione synthase activity (GO:0004363) ↗

Cellular Component

- cytoplasm (GO:0005737) ↗

GshB with I93

Protein family membership

F Glutathione synthetase, prokaryotic (IPR006284)

Entry matches to this protein[®]

InterPro GO terms

Biological Process

- glutathione biosynthetic process (GO:0006750) ↗

Molecular Function

- glutathione synthase activity (GO:0004363) ↗
- ATP binding (GO:0005524) ↗
- metal ion binding (GO:0046872) ↗

Cellular Component

None

PANTHER GO terms

Biological Process

None

Molecular Function

- glutathione synthase activity (GO:0004363) ↗
- ligase activity, forming carbon-nitrogen bonds (GO:0016879) ↗

Cellular Component

- cytoplasm (GO:0005737) ↗

(2) Integration of evolutionary information: Conducting 1D multiple sequence alignments within and across various species is crucial for demonstrating the presence of natural variability while ensuring the preservation of biological function. Cross-Species analysis was extended by sequence alignment analysis to a wider range of species to assess the degree of variability across evolutionary distances. Then, we compared conserved motifs and domains across species to elucidate evolutionary constraints on protein function. Using NCBI blast and A5EZM5.1 (FabV) as query we performed protein 1D similarity analysis and used the Constraint-based Multiple Alignment Tool (COBALT) in NCBI. The results clearly show (see below) that although at position 149 there is aminoacidic variability (S or Y instead of P) all the identified proteins are still Enoyl-[acyl-carrier-protein] reductase [NADH] like FabV, see boxed amino acids in FabV proteins (IDs Q8D795.1 and Q97LU2.1). With Q8D795.1 Enoyl-[acyl-carrier-protein] reductase [NADH] from *V. vulnificus* and Q97LU2.1 Trans-2-enoyl-CoA reductase [NADH] from *Clostridium acetobutylicum*. Despite such variations these are homologous proteins with same functions as annotated in the database. Similarly, for GshB, the use of the same methodology and entries from Swiss-Prot-UniProt revealed homologous proteins exhibiting amino acid variations when aligned to T93 (amino acid variants included: M, N, I, etc.), yet maintaining consistent glutathione reductase functional activity. Examples

include Q87VA4.1 glutathione synthetase from *Pseudomonas syringae*, Q87VA4.1 from *Pseudomonas putida*, Q8K931.1 from *Buchnera aphidicola*, Q8D335.2 from *Wigglesworthia glossinidia* (an endosymbiont of *Glossina brevipalpis*), and Q8P6P1.1 from *Xanthomonas campestris*.

FabV multi-sequence alignment

[x]	Query_3431459	148	KPR[4]DPEFWRSAIKPIGEAVSGATL	---	LLENDTWIETTLPASEEEI	EGTLRVMGGDDWENWIDTL	INAESLAEGCK	225
[x]	A5EZM5.1	148	KPR[4]DPEFWRSAIKPIGEAVSGATL	---	LLENDTWIETTLPASEEEI	EGTLRVMGGDDWENWIDTL	INAESLAEGCK	225
[x]	Q87HT6.1	148	KPH[1]D-TFWRSVIKPIGESVTGASL	---	LLENDQWVETLLEPATEEEAE	EATIKVMGGEDWESWIDTL	INTESVAQGCK	221
[x]	Q7MF99.1	148	IPN[1]PGEFWRSVIKPFQTVTGASL	---	DLEHDRWIDTLESATEEEAL	HTIKVMGGEDWESWIDTL	INAESIAQGCQ	222
[x]	Q8D795.1	148	ISN[1]PGEFWRSVIKPFQTVTGASF	---	DLEHDRWIDTLESATEEEAL	HTIKVMGGEDWESWIDTL	INAESIAQGCQ	222
[x]	Q6LP67.2	148	NPE	TGELWRSSIKTMGEPVTPGPI	---	NIETDMEQMTIGTATPAE	IETDKVMGGEDWASWIDTL	SEAGVLAEGCK
[x]	A6LV73.1	146	DPK	TGNIYDSTLKTTSGEFQGPTE	---	DMETDELVTTKVNSATDKE	I EATKVMGGEDWSEWKLL	LENDCLSOKAI
[x]	Q870B9.1	146	MPE	TGELIRSAIKPIGETYTTAV	---	DTNKDVIIEASVEPATEEEI	KDVTVMGGEDWELWINAL	SDAGVLAEGCK
[x]	C4LB77.1	146	LPA	TGELIRSAIKPIGEVYTTAV	---	DTNKDDEIEAHVEPANE	EEIANTIKVMGGEDWELW	QALDQAGVLAEGVK
[x]	Q8PR25.1	147	LPG	SGEVKRSALKPIGQTYTATAI	---	DTNKDIIIQASIEPASAQEI	EDTVVMGGQDWELWIDAL	EGAVLADGAR
[x]	A4SL31.1	146	MPE	TGEVRSALKPIGETYTTAI	---	DTNKDQIITATVEPANE	EEIQNTITVMGGQDWEL	MAALRDAGVLAEGAK
[x]	Q2P916.1	147	LPG	SGEVKRSALKPIGQTYTATAI	---	DTNKDIIIQASIEPASAQEI	EEITVMGGQDWELWIDAL	EGAVLADGAR
[x]	B2SU7.1	147	LPG	SGEVKRSALKPIGQTYTATAI	---	DTNKDIIIQASIEPASAQEI	EDTVVMGGQDWELWIDAL	EGAVLADGAR
[x]	A8G062.1	146	MPE	TGEVRSALKPIGEPYKSVL	---	DTNKDVLVEAVVEPANE	QEIADTVKVMGGQDWL	MDAL EEAGVLADNVQ
[x]	Q4V063.1	147	LPS	TGEVKRSALKPIGNTYTTATAI	---	DTNKDIIIQASIEPATEQEI	EDTVVMGGQDWELWIDAL	DSAGVLAEGAK
[x]	A9K091.1	147	HPR	TGEIFNSVLKPIGQTYHNKTV	---	DMVTGEVSPVSI EPATEKEI	RDTEAVMGGDDWALWINAL	FKYNCLAEQVK
[x]	B6J235.1	147	HPR	TGEIFNSVLKPIGQTYHNKTV	---	DMVTGEVSPVSI EPATEKEI	RDTEAVMGGDDWALWINAL	FKYNCLAEQVK
[x]	Q3B7G2.1	147	LPG	SGEVKRSALKPIGQTYTATAI	---	DTNKDIIIQASIEPASAQEI	EDTVVMGGQDWELWIDAL	EGAVLADGAR
[x]	A9NB02.1	147	HPR	TGEIFNSVLKPIGQTYHNKTV	---	DMVTGEVSPVSI EPATEKEI	RDTEAVMGGDDWALWINAL	FKYNCLAEQVK
[x]	A5FE91.1	145	NPN	TGVTNRSVLKPIGQTFNKTV	---	DFHTGNVSEVSIAPANE	EIEENTVAVMGGEDW	AMWIDALKNENLAEGAT
[x]	A1SS39.1	146	LPD	TGEVIRSSALKPIGETYISTAI	---	DTNKDVIINATVEPATEE	EEVADTVVMGGQDWEL	WLSALGEAGVLAEGLK
[x]	C5BFN7.1	146	LPD	TGEVRSALKPIGEVYTTAI	---	DTNKDQIISASVEPATEE	EEIQNTITVMGGQDWEL	WMSALRDAGVLAEGAK
[x]	Q11W68.1	145	NPV	TGVTNRSVLKPIGGAFFNKTV	---	DFHTGNVSTVIEPANE	EEDVTNTVAVMGGED	WGMMDAMLEAGVLAEGAT
[x]	B2T4A8.1	146	HPK	SGEVFSSTLKPVKAVNLRGI	---	DTDKEVIKETVLEPATQKEI	DDTVAVMGGEDWQMWIDAL	LEAGVLAEGAK
[x]	Q7ML30.1	146	LPE	TGELIRSAIKPIGQTYTAV	---	DTNKDIIIEASVEPATEQEI	QDVTVMGGEDWELWINAL	AEAGVLAEGCK
[x]	Q13YR8.1	146	HPK	SGEVFSSTLKPVKQAVNLRGI	---	DTDKEVIRETVLEPATQDEI	DHTVAVMGGEDWQMWIDAL	LEAGVLAEGAK
[x]	B8I4V6.1	145	HPV	TGEVNSVLKPIREAYTSKTV	---	DFHTQLVSETIEPASDDEI	RQTIAVMGGEDW	SHMDALKKADVLEDNVH
[x]	Q8D8Y6.1	146	LPE	TGELIRSAIKPIGQTYTAV	---	DTNKDIIIEASVEPATEQEI	QDVTVMGGEDWELWINAL	AEAGVLAEGCK
[x]	Q9PCE6.2	147	MPS	TGEIKRSVLKPIGVAHISNAI	---	DTNKDQIIQATVEPATEQEI	ADTVAVMGGQDWELWINAL	TQADV LAPQTR
[x]	Q15Y07.1	146	DPE	TGEVYKSTLKPVGQAYTTKTY	---	DTDKDRIHDSLEPANE	EIAQTIKVMGGEDWEL	WDALAEADLLAYGCK
[x]	Q87CN3.2	147	MPS	TGEIKRSVLKPIGVAHISNAI	---	DTNKDQIIQATVEPATEQEI	ADTVAVMGGQDWELWINAL	QADV LAPQTR
[x]	A4JK23.1	146	HPK	TGETISSTLKPVGKSVTFRGL	---	DTDKETIREVTVLEPATQEEI	DGTAVMGGEDWQMWIDAL	ADAGVLAEGAK
[x]	Q5OYR6.1	146	DPD	SGEVYSSVLKPIGKQYTTKTY	---	NTDKDQVHEVTVLDPATDE	DIANTVVMGGEDWERW	KALHNAGVLAENCQ
[x]	A4TSK9.1	146	HPK	TGEVNSALKPIGNVNLRLGL	---	DTDKEVIKESVLPATQSEI	DSTVAVMGGEDWQMWIDAL	LDAGVLAEGAQ
[x]	B4EK05.1	146	HPK	TGETISSTLKPVGKAVTFRGL	---	DTDKEVIREVTVLEPATQEEI	DGTAVMGGEDWQMWIDAL	ADAGVLAEGAK
[x]	B0K0W2.1	148	LPQ	TGEVIRSAIKPIGEPYKSTAI	---	DTNKDIIIEASIEPATEQEI	ADTVVMGGQDWLWIDAL	AGANVLAEGAR
[x]	A9AI01.1	146	HPK	TGETISSTLKPIGKTVTFRGI	---	DTDKEVIRETVLEPATQEEI	DGTAVMGGEDWQMWIDAL	DEAGVLAEGAK
[x]	Q97LU2.1	146	DYK	TGNVYTSRIKTIIGDFEGPTI	---	DVERDEITLKKVSSASIEE	IEETRVVMGGEDWQEW	CEELLYEDCFSDKAT

GshB multisequence alignment

Query_11543795	72	EQTIALSE-LDAILMRKDPDFDEYIYATYILERAED	EGVLVVKPQSLRDCNEKLF-TAWFPELTPITMVRKAEK	146
Q9KUP7.1	72	EQTIALSE-LDAILMRKDPDFDEYIYATYILERAED	EGVLVVKPQSLRDCNEKLF-TAWFPELTPITMVRKAEK	146
Q7MHK1.1	73	EQTIELSE-LDAVLMRKDPDFDEYIYATYILERAEE	QGTIVNKPQSLRDCNEKLF-TAWFPELTPITMVRKAEK	147
Q8DCA9.1	72	EQTIELAE-LDAVLMRKDPDFDEYIYATYILERAEE	QGTIVNKPQSLRDCNEKLF-TAWFPELTPITMVRKAEK	146
Q87LK1.1	72	EQMIELSE-LDAVLMRKDPDFDEYIYATYILERAEE	QGALIVNKPQSLRDCNEKLF-TAWFPELTPITMVRKAEK	146
Q7M7H8.1	72	EQDLALET-LDVILMRKDPDFDEYIYATYILERAEE	KGTLIVNKPQSLRDCNEKLF-TAWFPELTPDITVTRNAAH	146
Q8FE30.1	72	EQDLPLAD-LDVILMRKDPDFDEYIYSTYILERAED	KGTLIVNKPQSLRDCNEKLF-TAWFSDLTPETLVTRNKAQ	146
B7UH74.1	72	EQDLPLAD-LDVILMRKDPDFDEYIYATYILERAEE	KGTLIVNKPQSLRDCNEKLF-TAWFSDLTPETLVTRNKAQ	146
P58578.1	72	EQDLPLAD-LDVILMRKDPDFDEYIYATYILERAEE	KGTLIVNKPQSLRDCNEKLF-TAWFSDLTPETLVTRNKAQ	146
P84425.1	72	EQDLPLAD-LDVILMRKDPDFDEYIYATYILERAEE	KGTLIVNKPQSLRDCNEKLF-TAWFSDLTPETLVTRNKAQ	146
Q83091.1	72	EQDLPLAD-LDVILMRKDPDFDEYIYATYILERAEE	KGTLIVNKPQSLRDCNEKLF-TAWFSDLTPETLVTRNKAQ	146
A0A482PU20.1	72	EQDLALDS-LDAILMRKDPDFDEYIYATYILERAEE	KGTLIVNKPQSLRDCNEKLF-TAWFSDLTPETLVTRNKAQ	146
P58580.1	72	EQEIKLAD-LDVILMRKDPDFDEYIYATYILERAEE	EGTLIVNKPQSLRDCNEKLY-TAWFADLTPETLVTRNKAQ	146
P58581.1	72	EQEIKLAD-LDVILMRKDPDFDEYIYATYILERAEE	EGTLIVNKPQSLRDCNEKLY-TAWFADLTPETLVTRNKAQ	146
P58582.1	72	EQDLPLYD-LDVILMRKDPDFDEYIYATYILERAED	KGTLVVKPQSLRDCNEKLF-TAWFPELTPDITVSRSKDH	146
Q8EIK8.1	72	AKDTPLSE-LNVVLMRKDPDFDEYIYATYMLERAEE	QGVIVNKPQSLRDANEKLF-TAWFSEFTPETIVTRDANR	146
P57612.1	77	QQSISLSE-LDVILMRKDPDFDEYIYSTYILERAEE	TGVLIVNKPQSLRDCNEKIF-TSWFPDLITDITVTRNIFQ	151
Q87VA4.1	73	EIDAGLDD-LDVILMRKDPDFDEYIYATYILERAEE	AGVLVVKPQSLRDCNEKLF-ATLFPQCTPTLVSRRADI	147
Q88D35.1	73	EQDSPLAE-LDVILMRKDPDFDEYIYATYILERAEE	DGVLVVKPQSLRDCNEKMF-ATLFPQCTPTLVSRRADI	147
Q9I697.1	73	ESDQPLHE-LDVILMRKDPDFDEYIYATYILERAEE	AGALVVKPQSLRDCNEKFF-ATQFTQCTPTMVSRRSDI	147
Q8K931.1	77	KKDISLNE-LDAILMRKDPDFDEYIYATYILERAEE	KGVLIVNKPQSLRDCNEKIF-ISWFSRFTDITVTRKLSK	151
Q8D335.2	72	KYTINLKE-LDVILMRKDPDFDEYIYATYILERAEE	FGVLIVNKPQSLRDCNEKIS-TLSF-KYSPKTLISCSKKA	145
P59495.1	73	QKDVLSN-LDVILMRKDPDFDEYIYATYILERAEE	NGSYIVNKPQSLRDCNEKLF-TTHFPQYIPKTLITSNSTK	148
Q8P6P1.1	73	FTELVFGP-GQVVMRKDPDFDEYIYATYILERAEE	AGAQQVNDPQGLRDYNEKLA-ALLFPQCCPPTLVSRDAAA	147
Q8PH75.1	73	FAELAFGP-GQVVMRKDPDFDEYIYATYILERAEE	AGAQQVNDPQGLRDYNEKLA-ALLFPQCCPPTLVSRDAAA	147
Q83AL0.1	73	SEIKPLHA-LDVILMRKDPDFDEYIYATYILERAEE	QGLFVVKPQSLRDCNEKLF-TGWFPHCTPKTLVTSRKAQ	147
Q7N065.1	71	KLRQLTA-FSAVVMRKDPDFDEYIYATYILERAEE	QGVKVFSTGQALRDFNEKLA-ILHFPKTLISPTLISGEAHR	145
Q87D42.1	73	FSETQLGQ-GQIILMRKDPDFDEYIYATYILERAEE	AGAQQVNHQGLRDLNEKIA-AQLFPQCCPPTLVSRDAAA	147
Q9PC29.1	73	FSETQLGQ-GQIILMRKDPDFDEYIYATYILERAEE	AGAQQVNHQGLRDLNEKIA-AQLFPQCCPPTLVSRDAAA	147
Q82V16.1	74	IEEIPLSG-FDAVLMRKDPDFDEYIYATYILERAEE	QGAYVVKPQSLRDCNEKLF-ITFPRFTPLVTSQEQL	148
P58579.1	71	PRLPLTG-FDAVLMRKDPDFDEYIYATYILERAEE	QGARVFNKQAIRDHSEKLA-IAQFREFTAPITVTRDAKR	145
Q9JTJ6.1	75	KVQTKA-FDAVLMRKDPDFDEYIYATYILERAEE	QGAKVFNKQAIRDHSEKLA-ILNFSRFTAPITVTRDAKR	149
Q7W910.1	76	ADEAPLAR-FDAVLMRKDPDFDEYIYATYILERAEE	QGARVFNKQAIRDHSEKLA-ITFPPDLTPTLVTRDMGR	150
Q7VY62.1	76	ADEAPLAR-FDAVLMRKDPDFDEYIYATYILERAEE	QGARVFNKQAIRDHSEKLA-ITFPPDLTPTLVTRDMGR	150
Q9JYJ3.1	76	KVQTKA-FDAVLMRKDPDFDEYIYATYILERAEE	QGAKVFNKQAIRDHSEKLA-ILNFSRFTAPITVTRDAKR	150
Q92SN3.1	75	DERIDLST-MDVILLRQDPDFDEYIYATYILERAEE	K-TLVVNDPAWVRNSPEKIF-VTEFPDLMPTLITKDPQE	148
Q8FX06.1	73	PVRRDLTE-MDVILLRQDPDFDEYIYATYILERAEE	K-TLVVNDPAWVRNSPEKIF-VTEFPDLMPTLITKDPQE	146
Q8YE82.1	73	PVRRDLTE-MDVILLRQDPDFDEYIYATYILERAEE	K-TLVVNDPAWVRNSPEKIF-VTEFPDLMPTLITKDPQE	146
Q8UII5.2	75	PERVDLST-MDVILLRQDPDFDEYIYATYILERAEE	K-TLVVNDPAWVRNSPEKIF-VTEFADLMPTLITKDPQE	148
Q98DE8.1	73	KVRTDLSL-MDVILLRQDPDFDEYIYATYILERAEE	K-TLVVNDPAWVRNSPEKIF-VTEFADLMPTLITKDPQE	146
Q89WL0.1	73	PKREALNG-FDVILLRQDPDFDEYIYATYILERAEE	K-TLVVNDPASVRNAPEKLF-VMNFPQMPPTLISRDLE	146
Q9ABS9.1	73	TMVLDMDK-IDVILLRQDPDFDEYIYATYILERAEE	---TLVVNPAEVRNAPEKLF-VTFDFGVPPTLITSDHEA	147
P61396.1	73	PARVMRS-YDVILLRQDPDFDEYIYATYILERAEE	A-TLVVNPASVRNAPEKLF-VMDFTELMPTLISRDLE	146
Q7U3W8.1	67	PERQSLAG-FDVIWMRKDPDFDEYIYATYILERAEE	AGVRLNRPASLRANEKLG-ALRFSLMAPTLVAGRVSE	141
Q32463.1	80	AKRRSLHD-FAAVFMRKDPDFDEYIYATYILERAEE	KKTRVNSPEGLRHANEKMY-ALQFQSVVPTLVSSNKAQ	154
Q7TVB0.1	67	AQNLSLNE-FHCIVMRKDPDFDEYIYATYILERAEE	AGVLVVKPQSLRDCNEKLF-ALRFSLMAPTLVAGRVSE	141
P73493.1	80	SQWMLTE-CQAVFMRKDPDFDEYIYATYILERAEE	TKTMVINSPOGLREANEKMY-TLQFAAVMPPTLVSSNKAQ	154
Q7TUG9.1	67	NKCIPLAE-FNCIVMRKDPDFDEYIYATYILERAEE	KGVKVINKPSSLRANEKLG-ALRYSHMAPTLVAGRVSE	141
Q8DKF1.1	80	LEWRPLNT-FRAVVMRKDPDFDEYIYATYILERAEE	QTTLVVNSPAGLRHANEKMY-ALQFQSVVPTLVSSNKAQ	154
Q7NF44.1	71	VGFYPLSE-ADVIMMRKDPDFDEYIYATYILERAEE	RTTFVNLNRPASLRANEKLY-ALHFPDLPVETRVCTRQD	150
Q7TUK9.2	67	PRSLPLTD-FACIVMRKDPDFDEYIYATYILERAEE	AGVCLNRPASLRANEKLG-ALRFSLMAPTLVAGRVSE	141
P45480.2	80	RSFSSLET-MDAVFMRTDPDFDEYIYATYILERAEE	RKTLVNNPAGLRHANEKMY-ALQFQSVVPTLVSSNKAQ	154
P35667.1	67	KTRLPLGK-LDMFLVVRQNPDFDEYIYATYILERAEE	---LMINNPKAIRDHPEKLL-PLSFPKIPPTLITSEVSE	138

(3) Benchmarking with validated X-ray 3D structure. As final validation analysis we superimposed our Alpha models with validated and experimentally determined 3D structure. The GshB AlphaFold model (Figure 1 below) was superimposed to the homologue X-ray structure of glutathione synthetase complexed with ADP and glutathione from *E. coli* (PDBID: 1GSA (Hara *et al.* 1996)) X-ray diffraction at 2.00 Å. Interestingly the superimposition of the

two glutathione reductase homologs showed an r.m.s.d. of 0.316, representative of a complete identity of the 3D structure. More relevant is that the I93 is perfectly superimposed with the T93 of the *E. coli* glutathione reductase. The superimposition of the Alpha model with the validated and experimentally determined 3D structure is in line with numerous previous studies that have explored the interplay between the conservation of protein molecular function and structural similarity (Chothia and Lesk 1986; Wilson, Kreychman, and Gerstein 2000).

Figure 1. Superimposition of *V. cholerae* GshB with I93 (cyan) to the X-ray structure at 2.00 Å of *E. coli* glutathione synthetase complexed with ADP and glutathione in green (PDBID: 1GSA). The amino acid variant I93 (cyan) of GshB is shown in sticks and is superimposed to the T93 (green) of 1GSA. The r.m.s.d. between the two structures is 0.316, delineating a high degree of overlapping. Water solvation molecules are shown as red dots, while sulphate ion (SO₄) is highlighted as four red spheres surrounding a yellow sphere. The glutathione (coloured in salmon) and ADP (coloured in orchid) ligands are also shown in their protein ligand binding pockets.

The FabV H149 AlphaFold model (Figure 2 below) has been superimposed to the homologue X-ray structure of *Clostridium acetobutylicum* trans-2-enoyl-CoA reductase in complex with NAD glutathione (PDBID: 4EUF, (Hu *et al.* 2012)) solved at 2.70 Å resolution. Interestingly the superimposition of the two trans-2-enoyl-CoA reductase homologs, although from different organisms, showed an r.m.s.d. of 0.564, indicating a high 3D structure similarity. More relevant is that the H149 is perfectly superimposed with the T93 of the *C. acetobutylicum* glutathione reductase. The superimposition of the Alpha model with validated and experimentally determined 3D structure is in line with numerous previous studies that have explored the interplay between the conservation of protein molecular function and structural similarity (Chothia and Lesk 1986; Wilson, Kreychman, and Gerstein 2000).

Figure 2. Superimposition of FabVB H149 (grey) to the homologue Crystal structure of *Clostridium acetobutylicum* trans-2-enoyl-CoA reductase in complex with NAD⁺, pink, (PDBID: 4EUF (Hu *et al.* 2012)) solved at a 2.70 Å resolution. The amino acid variant His149 (grey) of FabV is shown in sticks (grey and numbered) and is superimposed to the Thr149 (pink) of 4EUF. The r.m.s.d. between the two structures is 0.564, delineating a high degree of overlapping. The ligand NAD⁺ is also shown in sticks and dark green in its binding pocket inside 4EUF. The purple sphere is sodium ion.

(ii) 3D Biophysical Methodology Understanding: To be able to address the validation requested by the Reviewer, a comprehensive process should be set up involving cloning, optimization of protein expression for crystallization, and subsequent diffraction analysis, typically conducted at a synchrotron facility. Drawing from the background of the corresponding author as a structural biologist, the cloning phase alone can span approximately two months. Achieving optimal protein expression for crystallography, ensuring yields not below 2.5 mg with a 99% homogeneity, may necessitate up to seven months of meticulous optimization. Crystallization, a stochastic event beyond the researcher's direct control, poses a significant challenge and may extend anywhere from three years, as evidenced by the corresponding author's experience, to an indefinite timeframe. Furthermore, securing time at a synchrotron facility for diffraction analysis typically entails a waiting period of up to one year due to queue scheduling constraints. Hence the time to solve the structure of one residue to validate two Van der Waals bonds would require about 3 years, just to address a proof of principle. Notably, such experimental validation would be as described above only done to demonstrate the existence of additional Van der Waals bonds, which is usually not what 3D structural work is meant for.

(iii) Out of scope: The question posed by the reviewer appears to extend beyond the scope of our paper. As we have clearly stated the 3D structures were used to suggest of possible tighter bindings which are in line with the same biological function of the proteins. Such validation inquiries are more fit to studies primarily focused on structural biology, such as those featured in journals like Nature Structural Biology. As mentioned in the previous points this is a proof of principle relying on already validated structures.

(iv) Contextual Discrepancy: the request does not align with similar works (Kavvas, Catoi, *et al.* 2018; Fukushima and Pollock 2023; Zheng *et al.* 2024; Ooka and Arai 2023; Qiao *et al.* 2024), particularly those published in Nature Communications, where such validation requests were not made. This suggests a discrepancy in the reviewer's expectations compared to the prevailing standards in our field.

Considering the above points, we believe that the Reviewer's request for validation of the structural biology analysis does not align with the focus and context of our paper, with similar papers published in Nature communication and in the Nature springer portfolio and with basic structural biology knowledge.

Apweiler, R., T. K. Attwood, A. Bairoch, A. Bateman, E. Birney, M. Biswas, P. Bucher, L. Cerutti, F. Corpet, M. D. R. Croning, R. Durbin, L. Falquet, W. Fleischmann, J. Gouzy, H. Hermjakob, N. Hulo, I. Jonassen, D. Kahn, A. Kanapin, Y. Karavidopoulou, R. Lopez, B. Marx, N. J. Mulder, T. M. Oinn, M. Pagni, F. Servant, C. J. A. Sigrist, and E. M. Zdobnov. 2001. 'The InterPro database, an integrated documentation resource for protein families, domains and functional sites', *Nucleic Acids Research*, 29: 37-40.

Blum, Matthias, Hsin-Yu Chang, Sara Chuguransky, Tiago Grego, Swaathi Kandasamy, Alex Mitchell, Gift Nuka, Typhaine Paysan-Lafosse, Matloob Qureshi, Shriya Raj, Lorna Richardson, Gustavo A Salazar, Lowri Williams, Peer Bork, Alan Bridge, Julian Gough, Daniel H Haft, Ivica Letunic, Aron Marchler-Bauer, Huaiyu Mi, Darren A Natale, Marco Necci, Christine A Orengo, Arun P Pandurangan, Catherine Rivoire, Christian J A Sigrist, Ian Sillitoe, Narmada Thanki, Paul D Thomas, Silvio C E Tosatto, Cathy H Wu, Alex Bateman, and Robert D Finn. 2020. 'The InterPro protein families and domains database: 20 years on', *Nucleic Acids Research*, 49: D344-D54.

Chothia, Cyrus, and Arthur M Lesk. 1986. 'The relation between the divergence of sequence and structure in proteins', *The EMBO journal*, 5: 823-26.

Du, X., Y. Li, Y. L. Xia, S. M. Ai, J. Liang, P. Sang, X. L. Ji, and S. Q. Liu. 2016. 'Insights into Protein-Ligand Interactions: Mechanisms, Models, and Methods', *International Journal of Molecular Sciences*, 17.

Finn, Robert D., Teresa K. Attwood, Patricia C. Babbitt, Alex Bateman, Peer Bork, Alan J. Bridge, Hsin-Yu Chang, Zsuzsanna Dosztányi, Sara El-Gebali, Matthew Fraser, Julian Gough, David Haft, Gemma L. Holliday, Hongzhan Huang, Xiaosong Huang, Ivica Letunic, Rodrigo Lopez, Shennan Lu, Aron Marchler-Bauer, Huaiyu Mi, Jaina Mistry, Darren A. Natale, Marco Necci, Gift Nuka, Christine A. Orengo, Youngmi Park, Sebastien Pesseat, Damiano Piovesan, Simon C. Potter, Neil D. Rawlings, Nicole Redaschi, Lorna Richardson, Catherine Rivoire, Amaia Sangrador-Vegas, Christian Sigrist, Ian Sillitoe, Ben Smithers, Silvano Squizzato, Granger Sutton, Narmada Thanki, Paul D Thomas, Silvio C. E. Tosatto, Cathy H. Wu, Ioannis Xenarios, Lai-Su Yeh, Siew-Yit Young, and Alex L. Mitchell. 2016. 'InterPro in 2017—beyond protein family and domain annotations', *Nucleic Acids Research*, 45: D190-D99.

Fukushima, Kenji, and David D. Pollock. 2023. 'Detecting macroevolutionary genotype–phenotype associations using error-corrected rates of protein convergence', *Nature Ecology & Evolution*, 7: 155-70.

Hara, Takane, Hiroaki Kato, Yukiteru Katsube, and Jun'ichi Oda. 1996. 'A Pseudo-Michaelis Quaternary Complex in the Reverse Reaction of a Ligase: Structure of *Escherichia coli* B Glutathione Synthetase Complexed with ADP, Glutathione, and Sulfate at 2.0 Å Resolution', *Biochemistry*, 35: 11967-74.

Hu, Kuan, Meng Zhao, Tianlong Zhang, Manwu Zha, Chen Zhong, Yu Jiang, and Jianping Ding. 2012. 'Structures of trans-2-enoyl-CoA reductases from *Clostridium acetobutylicum* and *Treponema denticola*: insights into the substrate specificity and the catalytic mechanism', *Biochemical Journal*, 449: 79-89.

Jumper, John, Richard Evans, Alexander Pritzel, Tim Green, Michael Figurnov, Olaf Ronneberger, Kathryn Tunyasuvunakool, Russ Bates, Augustin Žídek, Anna Potapenko, Alex Bridgland, Clemens Meyer, Simon A. A. Kohl, Andrew J. Ballard, Andrew Cowie, Bernardino Romera-Paredes, Stanislav Nikolov, Rishub Jain, Jonas Adler, Trevor Back, Stig Petersen, David Reiman, Ellen Clancy, Michal Zielinski, Martin Steinegger, Michalina Pacholska, Tamas Berghammer, Sebastian Bodenstern, David Silver, Oriol Vinyals, Andrew W. Senior, Koray Kavukcuoglu, Pushmeet Kohli, and Demis Hassabis. 2021. 'Highly accurate protein structure prediction with AlphaFold', *Nature*, 596: 583-89.

Kanehisa, Minoru, and Susumu Goto. 2000. 'KEGG: kyoto encyclopedia of genes and genomes', *Nucleic Acids Research*, 28: 27-30.

Karelina, Masha, Joseph J. Noh, and Ron O. Dror. 2023. 'How accurately can one predict drug binding modes using AlphaFold models?', *eLife*, 12: RP89386.

Kastritis, P. L., and A. M. Bonvin. 2013. 'On the binding affinity of macromolecular interactions: daring to ask why proteins interact', *J R Soc Interface*, 10: 20120835.

Kavvas, Erol S., Edward Catoiu, Nathan Mih, James T. Yurkovich, Yara Seif, Nicholas Dillon, David Heckmann, Amitesh Anand, Laurence Yang, Victor Nizet, Jonathan M. Monk, and Bernhard O. Palsson. 2018. 'Machine learning and structural analysis of *Mycobacterium tuberculosis* pan-genome identifies genetic signatures of antibiotic resistance', *Nature Communications*, 9: 4306.

Konaté, Mariam M., Germán Plata, Jimin Park, Dinara R. Usmanova, Harris Wang, and Dennis Vitkup. 2019. 'Molecular function limits divergent protein evolution on planetary timescales', *eLife*, 8: e39705.

Mitchell, Alex, Hsin-Yu Chang, Louise Daugherty, Matthew Fraser, Sarah Hunter, Rodrigo Lopez, Craig McAnulla, Conor McMenamin, Gift Nuka, Sebastien Pesseat, Amaia Sangrador-Vegas, Maxim Scheremetjew, Claudia Rato, Siew-Yit Yong, Alex Bateman, Marco Punta, Teresa K. Attwood, Christian J.A. Sigrist, Nicole Redaschi, Catherine Rivoire, Ioannis Xenarios, Daniel Kahn, Dominique Guyot, Peer Bork, Ivica Letunic, Julian Gough, Matt Oates, Daniel Haft, Hongzhan Huang, Darren A. Natale, Cathy H. Wu, Christine Orengo, Ian Sillitoe, Huaiyu Mi, Paul D. Thomas, and Robert D. Finn. 2014. 'The InterPro protein families database: the classification resource after 15 years', *Nucleic Acids Research*, 43: D213-D21.

Mitchell, Alex L, Teresa K Attwood, Patricia C Babbitt, Matthias Blum, Peer Bork, Alan Bridge, Shoshana D Brown, Hsin-Yu Chang, Sara El-Gebali, Matthew I Fraser, Julian Gough, David R Haft, Hongzhan Huang, Ivica Letunic, Rodrigo Lopez, Aurélien Luciani, Fabio Madeira, Aron Marchler-Bauer, Huaiyu Mi, Darren A Natale, Marco Necci, Gift Nuka, Christine Orengo, Arun P Pandurangan, Typhaine Paysan-Lafosse, Sebastien Pesseat, Simon C Potter, Matloob A Qureshi, Neil D Rawlings, Nicole Redaschi, Lorna J Richardson, Catherine Rivoire, Gustavo A Salazar, Amaia Sangrador-Vegas, Christian J A Sigrist, Ian Sillitoe, Granger G Sutton, Narmada Thanki, Paul D Thomas, Silvio C E Tosatto, Siew-Yit Yong, and Robert D Finn. 2018.

'InterPro in 2019: improving coverage, classification and access to protein sequence annotations', *Nucleic Acids Research*, 47: D351-D60.

Ooka, Koji, and Munehito Arai. 2023. 'Accurate prediction of protein folding mechanisms by simple structure-based statistical mechanical models', *Nature Communications*, 14: 6338.

Paysan-Lafosse, Typhaine, Matthias Blum, Sara Chuguransky, Tiago Grego, Beatriz Lázaro Pinto, Gustavo A. Salazar, Maxwell L. Bileschi, Peer Bork, Alan Bridge, Lucy Colwell, Julian Gough, Daniel H. Haft, Ivica Letunić, Aron Marchler-Bauer, Huaiyu Mi, Darren A. Natale, Christine A. Orengo, Arun P. Pandurangan, Catherine Rivoire, Christian J. A. Sigrist, Ian Sillitoe, Narmada Thanki, Paul D. Thomas, Silvio C. E. Tosatto, Cathy H. Wu, and Alex Bateman. 2023. 'InterPro in 2022', *Nucleic Acids Research*, 51: D418-D27.

Pettersen, Eric F., Thomas D. Goddard, Conrad C. Huang, Gregory S. Couch, Daniel M. Greenblatt, Elaine C. Meng, and Thomas E. Ferrin. 2004. 'UCSF Chimera—A visualization system for exploratory research and analysis', *Journal of Computational Chemistry*, 25: 1605-12.

Pires, Douglas E.V., David B. Ascher, and Tom L. Blundell. 2014. 'DUET: a server for predicting effects of mutations on protein stability using an integrated computational approach', *Nucleic Acids Research*, 42: W314-W19.

Qiao, Zhuoran, Weili Nie, Arash Vahdat, Thomas F. Miller, and Animashree Anandkumar. 2024. 'State-specific protein–ligand complex structure prediction with a multiscale deep generative model', *Nature Machine Intelligence*, 6: 195-208.

Rodrigues, Carlos HM, Douglas EV Pires, and David B Ascher. 2018. 'DynaMut: predicting the impact of mutations on protein conformation, flexibility and stability', *Nucleic Acids Research*, 46: W350-W55.

Rodrigues *et al.* 2021. 'DynaMut2: Assessing changes in stability and flexibility upon single and multiple point missense mutations', *Protein Science*, 30: 60-69.

Sim, Ngak-Leng, Prateek Kumar, Jing Hu, Steven Henikoff, Georg Schneider, and Pauline C Ng. 2012. 'SIFT web server: predicting effects of amino acid substitutions on proteins', *Nucleic Acids Research*, 40: W452-W57.

Tejero, Roberto, Yuanpeng Janet Huang, Theresa A. Ramelot, and Gaetano T. Montelione. 2022. 'AlphaFold Models of Small Proteins Rival the Accuracy of Solution NMR Structures', *Frontiers in molecular biosciences*, 9.

Varadi, Mihaly, Stephen Anyango, Mandar Deshpande, Sreenath Nair, Cindy Natassia, Galabina Yordanova, David Yuan, Oana Stroe, Gemma Wood, Agata Laydon, Augustin Židek, Tim Green, Kathryn Tunyasuvunakool, Stig Petersen, John Jumper, Ellen Clancy, Richard Green, Ankur Vora, Mira Lutfi, Michael Figurnov, Andrew Cowie, Nicole Hobbs, Pushmeet Kohli, Gerard Kleywegt, Ewan Birney, Demis Hassabis, and Sameer Velankar. 2021. 'AlphaFold Protein Structure Database: massively expanding the structural coverage of protein-sequence space with high-accuracy models', *Nucleic Acids Research*, 50: D439-D44.

Wilson, Cyrus A, Julia Kreychman, and Mark Gerstein. 2000. 'Assessing annotation transfer for genomics: quantifying the relations between protein sequence, structure and function through traditional and probabilistic scores', *Journal of Molecular Biology*, 297: 233-49.

Zheng, Wei, Qiqige Wuyun, Yang Li, Chengxin Zhang, P. Lydia Freddolino, and Yang Zhang. 2024. 'Improving deep learning protein monomer and complex structure prediction using DeepMSA2 with huge metagenomics data', *Nature Methods*, 21: 279-89.

Zhou, H. X., and X. Pang. 2018. 'Electrostatic Interactions in Protein Structure, Folding, Binding, and Condensation', *Chemical Reviews*, 118: 1691-741.

Reviewer #3

In this article, the genomic traits of different *Vibrio cholerae* strains were compared. A number of isolates from patients was sequenced. They then identified core genes and SNPs present in the different lineages through genomic analyses, genome-scale modeling, and protein-protein interaction networks. They also identified genetic determinants associated with infection symptoms through machine learning. Overall, this is an interesting work. The manuscript is also well-written and easy to follow. I would recommend it for publication.

I have the following comments:

1. Line 235-237: without looking up the supplemental table, it is not clear why the metabolite yield was computed and for which metabolites. The Methods section also does not clarify this. Generally, a concluding sentence on what the FBA and FVA results mean would be helpful.

We thank the Reviewer for this comment, and we agree that a better explanation would improve the manuscript. We have now added a clearer explanation of the rationale behind calculating the metabolite yield, and for which metabolites this was done, in the Results lines 220-236, 268-278, 456-459 and 514-516, as well as improving the description of the analysis in the Methods lines 908-919 and below. Specifically, the analysis of metabolite yield was conducted on all the 1,741 metabolites present in the GSM model (iAM-Vc-960), with those that reduced to zero in response to a gene knockout being reported in Datasets S11, S12, S17 and S18. The choice to include all metabolites, which followed the pipeline by Percy *et al* 2021, was to ensure that the results would not be unduly biased by assumptions regarding metabolites that may be expected to be affected.

Additionally, as suggested by the Reviewer, we have added a concluding sentence on what the FBA and the FVA results mean in both the sections: “Genetic and temporal differentiation of *V. cholerae* BD-1.2 and BD-2 lineages correlate with SNPs on coding and non-coding regions, and accessory genes” and “Machine learning unravels correlations between genomic determinants and clinical symptoms in humans”, Results lines 268-278 and 514-516.

Results lines 220-236: “Moreover, for these genes we sought to better understand their role by examining their effects on *V. cholerae* growth rate biochemical networks and metabolites production in the networks. As the effect of mutations/gene knockouts cannot always be observed as change in growth rate (due to the redundancy of the reactions in metabolic networks of bacteria), it can be useful to also consider the changes in metabolite yield. Changes in metabolite yield have been found to correlate with changes in the virulence, persistence, and fitness of some organisms (Somerville *et al*, 2003). Furthermore, *V. cholerae* are capable adapting to ecological niches by altering the metabolites they excrete to create a more favourable environment for *V. cholerae* and/or a less favourable environment for other species competing for the same resources (Keating *et al* 2000, Kostiuk *et al* 2023). Mutations disrupting larger numbers of metabolite yields may be suggestive of a larger systems-level impact on bacterial metabolic function. Therefore, gene essentiality, flux variability analysis (FVA) and flux balance analysis (FBA) were used to predict, through gene knockouts, the essentiality and the effects of the identified genetic determinants on the growth rates of *V. cholerae*, and also used to further explore their influence on metabolite yield. The latter was done by assessing the influence on metabolite flow within the complete metabolic network of

V. cholerae, encompassing all known metabolites and metabolic reactions (see methods). In this analysis it was important to consider all reactions and metabolites in the model rather than focussing on a subset, as doing so ensures no undue bias or assumptions underlie the results.”

Results lines 268-278: “In summary, a total of 15 genes found to underly the genetic and temporal differentiation of *V. cholerae* BD-1.2 and BD-2 lineages, were also found to significantly alter the growth, reaction flux, or metabolite yield of *V. cholerae* when knocked down, either in the generalised iAM-Vc960 GSM model or in the draft strain-specific models. Of interest was the gene *clcA*, which showed differences in both flux span and metabolite changes between lineages in the draft GSM models. The FVA and FBA results indicate that the genes identified by machine learning as strongly associated with the severity of symptoms play important metabolic roles. Disruption of these functions could potentially affect bacterial growth or metabolic output, which may contribute to the survival and dominance of one lineage over another. Although our analysis cannot pinpoint a single SNP as responsible for the loss of metabolic function, it suggests that an accumulation of SNPs or gene losses could collectively lead to metabolic changes. We observe the potential for metabolic alterations driven by multiple mutations (SNPs).”

Results lines 456-459: “As in our previous lineage analysis, we sought to better understand the importance of the genes which had been found to better correlate with the severity of the symptoms. We examined for those genes that were metabolic, through FVA and FBA, the effects of such genes on growth rate (gene essentiality), and beyond that, their influence on metabolite yield and reaction flux.”

Results lines 514-516: “In summary, in relation to gene essentiality, reaction flux and metabolite yield, our results show that *gshB* and *dapF* make interesting candidates for further analysis, as knockout models of these genes predict significant changes to the bacterial metabolic function.”

Methods lines 908-919: “For the FBA analysis, a drain reaction (i.e., a reaction that consumes the metabolite of interest) was added to the GSM model for each metabolite. The maximum theoretical yield of each metabolite was calculated by setting its corresponding drain reaction as the objective function, with glucose as the only carbon source in aerobic minimal M9 medium conditions. All metabolites contained within the model were considered in the FBA analysis. In iAM-Vc960 this was 1,741 different metabolites, whilst in the draft strain-specific GSM models the number of metabolites spanned the range 1321-1433. The simulations were carried out for the wild-type model and each gene knockout model. A gene knockout was considered to significantly affect metabolite yield if the yield of at least one metabolite was reduced to zero, given that it was non-zero in the wildtype. For each of the selected genes of interest, molecular function, pathways and biological processes were taken from the BioCyc database (Karp *et al.* 2019) using the SMART tables for *Vibrio cholerae* O1 biovar El Tor strain N16961. These were added to **Datasets S11, S12, S17 and S18** to give context to the analysis results”

Karp, P. D., R. Billington, R. Caspi, C. A. Fulcher, M. Latendresse, A. Kothari, I. M. Keseler, M. Krummenacker, P. E. Midford, Q. Ong, W. K. Ong, S. M. Paley, and P. Subhraveti. 2019. 'The BioCyc collection of microbial genomes and metabolic pathways', *Briefings in Bioinformatics*, 20: 1085-93.

Keating, T. A., Marshall, C. G., & Walsh, C. T. (2000). Vibriobactin biosynthesis in *Vibrio cholerae*: VibH is an amide synthase homologous to nonribosomal peptide synthetase condensation domains. *Biochemistry*, 39(50), 15513–15521. <https://doi.org/10.1021/bi001651a>

Kostiuk B, Becker ME, Churaman CN, Black JJ, Payne SM, Pukatzki S, Koestler BJ. 2023. *Vibrio cholerae* Alkalizes Its Environment via Citrate Metabolism to Inhibit Enteric Growth In Vitro. *Microbiol Spectr* 11:e04917-22.

<https://doi.org/10.1128/spectrum.04917-22>

Somerville GASaid-Salim B, Wickman JM, Raffel SJ, Kreiswirth BN, Musser JM, 2003. Correlation of Acetate Catabolism and Growth Yield in *Staphylococcus aureus*: Implications for Host-Pathogen Interactions. *Infect Immun* 71:4724-4732.

2. The authors used the same genome-scale reconstruction of one *V. cholerae* strain for all simulations despite analyzing a multitude of isolates from different *V. cholerae* strains. Why did they not generate isolate-specific reconstructions for the 129 genomes using iAM-Vc960 as a template?

We thank the Reviewer for this question (a similar one was made by Reviewer 4, question 2). We originally opted to use a published model as it already had been experimentally validated. However, we agree that strain-specific models can offer additional insight into the metabolism of the strains. Thus, we have now included a strain-specific analysis into our manuscript. We have used the software CarveMe to generate 129 strain-specific models using protein fasta files generated by Prokka v1.14.6. CarveMe, also suggested by Reviewer 4, was chosen as it is the most utilised, ahead of gapseq and Bactabolize (465 citations for CarveMe, vs 118 for gapseq, and 2 for Bactabolize). CarveMe was run using the CPLEX solver and a Gram negative template, with gap filling for LB and M9 media using the command: 'carve input.faa --gapfill M9, LB -u gramneg --solver cplex --output model.xml'. The strain-specific models were then subjected to three analyses. Respectively:

- (i) Gene Essentially analysis: for each gene identified by the statistical analysis of lineages and machine learning analysis of clinical symptoms, we performed knock-down in each of the 129 models and compared the biomass to wildtype to assess whether the gene was essential for the model.
- (ii) Flux variability analysis: for each gene selected by the statistical analysis of lineages and machine learning analysis of clinical symptoms, we performed knockdowns in each of the 129 models and compared the flux span of each reaction to the wild type, identifying significant changes in reaction flux.
- (iii) Flux balance analysis. for each gene selected by the statistical analysis of lineages and machine learning analysis of clinical symptoms), we performed a knockdown in each of the 129 models. We then compared the metabolite yields of each model to those of the wild type.

Overall, the number of genes and reactions in the strain-specific models was slightly higher than the genes in iAM-vc960 and varied between 973 – 988 genes and 1989 – 2163 reactions, with a mean of 984 genes and 2130 reactions. The number of metabolites in the strain-specific models was lower than that of the iAN-vc960 model with a range of 1321 – 1433 and mean 1411. The number of genes, reactions and metabolites statistically differed between BD-1.2 and BD-2 lineages (p -values < 0.0001, Mann Whitney U test).

For the lineage-based analysis, we used as input the core genome genes mapped to the non-synonymous SNPs found to be exclusively present in one lineage (i.e. present in BD-1.2 and absent in BD-2 or conversely, present in BD-2 and absent in BD-1.2). We found 14 metabolic genes which were present in the strain-specific GSM models (*clcA*, *mak*, *suhB*, *murl*, *glmM*, *appC*, *argG*, *ftsI*, *licH*, *phhA*, *dltA_1*, *putA*, *ycbB*, *hudF_2*). Of these 10 were in the generalised

model (*clcA*, *mak*, *suhB*, *murl*, *glmM*, *appC*, *argG*, *ftsI*, *licH*, *phhA*) and four were not (*dltA_1*, *putA*, *ycbB*, *hudF_2*). Three genes present in the generalised model (*dsbD*, *cysG_1* and *cob*) were not present in the strain-specific models. Comparing the gene essentiality results, *murl* was found to be essential in minimal media only in the generalised model. In the strain-specific models, it was essential in both rich and minimal media in a small proportion of models (n=16, 12%) and non-essential in both rich and minimal media in all other models. Experimental data from literature (Karp *et al.* 2019) shows essentiality of *murl* in the reference strain N16961 indicating some metabolic behavioural differences in our strains regarding this gene. Similarly, *glmM* was essential only in minimal media in the generalised model but essential in rich and minimal media in all the strain-specific models, consistent with its behaviour in the N16961 (Karp *et al.* 2019). Finally, *clcA*, which was essential in the generalised model was non-essential in all the strain-specific models, consistent with experimental results for the reference strain, where *clcA* is non-essential (Karp *et al.* 2019). In the flux variability analysis, which assessed whether gene knockouts significantly alter the reaction fluxes through the GSM models, a few changes were found in the strain-specific models compared to the generalised model. Knockouts of *suhB*, *phhA* and *licH* resulted in significant flux changes in the strain-specific models but not in the generalised models. Interestingly the gene *clcA*, which generated significant flux changes in the generalised model, also caused significant flux changes in most of the BD-2 strain-specific models (91%) but only in the small proportion of the BD-1.2 models (5%).

Regarding metabolite yield changes assessed by FBA, *murl* and *mak* knockouts affected metabolite yield in the generalized model but not in strain-specific models. Conversely, *suhB* and *clcA* knockouts impacted metabolite yield in the strain-specific models but not in the generalised model.

Regarding the clinical symptom-based analysis, 11 over the genes identified by machine learning were identified as metabolic genes in the strain-specific models (*add*, *dapF*, *dcuH*, *gshB*, *hpt*, *pckA*, *pepN*, *cysG_2*, *padC*, *tufB*, *tufB_2*). Among these, seven were also present in the generalised model (*add*, *dapF*, *dcuH*, *gshB*, *hpt*, *pckA*, *pepN*) while four were (*cysG_2*, *padC*, *tufB*, *tufB_2*) unique to the strain-specific models. Two genes that were present in the generalised model (*cdgL* and *fabH1*) are absent in strain-specific models.

Comparing the gene essentiality analysis between strain-specific models and the generalised model, *dapF* was found essential in rich media for 93% of strain-specific models, whereas it was non-essential in the generalised model. This finding aligns with experimental data (Karp *et al.* 2019), showing this gene as essential for the reference N16961, and indicating consistency between strain-specific model results and experimental observations. In flux variability analysis, all genes showing significant flux changes in the generalized model exhibited similar changes in strain-specific models. For the metabolites' yields analysis changes, while *dapF* knockouts showed significant metabolite yield changes in both generalized and strain-specific models, *gshB* knockouts did not show any metabolite yield changes in strain-specific models.

We thank the Reviewer for suggesting this additional analysis which has highlighted significant differences across strains in our cohort. Comparison between the curated generalised model (iAM-Vc960) and draft strain-specific models were added as Supplementary Note 5, manuscript results lines 256-267 and 493-513 and in the methods lines 883-919; all edits are also shown below. Details of the results obtained with the strain-specific models were added to Datasets S12 and S18.

Supplementary Note 5: "In addition to using the generalised *V. cholerae* O1 GSM model (iAM-vc960) to analyse the metabolic functions of our strains, draft strain-specific models were generated for each isolate using CarveMe (Machado *et al.* 2018). Overall, the number of

genes and reactions in the strain-specific models was slightly higher than the genes in iAM-vc960 and varied between 973 – 988 genes and 1989 – 2163 reactions, with a mean of 984 genes and 2130 reactions. The number of metabolites in the strain-specific models was lower than that of the iAM-Vc960 model with a range of 1321 – 1433 and mean 1411. The number of genes, reactions and metabolites statistically differed between BD-1.2 and BD-2 lineages (p-values < 0.0001, Mann Whitney U test).

For the lineage-based analysis, we used as input the core genome genes mapped to the non-synonymous SNPs found to be exclusively present in one lineage (i.e. present in BD-1.2 and absent in BD-2 or conversely, present in BD-2 and absent in BD-1.2). We found 14 metabolic genes which were present in the strain-specific GSM models (*clcA*, *mak*, *suhB*, *murl*, *glmM*, *appC*, *argG*, *ftsI*, *licH*, *phhA*, *dltA_1*, *putA*, *ycbB*, *hudF_2*). Of these 10 were in the generalised model (*clcA*, *mak*, *suhB*, *murl*, *glmM*, *appC*, *argG*, *ftsI*, *licH*, *phhA*) and four were not (*dltA_1*, *putA*, *ycbB*, *hudF_2*). Three genes present in the generalised model (*dsbD*, *cysG_1* and *cob*) were not present in the strain-specific models. Comparing the gene essentiality results, *murl* was found to be essential in minimal media only in the generalised model. In the strain-specific models, it was essential in both rich and minimal media in a small proportion of models (n=16, 12%) and non-essential in both rich and minimal media in all other models. Experimental data from literature (Karp *et al.* 2019) shows essentiality of *murl* in the reference strain N16961 indicating some metabolic behavioural differences in our strains regarding this gene. Similarly, *glmM* was essential only in minimal media in the generalised model but essential in rich and minimal media in all the strain-specific models, consistent with its behaviour in the N16961 (Karp *et al.* 2019). Finally, *clcA*, which was essential in the generalised model was non-essential in all the strain-specific models, consistent with experimental results for the reference strain, where *clcA* is non-essential (Karp *et al.* 2019). In the flux variability analysis, which assessed whether gene knockouts significantly alter the reaction fluxes through the GSM models, a few changes were found in the strain-specific models compared to the generalised model. Knockouts of *suhB*, *phhA* and *licH* resulted in significant flux changes in the strain-specific models but not in the generalised models. Interestingly the gene *clcA*, which generated significant flux changes in the generalised model, also caused significant flux changes in most of the BD-2 strain-specific models (91%) but only in the small proportion of the BD-1.2 models (5%).

Regarding metabolite yield changes assessed by FBA, *murl* and *mak* knockouts affected metabolite yield in the generalised model but not in strain-specific models. Conversely, *suhB* and *clcA* knockouts impacted metabolite yield in the strain-specific models but not in the generalised model.

Regarding the clinical symptom-based analysis, 11 over the genes identified by machine learning were identified as metabolic genes in the strain-specific models (*add*, *dapF*, *dcuH*, *gshB*, *hpt*, *pckA*, *pepN*, *cysG_2*, *padC*, *tufB*, *tufB_2*). Among these, seven were also present in the generalised model (*add*, *dapF*, *dcuH*, *gshB*, *hpt*, *pckA*, *pepN*) while four were (*cysG_2*, *padC*, *tufB*, *tufB_2*) unique to the strain-specific models. Two genes that were present in the generalised model (*cdgL* and *fabH1*) are absent in strain-specific models.

Comparing the gene essentiality analysis between strain-specific models and the generalised model, *dapF* was found essential in rich media for 93% of strain-specific models, whereas it was non-essential in the generalised model. This finding aligns with experimental data (Karp *et al.* 2019), showing this gene as essential for the reference N16961, and indicating consistency between strain-specific model results and experimental observations. In flux variability analysis, all genes showing significant flux changes in the generalised model exhibited similar changes in strain-specific models. For the metabolites' yields analysis changes, while *dapF* knockouts showed significant metabolite yield changes in both

generalized and strain-specific models, *gshB* knockouts did not show any metabolite yield changes in strain-specific models.”

Results Lines 256-267: “To further elucidate the metabolic differences between the BD-1.2 and BD-2 lineages, we repeated our previous analyses done on the generalized model using strain-specific models automatically generated by CarveMe (Machado *et al.* 2018). Gene essentiality analysis concurred with the general model (iAM-Vc960), with only a small number of differences (**Dataset S12**). The effect of *murl* gene knockouts differed between lineages, proving non-essential in 94% of BD-1.2 lineage models but only in 76% of BD-2 lineage models. Flux variability analysis of the individual models revealed that *clcA* knockouts led to significant changes in the flux span of the CLt3_2pp reaction, which controls chloride transport, in 96% BD-2 models compared to just 5% of BD-1.2 models. The *clcA* gene has been linked to bacterial acid resistance and it has been suggested that changes to the expression/repression of this gene may help facilitate survival during movement through the intestinal tract (Cakar *et al.* 2018). Similarly, flux balance analysis indicated that metabolite yield was changed differently across lineages in response to knocking out *clcA*, with the metabolite yield of chloride reduced to 0 in 95% of BD-1.2 isolates.”

Results Lines 493-513: “To further investigate the link between metabolic gene variations and the clinical symptoms observed in different strains, we utilized draft strain-specific models generated with CarveMe (Machado *et al.* 2018). The gene essentiality analysis results were largely consistent with those of the general model (iAM-Vc960), with only a few differences noted (**Dataset S18**). The effect of *dapF* gene knockouts varied between models with the gene being essential in 93% (n=20) and non-essential in 7% (n=9) of the models. Comparing symptoms between the ‘essential’ and ‘non-essential’ groups, dehydration was significantly more severe in the ‘non-essential’ group (Fisher exact test p value =0.05). All strains in this group exhibited severe dehydration, suggesting a link between non-essentiality of the *dapF* gene and the severity of *V. cholerae* symptoms. In relation to this, the flux balance analysis revealed changes in metabolite yields associated with the genes *dapF* and *cysG_2* across all strain-specific models. For *dapF*, altered metabolite yields were predominantly observed in strains where *dapF* was essential, while knocking out *dapF* in non-essential models had minimal impact on the metabolite yields of murein-related metabolites. This indicates metabolic adaptations linked to bacterial survival in these strains, potentially contributing to more severe disease outcomes. Additionally, knocking out the *padC* gene resulted in significant changes in metabolite yields only in the NGICDV-066 strain. Although conclusions drawn from a single strain are limited, it is notable that this isolate exhibited the most severe clinical symptoms across all measured symptoms, except for the duration of diarrhoea (presence of vomiting, presence of abdominal pain, number of stools (21+ times), presence of severe dehydration, duration of diarrhoea 1-3 days). Flux variability analysis in individual models indicated consistent behaviour across all strain-specific models regarding gene knockouts associated with clinical symptoms. Specifically, five gene knockouts (*add*, *dapF*, *gshB*, *padC*, *pckA*) showed significant flux span changes in all models.”

Methods Lines 883-919: “Genome-scale metabolic model

All simulations were performed using the Python cobra toolkit v0.26.2. The analysis was conducted on both a manually curated and validated model of *V. cholerae* O1 N16961, iAM-Vc960, taken from (Abdel-Haleem *et al.* 2020) and on automatically generated draft strain-specific GSM models. The strain-specific draft models were generated using CarveMe (Machado *et al.* 2018). CarveMe was run using the CPLEX solver and gram negative template, with gap filling for LB and M9 media using the command: ‘carve input.faa --gapfill M9,LB -u gramneg --solver cplex --output model.xml’. Gene essentiality, FVA and FBA analyses as described below were conducted on genes of interest in the generalised iAM-Vc960 and in

each of the 129 draft strain-specific models, based on the analysis pipeline in (Pearcy *et al.* 2021).

For all gene essentiality, FBA and FVA analyses, a knockout model for each gene of interest was constructed by blocking all corresponding reactions to zero, given that the reaction is not catalysed by an isozyme. We considered the essentiality of a gene under both rich medium conditions and M9 minimal medium conditions. To mimic rich medium conditions, the model was constrained to allow all carbon sources into the system, with a fixed uptake rate of 1 mmol/gDCW/h. If a feasible solution exists, while maximizing the biomass equation as the objective function, then the knockout of the gene was not essential. To mimic M9 minimal medium conditions, the model was constrained so one individual carbon source had a maximum uptake of 10 mmol/gDCW/h. This simulation (minimal medium condition) was repeated for each carbon source in the model. The genes whose corresponding knockout model achieved a growth rate of 0.0001 h⁻¹ or less were considered essential. Flux variability analysis (FVA) was applied to the wild-type model and each knockout model using the cobra toolbox in python (Ebrahim *et al.* 2013). FVA calculates the minimum and maximum flux through each reaction in the model, given a set of constraints, resulting in the range of possible fluxes for each reaction (flux span). FVA was simulated using glucose as the only carbon source in aerobic minimal M9 medium conditions. Note that reaction loops in the solution were not allowed. A gene knockout was considered to significantly affect the flux if the flux span of at least one reaction was changed by greater than 10% compared to the wildtype solution. For the FBA analysis, a drain reaction (i.e., a reaction that consumes the metabolite of interest) was added to the GSM model for each metabolite. The maximum theoretical yield of each metabolite was calculated by setting its corresponding drain reaction as the objective function, with glucose as the only carbon source in aerobic minimal M9 medium conditions. All metabolites contained within the model were considered in the FBA analysis. In iAM-Vc960 this was 1,741 different metabolites, whilst in the draft strain-specific GSM models the number of metabolites spanned the range 1321-1433. The simulations were carried out for the wild-type model and each gene knockout model. A gene knockout was considered to significantly affect metabolite yield if the yield of at least one metabolite was reduced to zero, given that it was non-zero in the wildtype. For each of the selected genes of interest, molecular function, pathways and biological processes were taken from the BioCyc database (Karp *et al.* 2019) using the SMART tables for *Vibrio cholerae* O1 biovar El Tor strain N16961. These were added to **Datasets S11, S12, S17 and S18** to give context to the analysis results.”

Abdel-Haleem, Alyaa M, Vaishnavi Ravikumar, Boyang Ji, Katsuhiko Mineta, Xin Gao, Jens Nielsen, Takashi Gojobori, and Ivan Mijakovic. 2020. 'Integrated metabolic modeling, culturing, and transcriptomics explain enhanced virulence of *Vibrio cholerae* during coinfection with enterotoxigenic *Escherichia coli*', *mSystems*, 5: e00491-20.

Ebrahim, Ali, Joshua A Lerman, Bernhard O Palsson, and Daniel R Hyduke. 2013. 'COBRApy: constraints-based reconstruction and analysis for python', *BMC Systems Biology*, 7: 1-6.

Karp, P. D., R. Billington, R. Caspi, C. A. Fulcher, M. Latendresse, A. Kothari, I. M. Keseler, M. Krummenacker, P. E. Midford, Q. Ong, W. K. Ong, S. M. Paley, and P. Subhraveti. 2019. 'The BioCyc collection of microbial genomes and metabolic pathways', *Briefings in Bioinformatics*, 20: 1085-93.

Machado, Daniel, Sergej Andrejev, Melanie Tramontano, and Kiran Raosaheb Patil. 2018. 'Fast automated reconstruction of genome-scale metabolic models for microbial species and communities', *Nucleic Acids Research*, 46: 7542-53.

Pearcy, Nicole, Yue Hu, Michelle Baker, Alexandre Maciel-Guerra, Ning Xue, Wei Wang, Jasmeet Kaler, Zixin Peng, Fengqin Li, Tania Dottorini, and Xiaoxia Lin. 2021. 'Genome-scale

metabolic models and machine Learning reveal genetic determinants of antibiotic resistance in *Escherichia coli* and unravel the underlying metabolic adaptation mechanisms', mSystems, 6: e00913-20.

3. The code available at <https://github.com/tan0101/VibrioCARE> does not seem to include the genome-scale modeling simulations.

We apologise that these scripts were not made available in the manuscript. The analysis script as well as the new draft strain-specific models have been added to the GitHub folder: <https://github.com/tan0101/VibrioCARE>.

Reviewer #3 (Remarks on code availability):

The code for flux balance analysis and flux variability analysis simulations does not seem to be present in the repository. The code as well as the simulated media need to be uploaded.

We apologise that these scripts were not made available in the manuscript. The media used was changed within the python code from the default, rather than uploaded as another file so there is no separate simulated media file. The analysis script as well as the new draft strain-specific models have been added to the GitHub folder: <https://github.com/tan0101/VibrioCARE>.

Reviewer #4

In their work, Maciel-Guerra et al. study virulence determinants in a collection of clinical isolates of *V. cholerae*. They identify key genomic differences between two prevalent lineages and associated them both with clinical characteristics of infection and characterize them using a variety of in silico approaches. Their work is of high relevance for improving our understanding of the determinants underlying genome evolution in pathogens. I particularly like their approach of functionally interpreting the relevance of genomic variants in the different isolates using genome-scale metabolic modelling and structural analysis.

Major points:

1) The authors use machine learning to associated clinical features with variants in the genomes of the individual isolates. I'm not sure if this approach is really adequate since it probably does not allow to account for potential confounders such as location or patient phenotypes. The authors should clarify or potentially repeat/complement the analysis with a statistical approach that can deal with confounders.

We sincerely appreciate the reviewer for this valuable suggestion, as it significantly contributed to enhancing the manuscript. We agree that in machine learning, confounding effects are crucial as they can distort the understanding and performance of predictive models. In consideration of this, we have now complemented our machine learning pipeline with a statistical approach to account for potential confounders. A confounding variable influences both the independent variables (such as input features) and the dependent variable (such as the model's predictions), leading to misleading conclusions about the model's efficacy (Whalen *et al.* 2022; Jager *et al.* 2008). Confounding variables introduce issues such as invalid correlations, increased variance, and biases that can skew model outcomes, resulting in predictions that do not generalize well to new data (Jager *et al.* 2008).

To address confounding effects, randomization is a fundamental strategy. For example, data resampling techniques, like cross-validation, involve randomizing the training and test datasets to obtain a more accurate estimate of model performance across different subsets of data (Gallicchio *et al.* 2017; Brownlee 2018; Jager *et al.* 2008). Repeated experiments with different random seeds ensure that results are not dependent on a particular initialization or sequence of operations. By varying these elements, the evaluation process can more reliably reflect the model's true capabilities and generalizability. Our initial pipeline already used a nested cross validation to select the best hyperparameters and calculate the performance using a stratified k-fold cross validation. Nested cross-validation provides a robust evaluation framework that helps prevent overfitting and ensures more reliable estimates of model performance. The importance of nested cross-validation lies in its ability to separate the processes of model evaluation and hyperparameter optimization, thus providing a more accurate assessment of how a model will perform on unseen data. Nested cross-validation uses two loops: an inner loop for hyperparameter tuning and an outer loop for model evaluation. In the inner loop, the data is split into training and validation sets multiple times to find the best hyperparameters. These optimized hyperparameters are then used in the outer loop, where the model is evaluated on a separate test set. This ensures that the evaluation of the model is not biased by the data used for hyperparameter tuning. By using nested cross-validation, we can obtain a more accurate and unbiased estimate of model performance, making it an essential practice for robust model validation. It ensures that the chosen model and hyperparameters generalize well to new, unseen data, thereby improving the reliability and applicability of the model in real-world scenarios. This methodology is particularly important in complex models with many hyperparameters, where the risk of overfitting is high, and accurate performance assessment is critical for model selection and deployment.

The choice of algorithms also introduces potential confounding effects. Different machine learning algorithms have varying assumptions, capabilities, and biases (Gallicchio *et al.* 2017). For example, random forest can handle non-linear relationships and interactions between variables, while linear regression assumes linearity and independence among predictors. Selecting an appropriate algorithm based on the data characteristics is essential to minimize confounding effects. Moreover, algorithm initialization (Brownlee 2018), such as the SMOTE (Synthetic Minority Over-sampling Technique) approach, can significantly impact the learning process and final model performance. Randomization in these initializations ensures that the model does not overfit to a particular sequence or initialization, enhancing its robustness. Based on this, we have now updated the way we use the SMOTE approach. Initially we were setting a single random seed for its initialization, now we are screening 1000 different random seeds so different initialization can be performed. Next, we evaluate how many times each feature is selected (based on a Fisher Exact test as previously done) and if the feature is selected over 75% of the times, we deem it important. Next, we look at all the different 1000 initializations to see which one contains all the features deemed important and randomly select one of such initializations to be used as a base for the prediction models.

Hyperparameter tuning is another critical aspect where confounding effects can emerge. Hyperparameters, such as learning rates, regularization parameters, the number of estimators and others, need to be carefully selected to optimize model performance (Gallicchio *et al.* 2017). Grid search is a common technique to explore different hyperparameter combinations. Based on this, we have included additional hyperparameters to better tune the models:

- Logistic Regression: inverse of regularization strength $C = [0.0001, 0.001, 0.01, 0.1, 1, 10, 100, 1000, 10000]$;
- Linear SVM: penalty parameter of the hinge loss error $C = [0.0001, 0.001, 0.01, 0.1, 1, 10, 100, 1000, 10000]$;
- Random Forests, Extra Trees and Adaboost: Number of estimators = $[2, 4, 8, 16, 32, 64, 128, 256]$;

- Non-linear SVM with RBF kernel: γ (RBF kernel coefficient) = [0.0001, 0.0001, 0.001, 0.01, 0.1, 1] and C (L2 penalty parameter) = [0.0001, 0.001, 0.01, 0.1, 1, 10, 100, 1000, 10000];
- XGBoost: Number of estimators = [2, 4, 8, 16, 32, 64, 128, 256] and learning rate = [0.0001, 0.001, 0.01, 0.1, 1].

Furthermore, it is possible to remove independent variables that are associated with a confounding effect, which is also associated with the dependent variable. Statistical methods can help detect these associations by identifying variables that show strong correlations with both the dependent variable and other independent variables, indicating potential confounding effects (Goren *et al.* 2021). This approach ensures that the final model remains valid and interpretable, accurately reflecting the relationships between key variables without undue influence from confounders.

Based on this, we propose a selection process which first analyses if the dependent variable is correlated to a confounder using a Chi-square test with Bonferroni correction and p-value < 0.01 (used to determine if there is an association between two categorical variables). If there is a correlation, we then proceed to check if there are features (independent variables) also correlated with the same confounder using again a Chi-square test with the same p-value. If there is a correlation, the features are removed. Below we show for each of the symptom prediction models used in this work, the correlation between them and the confounding effects (location of samples, year of collection, serology of *V. cholerae*, age of the patient and sex of the patient):

- Abdominal Pain: none of the potential confounders (age, sex, location, collection year and serology) were statistically correlated to this phenotype → no correction needed.

Factor	chi2
Location	0.03659
Year	0.065165
Serology	0.274619
Age	0.419484
Sex	0.719884

- Diarrhoea duration <1day vs 1-3 days: four of the five potential confounders (age, sex, location, and serology) were not statistically correlated to this phenotype; however Collection year was → correction needed (remove the feature). The removed factor is indicated in red.

Factor	chi2
Year	0.000343
Sex	0.005304
Serology	0.008613
Location	0.413819
Age	0.674257

- Number of stools 11-15 times vs 21+ times: none of the potential confounders (age, sex, location, collection year and serology) were statistically correlated to this phenotype → no correction needed.

Factor	chi2
Year	0.003607

Serology	0.021058
Location	0.051919
Age	0.524279
Sex	0.871291

- Vomiting: none of the potential confounders (age, sex, location, collection year and serology) were statistically correlated to this phenotype → no correction needed.

Factor	chi2
Location	0.1342
Serology	0.18635
Age	0.497593
Year	0.704445
Sex	0.953786

- Number of stools 11-15 times vs 16-20 times: none of the potential confounders (age, sex, location, collection year and serology) were statistically correlated to this phenotype → no correction needed.

Factor	chi2
Year	0.070809
Location	0.073439
Serology	0.140716
Age	0.267341
Sex	1

- Dehydration moderate vs severe: none of the potential confounders (age, sex, location, collection year and serology) were statistically correlated to this phenotype → no correction needed.

Factor	chi2
Location	0.039965
Year	0.096318
Age	0.58622
Sex	0.906549
Serology	1

By addressing confounding effects through careful data preparation, algorithm selection, initialization techniques, hyperparameter tuning and feature selection, we can develop more reliable and generalizable machine learning models. This comprehensive approach is crucial for achieving robust performance in real-world applications, where data variability and complexity are inherent challenges. Therefore, we have now updated our pipeline (Figure 1 below) to: i) randomize the initialization of the SMOTE approach to better select the features; ii) increased the number of hyperparameters used in the learning algorithms; iii) remove features that are correlated to confounding effects.

Figure 1. Flow diagram showing machine learning pipeline including data (green), pre-processing steps (yellow) and classification (blue).

Finally, the revised analysis resulted in a small change to the outputs of the GSM analysis, with the gene *oppA* no longer used as an input and removed from the analysis, and the metabolic gene *hpt* now included. However, the gene *hpt* was not essential nor significant in any of the GSM analyses. Dataset S17 (previously Table S10) has been updated to account for these changes.

We have updated the results (lines 339-437), methods (lines 965-994), and Figures 4, 5, S18, and S19 (previously S17 and S18), as well as Datasets S15 and S16 (previously Tables S8 and S9), to incorporate the results obtained from the new analysis conducted using a pipeline that integrates confounding effects. Dataset S14 was included in the manuscript showing the chi-square test of independence results between the clinical symptoms phenotypes and the confounding effects.

Results lines 339-437: “The symptom prediction models were developed with built-in robustness to potential confounding factors. Specifically, the following list of variables was

initially considered as potentially having confounding effects: year of collection, location of patient, sex of patient, age of patient and serology of *V. cholerae*. Each potential confounder was tested for correlation to the symptom being targeted by the prediction model. If the potential confounder was found correlated to the symptoms (hence moving from potential to proven confounder), then any other input variable also found correlated with the same confounder would be eliminated from the prediction model. All the correlation tests between inputs and symptoms, as well as between inputs themselves, were run using two-sided Chi-square tests. Further, possible confounding effects related to random initialisation parameters of SMOTE (see methods) were contained by running SMOTE multiple times.

The development and optimisation of each symptom prediction model powered by machine learning was based on running a comparative analysis of the predictive performances of different machine learning algorithms, namely: linear support vector machine (linear SVM), non-linear SVM with radial basis function (RBF SVM), random forest, extra-tree classifier and logistic regression) and two meta-methods (Adaboost and XGBoost). For each algorithm, multiple configurations of the hyperparameters of the learning algorithms were tested. A nested cross validation approach was used to select the best hyperparameters, based on randomly selecting different training and test sets, and using stratified k-fold cross validation metric. Finally, Friedman and Nemenyi tests were used to statistically compare and select the best performing algorithm for each prediction model (see **Methods** section).

In the end, based on a two-sided Chi-square test of independence (p-value < 0.01), the models for abdominal pain, vomit, number of stools 11-15 times vs. 21+ times, number of stools 11-15 times vs. 16-20 times, dehydration moderate vs severe were found immune to confounding effects due to year of collection, location of patient, sex of patient, age of patient and serology of *Vibrio cholerae*. The prediction model: diarrhoea duration <1day vs 1-3 days was found immune to confounding effects due to age of patient, sex of patient, location of patient, and serology of *Vibrio cholerae*. However, the prediction model was found to be influenced by year of collection; therefore, the inputs that were also correlated to year of collection were removed from the analysis (**Dataset S14**). Moreover, we were able to successfully develop six binary symptom prediction models featuring adequate prediction performance levels. These were dedicated to predicting the following binary phenotypical outcomes: i) stools 11-15 times vs. 16-20 times; ii) stools 11-15 times vs. 21+ times; iii) moderate vs. severe dehydration; iv) diarrhoea duration <1 day vs. 1-3 days; v) presence vs absence of vomit; and vi) presence vs absence of abdominal pain (**Dataset S13**). The remaining binary predictors were discarded for not performing adequately, either because of unbalanced available sets of observations (needed for training the supervised ML models), or because of more challenging separability of the phenotypes given the selected inputs (no features were statistically significant based on the Fisher exact test). Among the tested machine learning algorithms mentioned earlier, logistic regression was identified as the best performing one by the Friedman F-test and the Nemenyi post-hoc analysis (**Fig. S18**). Of the six binary prediction models, four had an AUC greater than 0.9, **Fig. 4. Dataset S15** indicates the performance metrics obtained by all binary predictors for each clinical symptom. **Fig. 4** and **S19** show the performance results for the logistic regression classifier.

Figure 4. Supervised machine learning pipeline accurately successfully predicts the clinical manifestations of hospitalized patients from the genomic determinants extracted from BD-1.2 isolates, collected among the same hospitalised patients. (A) Flow diagram showing machine learning pipeline including data (green), pre-processing steps (yellow) and classification (blue). (B) Machine learning performance results measured by the area under the curve (AUC) from 30 training runs for clinical symptom combination. The results shown are for the best classifier Logistic Regression, as defined by the Nemenyi test (**Fig. S18**). (C) Number of features (accessory genes, core genome SNPs and intergenic SNPs) selected for each symptom. Predictive models were generated for six different clinical symptoms (X axis): abdominal pain; dehydration “moderate” vs “severe”; duration of diarrhoea <1 day vs. 1-3 days; number of stools 11-15 times vs. 16-20 times; number of stools 11-15 times vs. 21+ times; and vomit.

Analysis of the best-performing symptom prediction models allowed us to identify the input features (core genome and intergenic SNPs, and accessory genes) most strongly correlated to each symptom (**Dataset S16**). Seventy-nine different features in total were selected as significantly correlated to at least one of the six symptom prediction models, with 68% being selected in two or more models (**Fig. 5**). No features were selected for all symptoms. All features associated with number of stools 11-15 times vs. 21+ times were found associated to at least one of the other five symptom prediction models. Forty-five accessory genes (nine known genes, *tufB_2*, *blc*, *pckA*, *luxR_2*, *hcpA_1*, *rpoS*, *dcuA*, *hpt*, *luxR*, and 36 hypothetical genes) and 28 core SNPs over 23 genes (14 known, *clpS*, *gshB*, *dapF*, *fabV_1*, *add*, *tufB*, *lpoA*, *phrB*, *yjcS*, *fabH1*, *cysG_2*, *padC*, *pepN*, *tadA_2*, and nine hypothetical genes) were identified as strongly associated to at least one of the symptoms. From the nine known accessory genes: four (*rpoS*, *hpt*, *luxR* and *pckA*) were found in the vomit model; *dcuA* was found in the abdominal pain model; *hcpA_1* was found only in the number of stools 11-15

times vs. 16-20 times; *luxR_2* was found in two models (vomit and dehydration moderate vs severe); *blc* and *tufB_2* were found in three models (vomit, number of stools 11-15 times vs. 16-20 times and number of stools 11-15 times vs. 21+ times) with *tufB_2* also found in abdominal pain and diarrhoea duration <1 day vs. 1-3 days models. Six SNPs from the genes *tufB*, *dapF*, *clpS*, *gshB* and *fabV* were associated to three symptom prediction models (vomit, number of stools 11-15 times vs. 16-20 times and number of stools 11-15 times vs. 21+ times) with the SNPs from the genes *dapF* and *fabV* also associated with abdominal pain and diarrhoea duration <1 day vs. 1-3 days and the SNP from the gene *tufB* associated with dehydration moderate vs severe.

Figure 5. Undirected graph network illustrating the genomic features associated with clinical symptom models for *V. cholerae*. Node colour denotes the genomic determinant category, (i.e. accessory genes and/or core genome coding, and intergenic SNPs) identified by machine learning. Nodes labelled with numbers corresponding to specific genes associated with each genomic determinant, are detailed in the "Genes Legend", while unnumbered nodes are related to unannotated (hypothetical) genes. The clinical symptom models are highlighted in different colours in nodes with a white centre circle and explained in the legend "Symptoms Legend" featuring abdominal pain; dehydration "moderate" vs "severe"; duration of diarrhoea

<1 day vs. 1-3 days; number of stools 11-15 times vs. 16-20 times; number of stools 11-15 times vs. 21+ times; and vomit.

Among the 45 accessory genes linked to clinical symptoms, six hypothetical genes were also statistically significant in distinguishing the two lineages.”

Methods lines 965-994: “The pipeline first removes features that are either present or absent in all the samples. Second, to measure the influence of confounding effects in the data, it uses a two-sided chi-square test of independence to measure the dependency between the confounding effects (sex of patient, age of patients, year of collection, location of patient, serology of *Vibrio cholerae*) and the phenotype classes (p-value < 0.01 with Bonferroni correction); if the null hypothesis is rejected (i.e. there is a dependency between the confound effect and the phenotype) the pipeline checks if there are features that are dependent on the confounding effect again based on a two-sided Chi-square test of independence (p-value < 0.01 with Bonferroni correction); if there are features where the null hypothesis is rejected, these features are removed from the analysis. Next, the pipeline oversamples the minority class using a SMOTE approach. Then based on the oversampled data it selects the most important features using a two-sided Fisher exact test (p-value < 0.1). This process is done in two parts: i) to improve randomization in the pipeline and avoid confounding effects, a loop over 1000 different random seeds is used for the SMOTE approach in order for it to have different initializations; for each loop the most important features are selected based on the Fisher exact test; ii) then, the features that are selected in over 75% of the different initializations are deemed important and a random initialization is selected that contains all these important features to be used for the prediction models. Next, a panel of machine learning methods (logistic regression (LR), linear support vector machine (L-SVM), radial basis function support vector machine (RBF-SVM), extra tree classifier, random forest, Adaboost and XGboost) was used to predict the clinical symptoms classes based on the pre-selected features described above. The hyperparameters used were:

- Logistic Regression: inverse of regularization strength C = [0.0001, 0.001, 0.01, 0.1, 1, 10, 100, 1000, 10000];
- Linear SVM: penalty parameter of the hinge loss error C = [0.0001, 0.001, 0.01, 0.1, 1, 10, 100, 1000, 10000];
- Random Forests, Extra Trees and Adaboost: Number of estimators = [2, 4, 8, 16, 32, 64, 128, 256];
- Non-linear SVM with RBF kernel: γ (RBF kernel coefficient) = [0.0001, 0.0001, 0.001, 0.01, 0.1, 1] and C (L2 penalty parameter) = [0.0001, 0.001, 0.01, 0.1, 1, 10, 100, 1000, 10000];
- XGBoost: Number of estimators = [2, 4, 8, 16, 32, 64, 128, 256] and learning rate = [0.0001, 0.001, 0.01, 0.1, 1].”

Dataset S14 legend: “**Dataset S14.** P-values from a two-sided Chi-square test of independence to measure the dependency between the clinical symptoms’ phenotypes (abdominal pain, vomit, number of stools 11-15 times vs. 21+ times, number of stools 11-15 times vs. 16-20 times, dehydration moderate vs severe and diarrhoea duration <1day vs 1-3 days) and the confound effects (sex of patient, age of patient, location of patient, year of collection and serology of *Vibrio cholerae*). Red-marked p-values indicate significant associations post-correction.”

Figure S18. Nemenyi *post-hoc* tests. Comparison of the performance of the 5 classifiers and 2 meta-methods, using their average ordinal rank over the clinical symptom analysed based on six performance metrics (A) AUC, (B) accuracy, (C) sensitivity, (D) specificity, (E) Cohen's kappa score and (F) precision for *V. cholerae*. The x-axis indicates the average ordinal rank of the machine learning methods. The scale is from 1 (best rank) to 7 (worst rank). The ordinal rank of a classifier is defined as follows: the ML method with the best AUC is given rank 1, the second-best AUC rank 2 and the n -th AUC best rank n , with n being the number of machine learning methods used. For each clinical symptom, the methods are ranked between 1 (highest AUC) and 7 (lowest AUC), since in this case there are 7 machine learning methods used. Next, for each method, the ranks are averaged based on the six clinical symptoms studied. The critical distance (CD) is defined based on the Nemenyi *post-hoc* test, all the methods that fall in the same bold bar below the axis are considered statistically equivalent based on the CD value.

Figure S19. A supervised machine learning pipeline successfully predicts the clinical manifestations of hospitalized patients based on genomic determinants extracted from BD-1.2 isolates obtained from those same patients during hospitalization. Machine learning performance results for four performance indicators: (A) accuracy, (B) sensitivity, (C) specificity, (D) Cohen's Kappa Score and (E) precision from 30 training runs for each clinical symptom. The results shown are for the best classifier Logistic Regression, as defined by the Nemenyi test (**Fig. S18**). Predictive models were generated for six different clinical symptoms (X axis): abdominal pain; dehydration “moderate” vs “severe”; duration of diarrhoea <1 day vs. 1-3 days; number of stools 11-15 times vs. 16-20 times; number of stools 11-15 times vs. 21+ times; and vomit.

2) The authors use genome-scale metabolic modelling to elucidate the relevance of key mutations they have identified for the metabolic activity of *V. cholerae*. For this purpose, they use a *V. cholerae* metabolic model reconstructed for the substrain O1 which they enrich with information for metabolic subsystems from *Escherichia coli* and other bacterial strains. I wonder why the authors did not simply use an automated tool for metabolic network reconstruction like CarveMe or gapseq. The advantage would be that they could reconstruct metabolic models that are specific to their isolates, the models directly come with proper pathway annotation and on top they could directly link SNPs to presence/absence of reactions if build a unique model for each of their isolates, though I'm not sure if there is sufficient genomic difference between the isolates for the latter analysis.

We thank the Reviewer for this question (a similar one was made by Reviewer 3, question 2). We originally opted to use a published model as it already had been experimentally validated. However, we agree that strain-specific models can offer additional insight into the metabolism of the strains. Thus, we have now included a strain-specific analysis into our manuscript. We have used the software CarveMe to generate 129 strain-specific models using protein fasta files generated by Prokka v1.14.6. CarveMe, also suggested by Reviewer 4, was chosen as it is the most utilised, ahead of gapseq and Bactabolize (465 citations for CarveMe, vs 118 for gapseq, and 2 for Bactabolize). CarveMe was run using the CPLEX solver and a Gram negative template, with gap filling for LB and M9 media using the command: 'carve input.faa --gapfill M9, LB -u gramneg --solver cplex --output model.xml'. The strain-specific models were then subjected to three analyses. Respectively:

- (i) Gene Essentially analysis: for each gene identified by the statistical analysis of lineages and machine learning analysis of clinical symptoms, we performed knock-down in each of the 129 models and compared the biomass to wildtype to assess whether the gene was essential for the model.
- (ii) Flux variability analysis: for each gene selected by the statistical analysis of lineages and machine learning analysis of clinical symptoms, we performed knockdowns in each of the 129 models and compared the flux span of each reaction to the wild type, identifying significant changes in reaction flux.
- (iii) Flux balance analysis. for each gene selected by the statistical analysis of lineages and machine learning analysis of clinical symptoms), we performed a knockdown in each of the 129 models. We then compared the metabolite yields of each model to those of the wild type.

Overall, the number of genes and reactions in the strain-specific models was slightly higher than the genes in iAM-vc960 and varied between 973 – 988 genes and 1989 – 2163 reactions, with a mean of 984 genes and 2130 reactions. The number of metabolites in the strain-specific models was lower than that of the iAN-vc960 model with a range of 1321 – 1433 and mean 1411. The number of genes, reactions and metabolites statistically differed between BD-1.2 and BD-2 lineages (p -values < 0.0001, Mann Whitney U test).

For the lineage-based analysis, we used as input the core genome genes mapped to the non-synonymous SNPs found to be exclusively present in one lineage (i.e. present in BD-1.2 and absent in BD-2 or conversely, present in BD-2 and absent in BD-1.2). We found 14 metabolic genes which were present in the strain-specific GSM models (*clcA*, *mak*, *suhB*, *murl*, *glmM*, *appC*, *argG*, *ftsI*, *licH*, *phhA*, *dltA_1*, *putA*, *ycbB*, *hudF_2*). Of these 10 were in the generalised model (*clcA*, *mak*, *suhB*, *murl*, *glmM*, *appC*, *argG*, *ftsI*, *licH*, *phhA*) and four were not (*dltA_1*, *putA*, *ycbB*, *hudF_2*). Three genes present in the generalised model (*dsbD*, *cysG_1* and *cob*) were not present in the strain-specific models. Comparing the gene essentiality results, *murl* was found to be essential in minimal media only in the generalised model. In the strain-specific models, it was essential in both rich and minimal media in a small proportion of models ($n=16$,

12%) and non-essential in both rich and minimal media in all other models. Experimental data from literature (Karp *et al.* 2019) shows essentiality of *murl* in the reference strain N16961 indicating some metabolic behavioural differences in our strains regarding this gene. Similarly, *glmM* was essential only in minimal media in the generalised model but essential in rich and minimal media in all the strain-specific models, consistent with its behaviour in the N16961 (Karp *et al.* 2019). Finally, *clcA*, which was essential in the generalised model was non-essential in all the strain-specific models, consistent with experimental results for the reference strain, where *clcA* is non-essential (Karp *et al.* 2019). In the flux variability analysis, which assessed whether gene knockouts significantly alter the reaction fluxes through the GSM models, a few changes were found in the strain-specific models compared to the generalised model. Knockouts of *suhB*, *phhA* and *licH* resulted in significant flux changes in the strain-specific models but not in the generalised models. Interestingly the gene *clcA*, which generated significant flux changes in the generalised model, also caused significant flux changes in most of the BD-2 strain-specific models (91%) but only in the small proportion of the BD-1.2 models (5%).

Regarding metabolite yield changes assessed by FBA, *murl* and *mak* knockouts affected metabolite yield in the generalised model but not in strain-specific models. Conversely, *suhB* and *clcA* knockouts impacted metabolite yield in the strain-specific models but not in the generalised model.

Regarding the clinical symptom-based analysis, 11 over the genes identified by machine learning were identified as metabolic genes in the strain-specific models (*add*, *dapf*, *dcuH*, *gshB*, *hpt*, *pckA*, *pepN*, *cysG_2*, *padC*, *tufB*, *tufB_2*). Among these, seven were also present in the generalised model (*add*, *dapf*, *dcuH*, *gshB*, *hpt*, *pckA*, *pepN*) while four were (*cysG_2*, *padC*, *tufB*, *tufB_2*) unique to the strain-specific models. Two genes that were present in the generalised model (*cdgL* and *fabH1*) are absent in strain-specific models.

Comparing the gene essentiality analysis between strain-specific models and the generalised model, *dapF* was found essential in rich media for 93% of strain-specific models, whereas it was non-essential in the generalised model. This finding aligns with experimental data (Karp *et al.* 2019), showing this gene as essential for the reference N16961, and indicating consistency between strain-specific model results and experimental observations. In flux variability analysis, all genes showing significant flux changes in the generalised model exhibited similar changes in strain-specific models. For the metabolites' yields analysis changes, while *dapF* knockouts showed significant metabolite yield changes in both generalised and strain-specific models, *gshB* knockouts did not show any metabolite yield changes in strain-specific models.

Finally, to clarify the reviewer's point "For this purpose, they use a V. cholerae metabolic model reconstructed for the substrain O1 which they enrich with information for metabolic subsystems from Escherichia coli and other bacterial strains.". We apologise for any confusion regarding this, the metabolic pathways and reactions within the model were not altered from the published iAM-Vc960 model. In earlier version of our analysis, we used metabolic subsystems to categorise our metabolic genes into subsystem groups for a statistical network-based analysis, which is what this description in the methods referred to. However, as this did not generate important results, they were not included in the submitted manuscript, but the methods were erroneously left in. We apologise for this oversight, and this has now been corrected in the methods lines 883-919. We have now included a list of pathways, molecular functions and biological processes for the genes of interest based only on SMART tables for the reference *Vibrio cholerae* O1 biovar El Tor strain N16961 genome taken from the BioCyc database (Karp *et al.* 2019).

We thank the Reviewer for suggesting this additional analysis which has highlighted significant differences across strains in our cohort. Comparison between the curated generalised model

(iAM-Vc960) and draft strain-specific models were added as Supplementary Note 5, manuscript results lines 256-267 and 493-513 and in the methods lines 883-919; all edits are also shown below. Details of the results obtained with the strain-specific models were added to Datasets S12 and S18.

Supplementary Note 5: “In addition to using the generalised *V. cholerae* O1 GSM model (iAM-vc960) to analyse the metabolic functions of our strains, draft strain-specific models were generated for each isolate using CarveMe (Machado *et al.* 2018). Overall, the number of genes and reactions in the strain-specific models was slightly higher than the genes in iAM-vc960 and varied between 973 – 988 genes and 1989 – 2163 reactions, with a mean of 984 genes and 2130 reactions. The number of metabolites in the strain-specific models was lower than that of the iAM-Vc960 model with a range of 1321 – 1433 and mean 1411. The number of genes, reactions and metabolites statistically differed between BD-1.2 and BD-2 lineages (p -values < 0.0001, Mann Whitney U test).

For the lineage-based analysis, we used as input the core genome genes mapped to the non-synonymous SNPs found to be exclusively present in one lineage (i.e. present in BD-1.2 and absent in BD-2 or conversely, present in BD-2 and absent in BD-1.2). We found 14 metabolic genes which were present in the strain-specific GSM models (*clcA*, *mak*, *suhB*, *murl*, *glmM*, *appC*, *argG*, *ftsI*, *licH*, *phhA*, *dltA_1*, *putA*, *ycbB*, *hudF_2*). Of these 10 were in the generalised model (*clcA*, *mak*, *suhB*, *murl*, *glmM*, *appC*, *argG*, *ftsI*, *licH*, *phhA*) and four were not (*dltA_1*, *putA*, *ycbB*, *hudF_2*). Three genes present in the generalised model (*dsbD*, *cysG_1* and *cob*) were not present in the strain-specific models. Comparing the gene essentiality results, *murl* was found to be essential in minimal media only in the generalised model. In the strain-specific models, it was essential in both rich and minimal media in a small proportion of models ($n=16$, 12%) and non-essential in both rich and minimal media in all other models. Experimental data from literature (Karp *et al.* 2019) shows essentiality of *murl* in the reference strain N16961 indicating some metabolic behavioural differences in our strains regarding this gene. Similarly, *glmM* was essential only in minimal media in the generalised model but essential in rich and minimal media in all the strain-specific models, consistent with its behaviour in the N16961 (Karp *et al.* 2019). Finally, *clcA*, which was essential in the generalised model was non-essential in all the strain-specific models, consistent with experimental results for the reference strain, where *clcA* is non-essential (Karp *et al.* 2019). In the flux variability analysis, which assessed whether gene knockouts significantly alter the reaction fluxes through the GSM models, a few changes were found in the strain-specific models compared to the generalised model. Knockouts of *suhB*, *phhA* and *licH* resulted in significant flux changes in the strain-specific models but not in the generalised models. Interestingly the gene *clcA*, which generated significant flux changes in the generalised model, also caused significant flux changes in most of the BD-2 strain-specific models (91%) but only in the small proportion of the BD-1.2 models (5%).

Regarding metabolite yield changes assessed by FBA, *murl* and *mak* knockouts affected metabolite yield in the generalised model but not in strain-specific models. Conversely, *suhB* and *clcA* knockouts impacted metabolite yield in the strain-specific models but not in the generalised model.

Regarding the clinical symptom-based analysis, 11 over the genes identified by machine learning were identified as metabolic genes in the strain-specific models (*add*, *dapf*, *dcuH*, *gshB*, *hpt*, *pckA*, *pepN*, *cysG_2*, *padC*, *tufB*, *tufB_2*). Among these, seven were also present in the generalised model (*add*, *dapf*, *dcuH*, *gshB*, *hpt*, *pckA*, *pepN*) while four were (*cysG_2*, *padC*, *tufB*, *tufB_2*) unique to the strain-specific models. Two genes that were present in the generalised model (*cdgL* and *fabH1*) are absent in strain-specific models.

Comparing the gene essentiality analysis between strain-specific models and the generalised model, *dapF* was found essential in rich media for 93% of strain-specific models, whereas it was non-essential in the generalised model. This finding aligns with experimental data (Karp *et al.* 2019), showing this gene as essential for the reference N16961, and indicating consistency between strain-specific model results and experimental observations. In flux variability analysis, all genes showing significant flux changes in the generalized model exhibited similar changes in strain-specific models. For the metabolites' yields analysis changes, while *dapF* knockouts showed significant metabolite yield changes in both generalized and strain-specific models, *gshB* knockouts did not show any metabolite yield changes in strain-specific models."

Results Lines 256-267: "To further elucidate the metabolic differences between the BD-1.2 and BD-2 lineages, we repeated our previous analyses done on the generalized model using strain-specific models automatically generated by CarveMe (Machado *et al.* 2018). Gene essentiality analysis concurred with the general model (iAM-Vc960), with only a small number of differences (**Dataset S12**). The effect of *murl* gene knockouts differed between lineages, proving non-essential in 94% of BD-1.2 lineage models but only in 76% of BD-2 lineage models. Flux variability analysis of the individual models revealed that *clcA* knockouts led to significant changes in the flux span of the CLt3_2pp reaction, which controls chloride transport, in 96% BD-2 models compared to just 5% of BD-1.2 models. The *clcA* gene has been linked to bacterial acid resistance and it has been suggested that changes to the expression/repression of this gene may help facilitate survival during movement through the intestinal tract (Cakar *et al.* 2018). Similarly, flux balance analysis indicated that metabolite yield was changed differently across lineages in response to knocking out *clcA*, with the metabolite yield of chloride reduced to 0 in 95% of BD-1.2 isolates."

Results Lines 493-513: "To further investigate the link between metabolic gene variations and the clinical symptoms observed in different strains, we utilized draft strain-specific models generated with CarveMe (Machado *et al.* 2018). The gene essentiality analysis results were largely consistent with those of the general model (iAM-Vc960), with only a few differences noted (**Dataset S18**). The effect of *dapF* gene knockouts varied between models with the gene being essential in 93% (n=20) and non-essential in 7% (n=9) of the models. Comparing symptoms between the 'essential' and 'non-essential' groups, dehydration was significantly more severe in the 'non-essential' group (Fisher exact test p value =0.05). All strains in this group exhibited severe dehydration, suggesting a link between non-essentiality of the *dapF* gene and the severity of *V. cholerae* symptoms. In relation to this, the flux balance analysis revealed changes in metabolite yields associated with the genes *dapF* and *cysG_2* across all strain-specific models. For *dapF*, altered metabolite yields were predominantly observed in strains where *dapF* was essential, while knocking out *dapF* in non-essential models had minimal impact on the metabolite yields of murein-related metabolites. This indicates metabolic adaptations linked to bacterial survival in these strains, potentially contributing to more severe disease outcomes. Additionally, knocking out the *padC* gene resulted in significant changes in metabolite yields only in the NGICDV-066 strain. Although conclusions drawn from a single strain are limited, it is notable that this isolate exhibited the most severe clinical symptoms across all measured symptoms, except for the duration of diarrhoea (presence of vomiting, presence of abdominal pain, number of stools (21+ times), presence of severe dehydration, duration of diarrhoea 1-3 days). Flux variability analysis in individual models indicated consistent behaviour across all strain-specific models regarding gene knockouts associated with clinical symptoms. Specifically, five gene knockouts (*add*, *dapF*, *gshB*, *padC*, *pckA*) showed significant flux span changes in all models."

Methods Lines 883-919: "Genome-scale metabolic model

All simulations were performed using the Python cobra toolkit v0.26.2. The analysis was conducted on both a manually curated and validated model of *V. cholerae* O1 N16961, iAM-Vc960, taken from (Abdel-Haleem *et al.* 2020) and on automatically generated draft strain-specific GSM models. The strain-specific draft models were generated using CarveMe (Machado *et al.* 2018). CarveMe was run using the CPLEX solver and gram negative template, with gap filling for LB and M9 media using the command: 'carve input.faa --gapfill M9, LB -u gramneg --solver cplex --output model.xml'. Gene essentiality, FVA and FBA analyses as described below were conducted on genes of interest in the generalised iAM-Vc960 and in each of the 129 draft strain-specific models, based on the analysis pipeline in (Pearcy *et al.* 2021).

For all gene essentiality, FBA and FVA analyses, a knockout model for each gene of interest was constructed by blocking all corresponding reactions to zero, given that the reaction is not catalysed by an isozyme. We considered the essentiality of a gene under both rich medium conditions and M9 minimal medium conditions. To mimic rich medium conditions, the model was constrained to allow all carbon sources into the system, with a fixed uptake rate of 1 mmol/gDCW/h. If a feasible solution exists, while maximizing the biomass equation as the objective function, then the knockout of the gene was not essential. To mimic M9 minimal medium conditions, the model was constrained so one individual carbon source had a maximum uptake of 10 mmol/gDCW/h. This simulation (minimal medium condition) was repeated for each carbon source in the model. The genes whose corresponding knockout model achieved a growth rate of 0.0001 h⁻¹ or less were considered essential. Flux variability analysis (FVA) was applied to the wild-type model and each knockout model using the cobra toolbox in python (Ebrahim *et al.* 2013). FVA calculates the minimum and maximum flux through each reaction in the model, given a set of constraints, resulting in the range of possible fluxes for each reaction (flux span). FVA was simulated using glucose as the only carbon source in aerobic minimal M9 medium conditions. Note that reaction loops in the solution were not allowed. A gene knockout was considered to significantly affect the flux if the flux span of at least one reaction was changed by greater than 10% compared to the wildtype solution. For the FBA analysis, a drain reaction (i.e., a reaction that consumes the metabolite of interest) was added to the GSM model for each metabolite. The maximum theoretical yield of each metabolite was calculated by setting its corresponding drain reaction as the objective function, with glucose as the only carbon source in aerobic minimal M9 medium conditions. All metabolites contained within the model were considered in the FBA analysis. In iAM-Vc960 this was 1,741 different metabolites, whilst in the draft strain-specific GSM models the number of metabolites spanned the range 1321-1433. The simulations were carried out for the wild-type model and each gene knockout model. A gene knockout was considered to significantly affect metabolite yield if the yield of at least one metabolite was reduced to zero, given that it was non-zero in the wildtype. For each of the selected genes of interest, molecular function, pathways and biological processes were taken from the BioCyc database (Karp *et al.* 2019) using the SMART tables for *Vibrio cholerae* O1 biovar El Tor strain N16961. These were added to **Datasets S11, S12, S17 and S18** to give context to the analysis results."

Abdel-Haleem, Alyaa M, Vaishnavi Ravikumar, Boyang Ji, Katsuhiko Mineta, Xin Gao, Jens Nielsen, Takashi Gojobori, and Ivan Mijakovic. 2020. 'Integrated metabolic modeling, culturing, and transcriptomics explain enhanced virulence of *Vibrio cholerae* during coinfection with enterotoxigenic *Escherichia coli*', *mSystems*, 5: e00491-20.

Ebrahim, Ali, Joshua A Lerman, Bernhard O Palsson, and Daniel R Hyduke. 2013. 'COBRApy: constraints-based reconstruction and analysis for python', *BMC Systems Biology*, 7: 1-6.

Karp, P. D., R. Billington, R. Caspi, C. A. Fulcher, M. Latendresse, A. Kothari, I. M. Keseler, M. Krummenacker, P. E. Midford, Q. Ong, W. K. Ong, S. M. Paley, and P. Subhraveti. 2019.

'The BioCyc collection of microbial genomes and metabolic pathways', Briefings in Bioinformatics, 20: 1085-93.

Machado, Daniel, Sergej Andrejev, Melanie Tramontano, and Kiran Raosaheb Patil. 2018. 'Fast automated reconstruction of genome-scale metabolic models for microbial species and communities', Nucleic Acids Research, 46: 7542-53.

Pearcy, Nicole, Yue Hu, Michelle Baker, Alexandre Maciel-Guerra, Ning Xue, Wei Wang, Jasmeet Kaler, Zixin Peng, Fengqin Li, Tania Dottorini, and Xiaoxia Lin. 2021. 'Genome-scale metabolic models and machine Learning reveal genetic determinants of antibiotic resistance in *Escherichia coli* and unravel the underlying metabolic adaptation mechanisms', mSystems, 6: e00913-20.

3) While the authors link mutations in metabolic genes to potential functional outcomes (e.g. changes in growth rate, reaction flux or metabolite production) it would be great to see if those mutations particularly strongly influence the production of individual metabolites and/or growth. Thus, the authors should compare their outcomes to changes in growth rates and/or production of individual metabolites from randomly selected non-essential genes.

We thank the reviewer for this inciteful suggestion. As requested, we have completed an additional analysis, and have based the analysis on the generalised iAM-Vc960 model, for simplicity. We took 100 randomly selected genes from the iAM-Vc960 model and conducted the same analysis pipeline as we used for the ML selected genes, Table 1 below. Comparing the 92 non-essential genes from this random sampling with the non-essential genes from our ML selected pipelines we found that the average growth rates, number of FVA reactions affect and average flux span changes did not significantly differ between the randomly selected genes and our ML selected genes. However, comparing metabolite production, genes that were found to be important in discriminating between the lineages BD-1.2 and BD-2 were found to affect more metabolites on average than the randomly selected genes (p-value 0.0429, Mann Whitney U test, two-sided). This finding has been added to the results on lines 252-255 and below.

Table 1	Growth	% with significant FVA	Average Number of FVA reactions per significant gene	Average % flux span relative to wildtype	% FBA with significant results	Average number of metabolites affected
Random Genes	1.244378	0.326087	40.16667	0.12183	0.358696	24.20588
Lineage genes	1.260928	0.357143	33.2	0.10319	0.285714	43.5
Clinical genes	1.261737	0.555556	11.333333	0.201158	0.222222	28.5

Results lines 252-255: "Interestingly, the average number of metabolite yields affected by knockouts of the genes discriminating lineages was significantly higher than a random selection of 100 metabolic genes (p-value 0.0429, Mann Whitney U test, two-sided), indicating a stronger influence on metabolite production for this subset of genes."

4) The authors seem to have made only parts of their code available (on <https://github.com/tan0101/VibrioCARE>). While the machine learning part is included, the metabolic modelling part is lacking and should be added.

Apologies for this oversight, this script (GSAnalysis.py) and the strain-specific draft models have now been added to the GitHub repository <https://github.com/tan0101/VibrioCARE>.

Minor:

1) Some of the Supplements were converted to PDF and so were not really readable (e.g. several thousand pages at the end of the main manuscript file or Supplementary Table S6).

Apologies for this, these files were submitted as Supplementary Tables in excel spreadsheets and converted to PDF by the manuscript submission system. We have now instead submitted as Supplementary Datasets to avoid this issue.

Reviewer #4 (Remarks on code availability):

Only parts of the code have been made available. The metabolic modelling part is lacking.

Apologies for this oversight, this script (GSManalysis.py) and the strain-specific draft models have now been added to the GitHub repository <https://github.com/tan0101/VibrioCARE>.

REVIEWERS' COMMENTS

Reviewer #1 (Remarks to the Author):

I thank the authors for their comprehensive response to my comments. I am happy with the additional clarifications over disease severity and appreciate their incorporation of statistical models to take into account host confounding factors.

I have no further concerns.

Reviewer #1 (Remarks on code availability):

Other reviewers are better placed to do this.

Reviewer #2 (Remarks to the Author):

The authors have provided a very comprehensive review of their manuscript, and in the response to the reviewers' issues they have added new analysis. This is particularly relevant in the case of the metabolic analysis. All in all, the conclusions are more robust.

The remaining limitation is the lack of experimental data to validate some of the conclusions. (i) The metabolic modelling does not replace running some Biolog-based experiments. Even the authors make the case for these experiments in their letter!. (ii) the data related to antimicrobial resistance is relevant; however antibiotics expelled by bcr are not known and, therefore, the authors have in fact only partially correlated the genome analysis with the antibiotic pattern data (this was also the reason to suggest testing detergents despite they are not part of the normal panel of drugs tested). The experiments to assess the function of bcr are fairly simple by, for example, expressing the gene in a surrogate background and/or testing the resistance profile of mutants. It will be extremely informative to mobilize this gene to a strain that it does not encode it to assess the functionality in a *Vibrio* background. All these are very simple experiments that will strengthen the bioinformatics analysis.

Unfortunately, it is apparent that the authors are inclined towards not doing these assays. In this case, the authors should add to the discussion section a comprehensive section detailing the limitations of the study by not running any validation experiment in particular for the metabolism, antibiotic resistance, and virulence.

Reviewer #3 (Remarks to the Author):

The authors have done a very thorough and comprehensive job adding additional analyses and revising the paper. I particularly appreciate the addition of strain-specific models built through CarveMe. I have no further comments and would recommend the paper for publication after the revision.

Reviewer #4 (Remarks to the Author):

The authors have very comprehensively addressed all of my comments.

Reviewer #4 (Remarks on code availability):

All necessary code is available now.

Point-by-point response to reviewers:

We thank the Reviewers for their useful comments, which no doubt contributed to make this a better paper. In the following, a point-by-point response to all the questions and comments is provided. The original questions are in blue, our replies in black.

Reviewer #1 (Remarks to the Author):

I thank the authors for their comprehensive response to my comments. I am happy with the additional clarifications over disease severity and appreciate their incorporation of statistical models to take into account host confounding factors. I have no further concerns.

Reviewer #1 (Remarks on code availability):

Other reviewers are better placed to do this.

We thank the reviewer for their comments.

Reviewer #2 (Remarks to the Author):

The authors have provided a very comprehensive review of their manuscript, and in the response to the reviewers' issues they have added new analysis. This is particularly relevant in the case of the metabolic analysis. All in all, the conclusions are more robust.

The remaining limitation is the lack of experimental data to validate some of the conclusions. (i) The metabolic modelling does not replace running some Biolog-based experiments. Even the authors make the case for these experiments in their letter!. (ii) the data related to antimicrobial resistance is relevant; however antibiotics expelled by *bcr* are not known and, therefore, the authors have in fact only partially correlated the genome analysis with the antibiotic pattern data (this was also the reason to suggest testing detergents despite they are not part of the normal panel of drugs tested). The experiments to assess the function of *bcr* are fairly simple by, for example, expressing the gene in a surrogate background and/or testing the resistance profile of mutants. It will be extremely informative to mobilize this gene to a strain that it does not encode it to assess the functionality in a *Vibrio* background. All these are very simple experiments that will strengthen the bioinformatics analysis.

Unfortunately, it is apparent that the authors are inclined towards not doing these assays. In this case, the authors should add to the discussion section a comprehensive section detailing the limitations of the study by not running any validation experiment in particularly for the metabolism, antibiotic resistance, and virulence.

We thank the reviewer for this comment. We agree that several aspects of this paper would be interesting to explore further in future studies. We have added a comment on this to the Discussion on lines 691-694 and below:

Discussion Lines 691-694: "In this study, we have identified promising targets related to metabolism (*clcA*, *cysG*, *adh*), antimicrobial resistance (i.e. *bcr*, *blc*), and virulence (i.e. *ompU*, *skp*, *tamA*, *valS*). These targets show significant potential for further investigation through experimental studies."

Reviewer #3 (Remarks to the Author):

The authors have done a very thorough and comprehensive job adding additional analyses and revising the paper. I particularly appreciate the addition of strain-specific models built through CarveMe. I have no further comments and would recommend the paper for publication after the revision.

We thank the reviewer for their comments.

Reviewer #4 (Remarks to the Author):

The authors have very comprehensively addressed all of my comments.

Reviewer #4 (Remarks on code availability):

All necessary code is available now.

We thank the reviewer for their comments.